# Existence of a continental-scale river system in eastern Tibet during the late Cretaceous–early Palaeogene

Xudong Zhao[1], Huiping Zhang [1,2✉], Ralf Hetzel[3], Eric Kirby [4], Alison R. Duvall [5], Kelin X. Whipple [6], Jianguo Xiong[1], Yifei Li[7], Jianzhang Pang[1], Ying Wang[1], Ping Wang[8], Kang Liu[9], Pengfei Ma[10], Bo Zhang[11], Xuemei Li[1], Jiawei Zhang[1] & Peizhen Zhang[2,9]

The establishment of continental-scale drainage systems on Earth is largely controlled by topography related to plate boundary deformation and buoyant mantle. Drainage patterns of the great rivers in Asia are thought to be highly dynamic during the Cenozoic collision of India and Eurasia, but the drainage pattern and landscape evolution prior to the development of high topography in eastern Tibet remain largely unknown. Here we report the results of petro-stratigraphy, heavy-mineral analysis, and detrital zircon U-Pb dating from late Cretaceous–early Palaeogene sedimentary basin strata along the present-day eastern margin of the Tibetan Plateau. Similarities in the provenance signatures among basins indicate that a continental-scale fluvial system once drained southward into the Neo-Tethyan Ocean. These results challenge existing models of drainage networks that flowed toward the East Asian marginal seas and require revisions to inference of palaeo-topography during the Late Cretaceous. The presence of a continent-scale river may have provided a stable long-term base level which, in turn, facilitated the development of an extensive low-relief landscape that is preserved atop interfluves above the deeply incised canyons of eastern Tibet.

[1] Lhasa National Geophysical Observation and Research Station, State Key Laboratory of Earthquake Dynamics, Institute of Geology, China Earthquake Administration, Beijing, China. [2] Southern Marine Science and Engineering Guangdong Laboratory (Zhuhai), Zhuhai, China. [3] Institute of Geology and Palaeontology, University of Münster, Münster, Germany. [4] Department of Earth, Marine, and Environmental Sciences, University of North Carolina, Chapel Hill, Chapel Hill, NC, USA. [5] Department of Earth and Space Sciences, University of Washington, Seattle, WA, USA. [6] School of Earth and Space Exploration, Arizona State University, Tempe, AZ, USA. [7] Key Laboratory of Computational Geodynamics, College of Earth and Planetary Sciences, University of Chinese Academy of Sciences, Beijing, China. [8] School of Geography, Nanjing Normal University, Nanjing, China. [9] School of Earth Sciences and Engineering, Sun Yat-Sen University, Guangzhou, China. [10] Department of Earth Science and Engineering, Taiyuan University of Technology, Taiyuan, China. [11] State Key Laboratory of Geological Processes and Mineral Resources, School of Earth Sciences and Resources, China University of Geosciences (Beijing), Beijing, China. ✉email: huiping@ies.ac.cn

Many continental-scale river systems on Earth have evolved over prolonged geological histories[1–5] and their regimes were often established in tectonically stable settings[1,4]. Hence, the development of such drainage systems has been argued to reflect long-term maintenance of low-amplitude but long-wavelength topographic gradients[4,6]. The persistence of a stable base level thus provides a governing factor that enables reduction of topographic relief and promotes the development of continental-scale landscapes with subdued topography[2,4,7].

The high topography associated with the India-Eurasia collision zone hosts several great rivers (Fig. 1a), and the history of their drainage networks is central to debates about the uplift and landscape evolution of the Tibetan Plateau[8–12]. The Yangtze River – the largest river in Asia – emanates from eastern Tibet

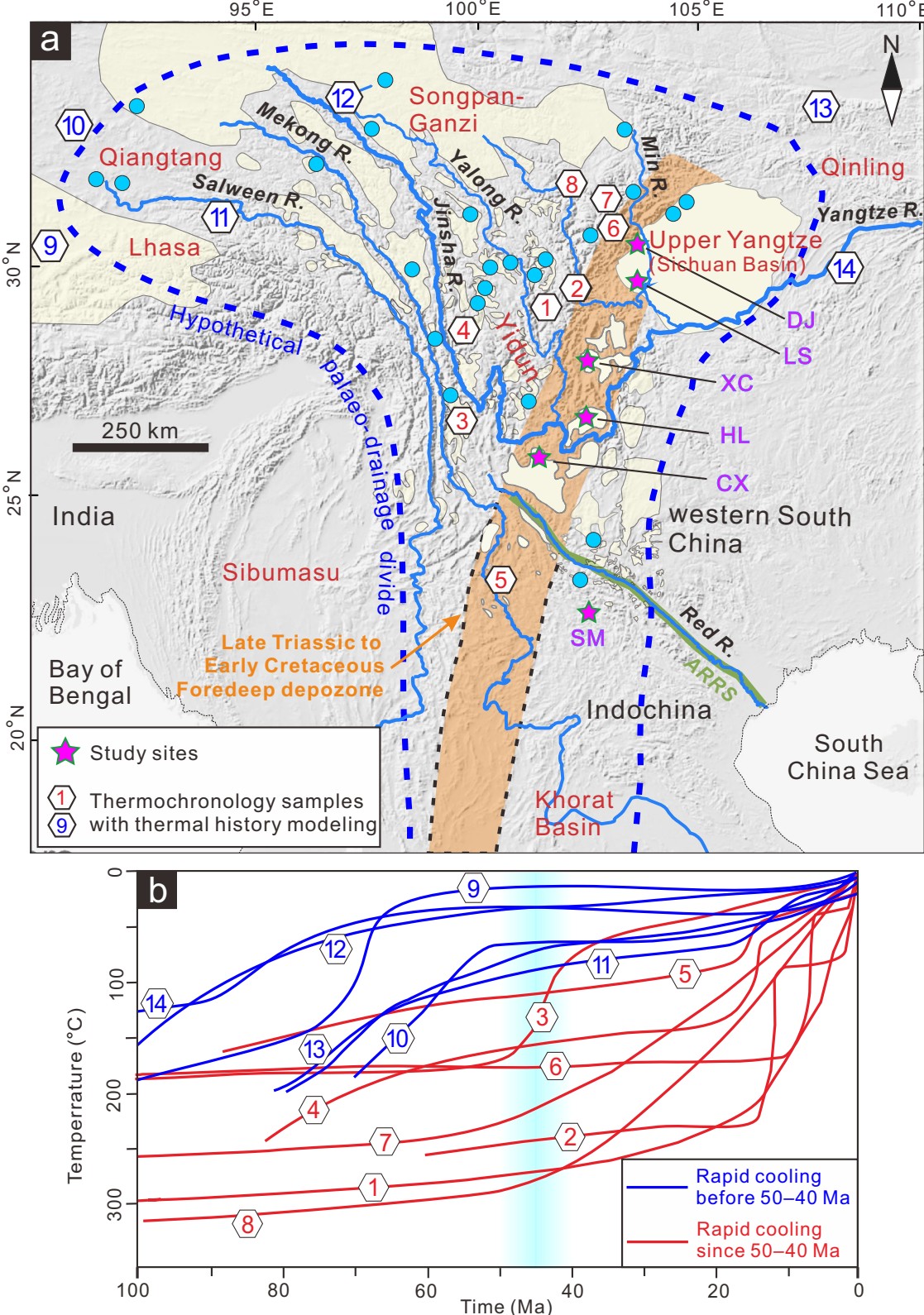

**Fig. 1 Location map of study area and thermal modelling results of thermochronological data in eastern Tibet and surrounding areas. a** Map showing main tectonic units (red text) and major river systems. Light yellow polygons indicate mapped low-relief surfaces (modified after refs. [20,61]). Blue circles correspond to AFT and AHe thermochronological ages of ~50–100 Ma from low-relief plateau areas[24,31–34,36,67]. Hexagons indicate sites with late Cretaceous–early Palaeogene cooling histories (shown in Fig. 1b). Dashed blue line represents the hypothetical drainage divide of late Cretaceous to early Palaeogene drainage system. Orange area marks the inferred location of a late Triassic to early Cretaceous foredeep depozone (cf. refs. [20,27]). Abbreviations for sites: ARRS Ailaoshan–Red River shear zone, DJ Dujiangyan area, LS Leshan area, XC Xichang Basin, HL Huili Basin, CX Chuxiong Basin, SM Simao Basin. **b** Red lines indicate slow cooling rates (~1 °C/Ma) during late Cretaceous to early Palaeogene, whereas blue lines show more rapid cooling rates (~5–10 °C/Ma) during the same period. Light blue vertical band indicates age range (~50–40 Ma) of the onset of rapid cooling in eastern Tibet. 1–ref.[24]; 2–ref.[12]; 3–ref.[36]; 4–ref.[65]; 5–ref.[66]; 6–ref.[35]; 7,8–ref.[28]; 9–ref.[31]; 10–ref.[37]; 11,12–ref.[32]; 13–ref.[34]; 14–ref.[38]. See Fig. 1a for localities. (The shaded-relief map is generated by Xudong Zhao using DEM data (https://www.ngdc.noaa.gov/mgg/topo/) and ArcGIS software).

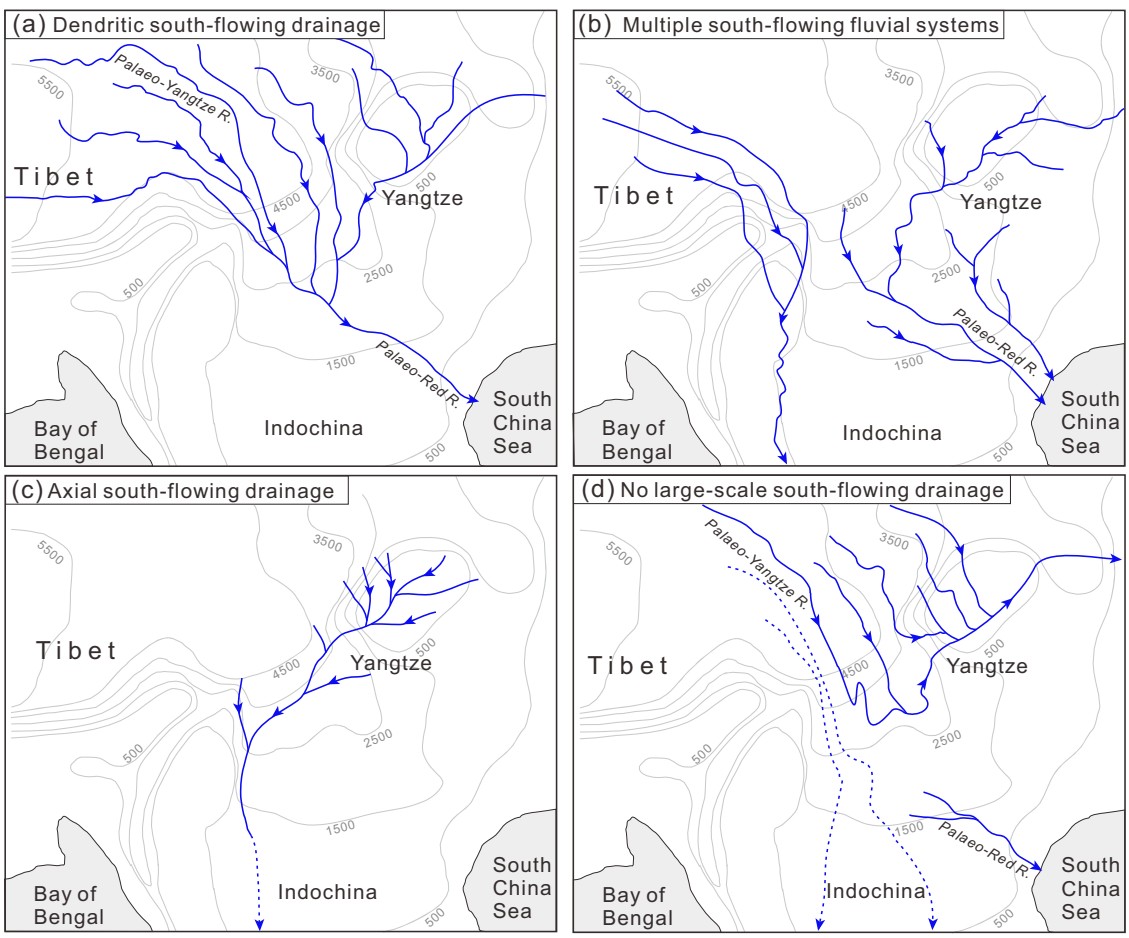

**Fig. 2 Postulated drainage scenarios in SE Tibet before establishment of the modern river network. a** Continental-scale, dendritic south-flowing river system which drained into the South China Sea[6,15]. **b** A south-flowing river system linking the Upper Yangtze terrane to the South China Sea[13]. **c** A south-flowing transcontinental drainage that developed on the western South China Block[16]. **d** An east-flowing river system similar to the modern drainage network, with no connection between the palaeo-Yangtze River and the palaeo-Red River[17,18] (or other south-flowing rivers).

before being successively joined by other major rivers and flowing into the East China Sea (Fig. 1a). The evolution of the Yangtze River system and its large tributaries, which drain the amalgamated terranes of continental China (e.g., Songpan-Ganzi, Yidun and Yangtze terranes), is intimately linked to Cenozoic uplift of the eastern Tibetan Plateau and is therefore of great interest for all studies attempting to understand the tectonic and landscape evolution of the plateau[3,8,9,13]. Geomorphic observations suggest that the modern rivers draining eastern Tibet were once tributaries to a big, southward-flowing river that had a Mississippi-like drainage pattern[6] (Fig. 2a). Many studies have addressed the origin and development of this river system from different perspectives, and it has been suggested that a palaeo-Red River once connected Tibet with the South China Sea[13–15]. However, both

the pattern and the age of this river system remain highly controversial (Fig. 2), with age estimates ranging from Cretaceous to Quaternary[11,13–16]. Moreover, the poor preservation of fluvial landscapes and a lack of studies focusing on regional sedimentary linkages have led some researchers to question the existence of a south-flowing river[17,18] (Fig. 2d).

The presence of a high-elevation, low-relief landscape in eastern Tibet has often been linked to drainage evolution[6,19]. Some interpreted these low-relief surfaces as a result of river incision into a pre-existing erosion surface[10,20–22]; others argued that the low-relief landscape formed at high elevation due to reduced erosional efficiency following drainage capture[19], highland denudation and basin filling within internal drainages[23], or smoothing of the highlands by efficient glacial erosion[24].

However, these previous interpretations were recently argued to be insufficient because the respective processes were only documented locally, and drainage capture and divide migration cannot form the observed low-relief landscape in eastern Tibet[21]. Thus, the question of how the low-relief landscape preserved in eastern Tibet has formed remains largely unresolved.

Late Cretaceous–early Palaeogene ($K_2$–$E_1$) terrestrial sediments are preserved in several basins in eastern Tibet[16,25,26] (Fig. 1a), enabling us to reconstruct the evolution of the drainage network prior to the establishment of high topography in the Late Cenozoic. Here, we characterise the provenance and dispersal pattern of the related drainage system prior to the India–Eurasia collision and show, for the first time, that much of the region now elevated as Tibetan Plateau, was drained by a low-gradient, continental-scale river that flowed into the Neo-Tethyan Ocean. We suggest that the relatively stable base level and long-term tectonic stability required for the maintenance of this river played a central role in the development of a low-relief region in the continental interior during Late Cretaceous–early Palaeogene time.

## Results

**Prolonged late Mesozoic tectonic stability in eastern Tibet.** The geologic provinces that comprise eastern Tibet represent distinct tectonostratigraphic terranes that were successively amalgamated during the closure of the Palaeo- and Meso-Tethyan oceans in the Triassic[27]. The Songpan-Ganzi and Yidun terranes, which now form much of eastern Tibet, were thrust eastward onto the Upper Yangtze terrane (equivalent to the Sichuan Basin) and western South China during the Late Triassic, forming an intracontinental foreland basin[16,27]. From the late Triassic to the early Cretaceous, this foreland basin continued to receive continental sediment during the waning of tectonism and the apparent decay of orogenic topography[28,29]. Consequently, a low-lying, long-wavelength topographic depression developed east of the Songpan-Ganzi and Yidun terranes[16,30]. On both flanks of this northeast-trending foredeep, low-relief bedrock surfaces were identified and hypothesised to represent a once-contiguous landscape[20] (Fig. 1a). Extensive low-temperature thermochronological data from these surfaces consistently yield late Cretaceous–early Palaeogene cooling ages[24,31–34] (Fig. 1a). Modelling of thermal histories from multiple thermochronologic systems revealed a long phase of slow cooling (~1 °C/Ma) from late Cretaceous to early Eocene time[24,28,33,35,36] (red lines in Fig. 1b), suggested to reflect a period of tectonic stability prior to the India-Asia collision[28,35]. In contrast, moderate cooling rates of 5–10 °C/Myr during the same time period in the Qiangtang, eastern Lhasa, northern Qinling, and eastern Upper Yangtze terranes[31,32,34,37,38] (blue lines in Fig. 1b) suggests modest to rapid exhumation consistent with sustained topographic relief. Such regional differences in erosion/exhumation rates could be explained by a large-scale south-flowing river system, the main axis of which followed the foredeep depozone between eastern Tibet and South China (Fig. 2c), as argued by Deng et al.[16].

**Provenance signature of sedimentary basins.** The late Cretaceous–early Palaeogene strata exposed in the investigated basins along the foredeep depozone (i.e., SW Sichuan, Xichang, Huili and Chuxiong basins; Fig. 1a) represent the youngest terrestrial clastic deposits at the eastern margin of Tibet, except for spatially limited Quaternary sediments[25]. The late Cretaceous–early Palaeogene age of the sedimentary rocks is based on diagnostic fossil assemblages (mainly ostracodes and charophyta) and on the youngest U-Pb ages from zircon age populations (see Supplementary Note). The deposits are consistently characterised by red, thick-bedded, fine- to medium-grained sandstone interbedded with siltstone and mudstone. Minor

conglomerates are restricted to the Dujiangyan area (Supplementary Fig. 1). Evaporite-bearing sequences are rare, although lacustrine-dominated sequences were deposited during arid climate conditions (Supplementary Note), which implies that flow discharge was relatively high and likely maintained an interconnected drainage network[3]. Thick sandstone beds in the sandstone-rich subsection are lenticular or tabular with erosional contacts and are laterally continuous over scales of tens to hundreds of metres. Cross stratification, parallel bedding and upward fining trend are common within individual sandstone bed, which are interpreted as fluvial deposits[5,16,25,26] (see Supplementary Note for details). Sandwiched mudstone and siltstone layers contain desiccation cracks, calcareous nodules and occasionally small-scale ripple marks, likely representing floodplain deposits[5] (Supplementary Fig. 1 and Supplementary Note). Such fluvial-lacustrine facies association has been ascribed to be typical of the deposits of hydrologically open lakes associated with perennial river systems[39,40], which makes the existence of a large throughgoing river system much likely. Palaeocurrent indicators reinforce this interpretation, showing a dominantly southward and southeastward flow direction[16,30]. The biofacies records in these basins provide further evidence. It shows that the fossil is dominated by ostracodes, freshwater lamellibranchia, fish specie and plant fossil which are diagnostic paleontological assemblages in many other hydrologically open lakes[39,40].

Petrographic analysis of basin strata shows that all $K_2$–$E_1$ samples contain abundant quartz grains, sedimentary and metamorphic rock fragments, but lack fragments derived from volcanic rocks (Supplementary Fig. 2), suggesting mature sandstone compositions. On quartz-feldspar-lithic grain diagrams, most samples plot in the recycled orogenic provenance fields, except for five samples from Leshan area, which plot near the boundary between continental block and recycled orogen, typical of sandstones deposited in a tectonically stable continent. Heavy mineral spectra from all samples are similar and dominated by stable minerals, including magnetite, haematite, zircon and leucoxene, and all lack readily weathered minerals (Supplementary Fig. 3). This pattern is consistent with the abundance of quartz and sedimentary lithic fragments, suggesting that the deposits were mainly recycled from older sedimentary rocks.

Multiple statistical methods that we applied to the zircon U-Pb age distributions (probability density function plots, multidimensional scaling (MDS), DZStats, and DZMix modelling; see Methods) indicate a high similarity between all studied basins. Zircon age populations from the basins fall mainly into five groups: ca. 200–300 Ma, 390–480 Ma, 700–900 Ma, 1700–2000 Ma, and 2300–2600 Ma (Fig. 3). The overall age patterns are similar to those of the Triassic flysch in the Songpan-Ganzi and Yidun terranes, and to the pre-late Cretaceous strata in the Upper Yangtze terrane (Fig. 3 and Supplementary Fig. 4). DZstats results show that the zircon age components of all $K_2$–$E_1$ samples are statistically consistent, and share a statistically significant similarity to those of the Songpan-Ganzi, Yidun, and Upper Yangtze terranes (Supplementary Table 1). DZMix modelling also indicate principal contributions from the Songpan-Ganzi, Yidun, and/or the Upper Yangtze terranes into the studied basins (Supplementary Fig. 5). These similarities are also expressed in the MDS results. Samples from $K_2$–$E_1$ strata all cluster near the source samples of the three terranes mentioned above in the MDS plot (Supplementary Fig. 6). Moreover, modern sand samples from the Jinsha and Minjiang rivers, which drain these terranes, have zircon age distributions that also resemble those of the $K_2$–$E_1$ rocks (Supplementary Figs. 4 and 6).

The consistent provenance signal from the different basins argue for the existence of a continuous fluvial system during the late Cretaceous–early Palaeogene. We attribute minor variations

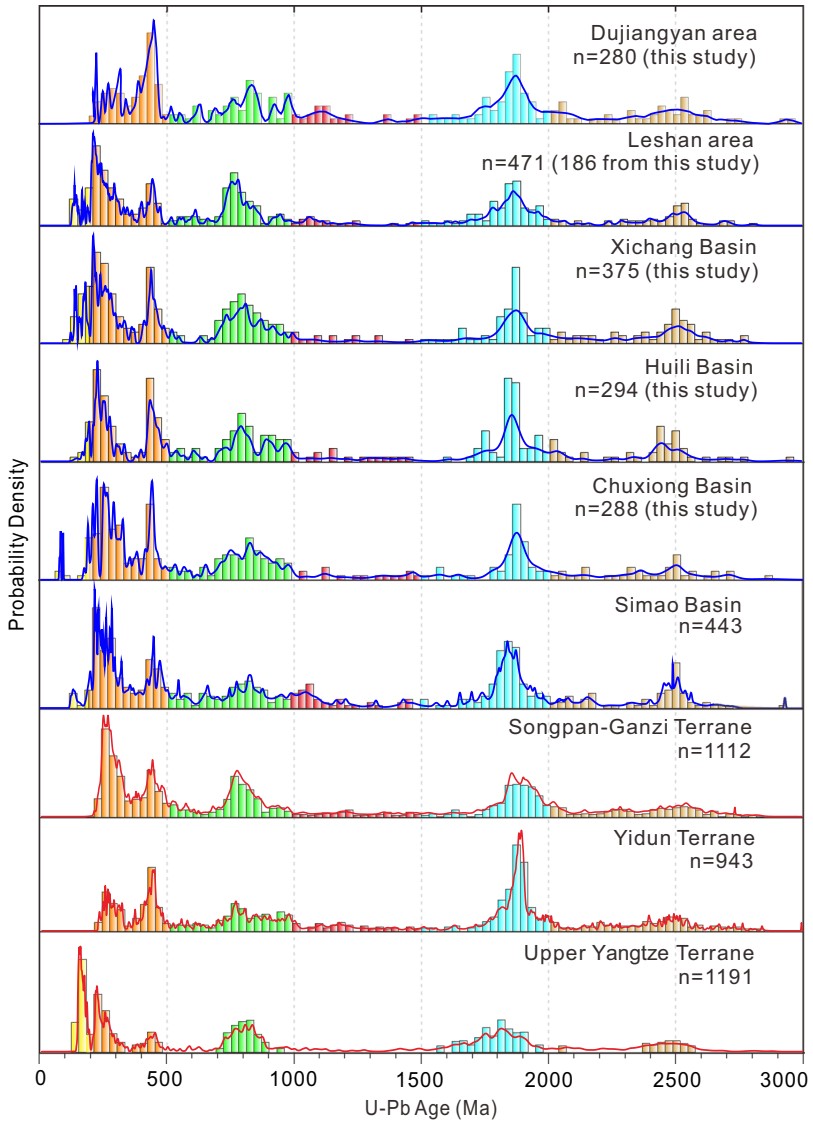

**Fig. 3 Zircon U-Pb data of late Cretaceous–early Palaeogene strata and potential source regions.** Zircon U-Pb data are from this study and were compiled from refs. [14,29,30,43]. Data of potential source areas are from refs. [14,30] and references therein. Blue and red lines represent normalised probability density functions of detrital zircon samples from late Cretaceous–early Palaeogene strata and potential source regions, respectively. Histograms are binned at 30 Ma intervals. *n* indicates number of concordant U-Pb ages. Source data information of this study is provided in Supplementary Data 1 and Supplementary Data 2.

of the age groups in the different areas to the spatial heterogeneity of sediment dispersal systems (Fig. 3 and Supplementary Figs. 4, 5 and 7). Age spectra from the Dujiangyan area are similar to that of Triassic flysch in the Songpan-Ganzi terrane; the absence of ages <200 Ma suggests a negligible input from the Upper Yangtze terrane. The young zircon population in the Leshan area and Xichang basin has a prominent peak at ca. 210–240 Ma, but contains distinct ages of <200 Ma as well, suggesting a mixture of material from the Songpan-Ganzi and Upper Yangtze terranes. To the south, zircon populations from the Huili Basin exhibit a decrease in the Upper Yangtze terrane signal (i.e., <200 Ma age peak) and an increased proportion of 1700–2000 Ma zircons compared to the Xichang Basin, reflecting an additional source from the Songpan-Ganzi or Yidun terranes. The provenance signal of the Chuxiong Basin changes little compared to the basins farther north, although a few late Cretaceous zircons suggest a minor source in the tectonically active Lhasa terrane, which was likely connected to the Neo-Tethyan Ocean by an external drainage system[31,41]. Alternatively, the Lhasa terrane

may have been drained by several independent river systems during the late Cretaceous–early Palaeogene.

## Discussion

**Existence of a continental-scale palaeo-drainage system**. We interpret our sediment-provenance data to support the existence of a late Cretaceous–early Palaeogene river system, which flowed southwards and was successively joined by tributaries from the Songpan-Ganzi, Yidun, and Upper Yangtze terranes, and from eastern Tibet (Fig. 4). Although local rivers from these source regions could display similar provenance signals, the abundance of several metres to tens of metres thick fluvial sand bodies, the scarcity of proximal-facies sediments, and the prolonged slow exhumation of the regions surrounding these basins, suggest that the $K_2$–$E_1$ sediments were not deposited in spatially separated endorheic basins. If these sediments were derived locally, wide-spread Neoproterozoic granitic rocks at the western margin of the South China Block[42] would provide a prominent source of detrital material for all studied basins, and a prominent peak at

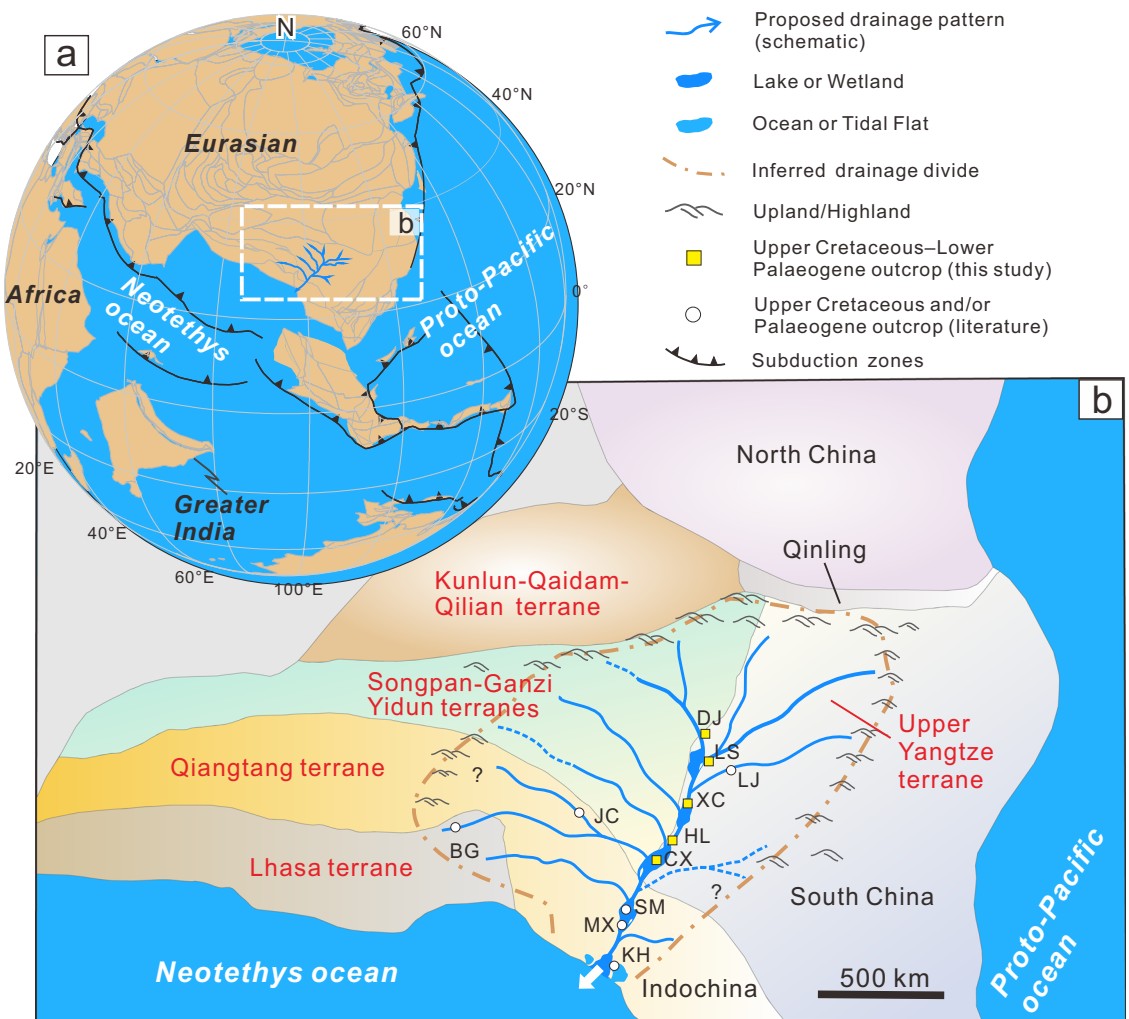

**Fig. 4 Schematic reconstruction of paleogeography before the India-Asia collision. a** Plate reconstruction of East Asia during the latest Cretaceous (generated from Müller et al.[68]). **b** Reconstruction of the late Cretaceous–early Palaeogene drainage system in eastern Tibet based on this study. KH Khorat Basin, MX Muang Xai Basin, LJ Liujia Section, JC Jianchuan Basin, BG Bangoin. Other abbreviations as in Fig. 1.

700–900 Ma would be expected. However, our samples are characterised by multimodal age spectra of detrital zircons (Fig. 3), thus supporting the existence of a single large-scale river system. The envisaged palaeo-river system followed the pre-existing foredeep depozone of the late Triassic–early Cretaceous foreland basin (Figs. 1a and 4). Other large rivers on Earth, such as the modern Yellow River, the Nile River, and the Amazon River also flow along ancestral structural lows[1,2,5], which can sustain the longevity of drainage systems[1,5].

When considered in a broader context, the $K_2$–$E_1$ deposits in the Simao Basin[14,43] and Muang Xai Basin[44] farther south and those from the Sichuan Basin to the Chuxiong Basin, which all exhibit similar zircon U-Pb age distributions (Fig. 3 and Supplementary Fig. 8), also require a highly-connected sediment transport pathway. This interpretation, along with the lack of $K_2$–$E_1$ terrestrial deposits in the western South China Sea[13,15], favours our palaeo-drainage reconstruction into the Neo-Tethyan Ocean (Fig. 4). A consistently south-directed paleocurrent from the sandstone bodies in the late Cretaceous units of the southernmost Khorat Basin provides additional physical evidence for this sediment transport pathway[45,46]. Furthermore, the widespread occurrence of late Cretaceous to early Palaeogene marine evaporites in the Simao and Khorat Basins[47,48] implies that these areas were close to sea level. Although some studies

inferred that paleoseawater likely originated from the northwest (e.g., the proto-Paratethys Sea; Wang et al.[49]), this recharge model cannot explain the observation that late Cretaceous marine deposits were absent between the Qiangtang and the Simao–Khorat Basins[47]. Rather, the pronounced southward thickening trends of the evaporite marker intervals[50], together with the terrestrial palaeogeographic reconstruction of East Asia spanning the late Cretaceous–early Palaeogene transition (e.g., Poblete et al.[51]) likely argue for a transgression of the Neo-Tethyan Ocean from the south or southwest (e.g., refs. [48,50]). Similar backwater effects extending hundred of kilometres inland can also be observed in the lower reaches of modern continental-scale river systems (e.g., Mississippi and Amazon rivers), where saltwater intrusions occur frequently[52]. Our interpretation of a connection between the Simao–Khorat Basins and the Neo-Tethyan does not preclude a more complex drainage configuration, potentially involving the assembly of multiple circuitous channels draining to the seaway. Additional sedimentary facies identification and systematic provenance analyses from $K_2$–$E_1$ strata in the Khorat basin and surrounding area, specifically for fluvial-deltaic and deep-water deposits, are necessary to fully explore this hypothesis. For example, in Borneo south of Khorat, a Mississippi-scale submarine fan system persisted throughout late Cretaceous–early Cenozoic time[53] (Supplementary Fig. 8),

leaving open the possibility that the sediment delivery system may have distally extended into deepwater areas. It is also possible that the paleo-drainage network discharged into the Neo-Tethyan Ocean during the late Cretaceous and Palaeocene, but later changed its course as a result of the India-Asia collision, and flowed into the proto-South China Sea starting in late Eocene.

**Implications for landscape evolution.** The existence of a large river system that persisted for tens of millions of years intrinsically reflects a prolonged period of tectonic stability and a relatively stable base level prior to the India-Asia collision and plateau uplift in eastern Tibet[28,35]. That these preconditions are met is indicated by the slow cooling rates from low-temperature thermochronology (Fig. 1b) and a slowly rising global sea level during $K_2$–$E_1$[54]. Due to its intercontinental setting, the region was prone to the generation of a low-relief landscape. Similar landscapes that formed in tectonically stable settings by slow erosion over tens to hundreds of millions of years are present in central Australia[55] and Mongolia[56].

A simple landscape evolution model confirms that the envisioned mechanism for generating the low-relief landscape is feasible (see Methods section). In our model, a $2000 \times 1200$ km drainage system with an initial relief of 3000 m (broadly constrained by structural data on crustal thickening in central Tibet[57]) is eroded by river incision and hillslope diffusion while being slowly uplifted at a rate of 40 m Myr$^{-1}$. After ~20 Ma, the total relief of the model is reduced to <1000 m and a low-relief landscape has formed (Fig. 5). Lateral channel migration and long-term continental weathering would further help to reduce topographic relief through time (cf., refs. [56,58]). In turn, decreased basin subsidence and reduction of drainage divides during the late Cretaceous–early Palaeogene led to overfill of these basins, also allowed the transcontinental river system to be entrenched.

Our conclusion that the low-relief landscape observed in eastern Tibet today formed at relatively low elevation is consistent with available palaeo-altimetric data, which indicate that most areas of eastern and southeastern Tibet were close to sea level before the middle Eocene[59,60]. Abundant low-relief landscape patches preserved north and south of the Ailaoshan–Red River shear zone[61] (Fig. 1a) indicate that this regional low-relief surface developed from the Tibetan hinterland to the Neo-Tethyan Ocean before the Ailaoshan–Red River shear zone was formed in the late Eocene (e.g., Gilley et al. [62]). Collectively, the arguments listed above show that a continental-scale river system existed during the late Cretaceous–early Palaeogene. This river system likely formed a low-relief landscape, which was uplifted and incised during Late Cenozoic deformation and uplift of the eastern Tibetan Plateau[10,63,64].

## Methods

**Petrography.** A total of twenty-two samples were collected from four (sub-) basins for thin section petrographic analysis (see Supplementary Table 3 for GPS data of samples). For each sample, at least 400 framework sand grains were identified and counted under the microscope. The results were plotted on ternary diagrams according to the Gazzi-Dickinson point-counting method[69,70]. Several kinds of provenance and component fields were established according to different mineral assemblages, including total quartz-feldspar-lithics (Q-F-L), volcanic and metamorphic lithic fragments-carbonate lithic fragments-terrestrial sedimentary clastic (Lv+Lm-Lc-Ls), and metamorphic lithic fragments-terrestrial sedimentary clastic-volcanic lithic fragments (Lm-Ls-Lv).

**Heavy mineral analysis.** Heavy minerals are defined as components with a density of more than ~2.9 g/cm$^3$ in detrital rock. We collected a total of 22 bulk samples from four (sub-) basins for heavy minerals analysis. Each sample consisted of 2–3 kg of fine to medium-grained sandstone. To obtain heavy mineral grains, bulk samples were crushed and clay and silt particles were removed. The remaining material of the 0.0625–0.5 mm grain-size fraction was washed using aluminium round wandering disc. The extracted

minerals were separated and identified by magnetic, electrostatic filters and heavy liquid of tribromomethane. To avoid a sampling bias, ~1000 heavy mineral grains were point-counted with regular spaced points.

**Detrital zircon geochronology.** In this study, we present 1423 new detrital zircon U-Pb ages from 14 late Cretaceous–early Palaeogene samples: five from the southwestern Sichuan basin (CX-02, CX-03, CX-07, CX-14, CX-19), three from the Xichang basin (CX-23, CX-24, CX-25), three from the Huili basin (CX-30, CX-31, CX-32) and three samples are from the Chuxiong basin (CX-34, CX-35, CX-36) (for detailed source data see Supplementary Data 1 and 2).

Zircon U-Pb dating was done at the State Key Laboratory of Earthquake Dynamics, Institute of Geology, China Earthquake Administration using laser ablation inductively coupled plasma mass spectrometry (LA-ICP-MS). At least 200 zircon grains were randomly selected to adhere to double-sided adhesive, and were poured into the laser sample target with epoxy resin. All samples were ablated by using a laser beam with a diameter of 28 μm, a frequency of 10 Hz and laser energy density of 4.0 J/cm$^2$. The zircon 91500 and GJ-1 standards were analysed every 10 grains, and were used as monitoring standards; element concentrations were determined using NIST610 as external standard and $^{29}$Si as internal standard (once every ten measuring points). For a detailed description of the methodology, equipment, and analytical procedures we refer to Yuan et al.[71]. The data were processed using the Glitter 4.0 software, whereas the common lead correction method followed Andersen[72]. To test the robustness and reliability of the results, the zircon U-Pb dating of samples CX-25-2, CX-31-2 and CX-36 were carried out at the Nanjing Normal University, zircon standard 91500 and Qinghu were used for external correction. The external standard zircons of 91500, GJ-1 and Qinghu yielded $^{206}$Pb/$^{238}$U weighted ages of $1062.5 \pm 1$ Ma ($n = 212$), $604.6 \pm 2$ Ma ($n = 137$) and $159.9 \pm 0.83$ Ma ($n = 34$), respectively. These are consistent with reference ages of $1063.1 \pm 8.1$ Ma (91,500, Yuan et al.[71]), $600.9 \pm 3.1$ Ma–$610 \pm 1.7$ Ma (GJ-1, refs. [73,74]) and $159 \pm 0.2$ Ma (Qinghu, ref. [75]). We apply a <10% discordance filter to the generated data. For zircon grains older than 1100 Ma, the best age was determined from $^{207}$Pb/$^{206}$Pb ratio[71]; while for those younger than 1100 Ma, the best age was calculated from $^{206}$Pb/$^{238}$U ratio[71]. In this study, all samples from the different basins show consistent detrital zircon components, and each sample yielded 68–123 concordant ages, which generally meets statistical requirements (Vermeesch[76]).

In order to objectively evaluate the similarities and differences of the zircon U-Pb age distributions between samples, we utilised multidimensional scaling analysis (MDS) and DZStats[77,78]. MDS is a robust and flexible principal components analysis (PCA) which makes fewer assumptions about the data, produces a "map" of points on which "similar" samples cluster closely together, whereas "dissimilar" samples plot farther apart. The calculations were carried out using the mdscale function of the Statistics Toolbox[77]. A MDS map can clearly visualise the 'dissimilarities' between the detrital zircon samples. The details about the MDS method are available in Vermeesch[77]. We also quantitatively distinguish the similarity between samples using DZstats, a quantitative MATLAB-based method. DZstats is based on intersample comparisons to compute Kolmogorov–Smirnov (K–S) test $D$ values and Kuiper test $V$ values (see Saylor et al.[78] for details). Two age distributions that yield low $D$ from the K-S test or low $V$ values from the Kuiper test imply that the two age spectra are nearly identical, whereas higher values have a larger probability of different parent populations.

Quantitative provenance analysis was also performed through the use of the DZMix modelling software based on Monte Carlo modelling[79]. This method allows an objective comparison between zircon U-Pb age distributions of individual mixed samples (late Cretaceous–early Palaeogene samples in this study) and those of known sources, extracting a best-fit model of the mixed samples and determining the relative proportions of each potential source for individual samples. Full details are provided by ref. [79]. In this study, we evaluate best-fit models using a $D$ value from the K-S test between observed and modelled zircon age components. We note that this is a relatively new technique with potential uncertainties induced by the number of zircon grains, sediment recycling and similar characteristics of the sources. Thus, the combination of zircon U-Pb age probability density functions, heavy mineral assemblages, petrographic frameworks and palaeocurrent data is essential for our provenance study.

**Landscape evolution simulation.** To assess the feasibility of the low-relief landscape formation envisioned for our late Cretaceous–early Palaeogene drainage reconstruction, we built a simple landscape evolution model. The numerical domain covers a leaf-shaped region of $2000 \times 1200$ km (analogous to the scale of our reconstructed palaeo-drainage system) with closed boundary conditions, except for one open boundary node. We start our numerical experiments by proportionally increasing the digital elevation of a quasi-steady-state topography, which we define as a change in elevation of <$10^{-5}$ m/a. This quasi-steady-state topography is generated by applying a random disturbance in elevation to a surface that is slightly tilted towards the main channel in the centre of the model. The initial total relief of the modelled domain is 3000 m. This value is estimated from structural data on crustal thickening in central Tibet before the India-Asia collision[57].

The landscape evolution model includes river incision and hillslope processes. We employed the stream power model for river incision and a linear diffusion law

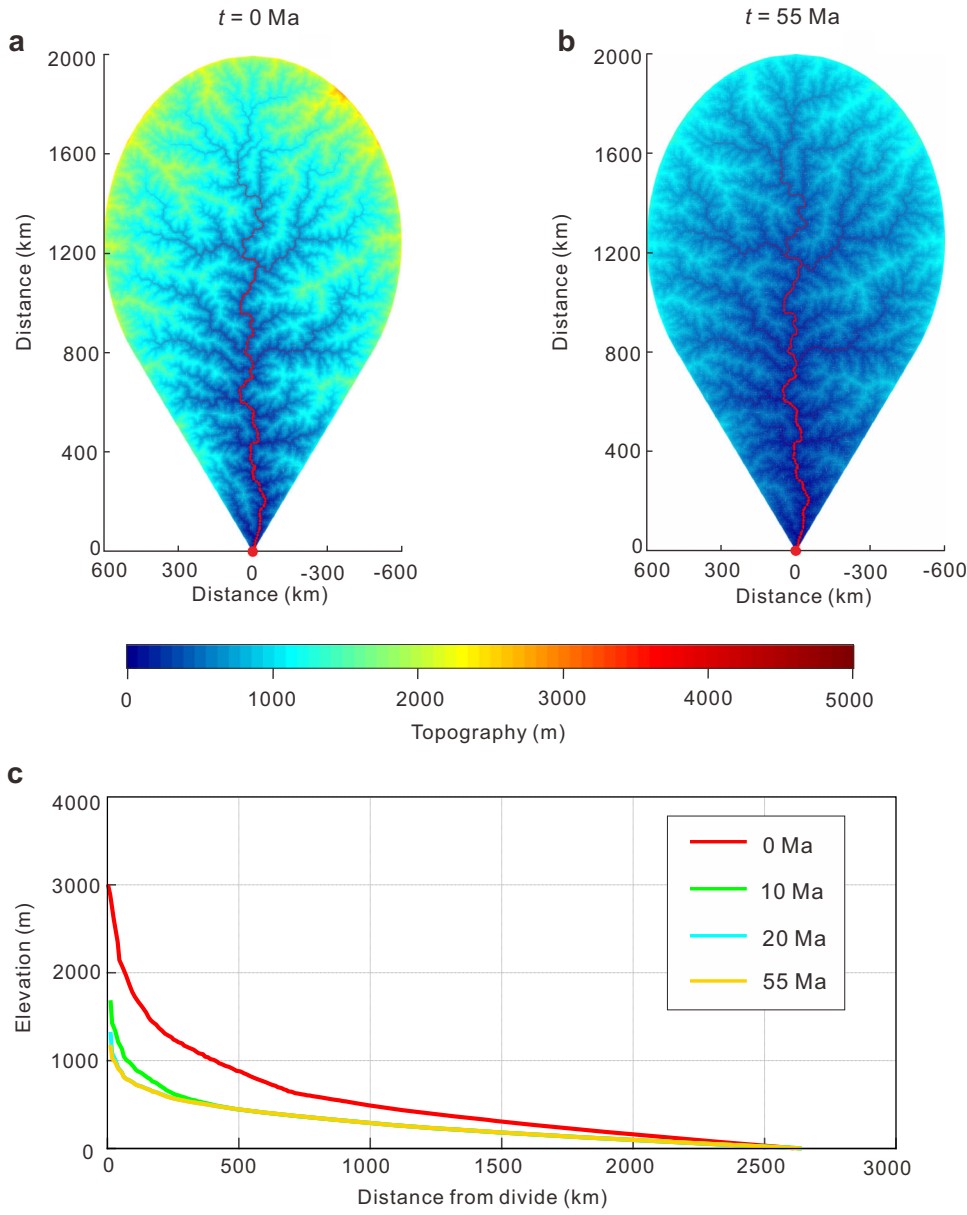

**Fig. 5 Results of landscape evolution models illustrating the decay of relief over ~55 Ma. a** Initial relief is 3000 m at 0 Ma. **b** Topographic relief is reduced to a regional low-relief topographic surface (< 1000 m) after 55 Ma. **c** Modelled change in elevation along the longitudinal profile of the trunk river from the continental interior to base level at 0 Ma, 10 Ma, 20 Ma and 55 Ma. Model parameters are given in Supplementary Table 2.

for describing hillslope erosion:

$$\frac{\partial z}{\partial t} = \nabla \cdot K_d \nabla z - K_s A^m |\nabla z|^n + U \qquad (1)$$

In the equation above, $z$ is elevation, $t$ is time, $K_d$ is the diffusive coefficient, $K_s$ is the incision coefficient, $A$ is the contributing drainage area, $U$ is the spatially uniform uplift rate and $m$ and $n$ are empirical constants. Note that the river incision coefficient ($K_s = 1.11 \times 10^{-6}$ m$^{0.1}$ per year) used in our simulation is the same that was obtained from the late Cenozoic Dadu River in eastern Tibet[80]. The values of all parameters used in our model are given in Supplementary Table 2.

## Data availability
All data used in this study are available in the Supplementary Information and Supplementary Data files. The detrital zircon geochronological data have also been deposited in the Figshare database https://doi.org/10.6084/m9.figshare.13506954.v4

## Code availability
The code used in landscape evolution simulation of this study is available from the corresponding author upon reasonable request.

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

## Acknowledgements
This work was supported by the Second Tibetan Plateau Scientific Expedition and Research (STEP) (2019QZKK0704), the NSF of China (41888101, 41622204), the Institute of Geology, CEA (IGCEA2003), and Innovation Group Project of Southern Marine Science and Engineering Guangdong Laboratory (Zhuhai) (311021002). We thank Peter Molnar and Andrew Laskowski for thoughtful comments and suggestions, and Maodu Yan and Bin Deng for discussions on the age constraints and facies of basin sediments.

## Author contributions
H.Z and X.Z designed the study. H.Z., X.Z., R.H., A.R.D., K.X.W., E.K., P.W. and P.Z. discussed early versions of the ideas presented here, which were refined following discussion among all authors. X.Z., J.X. and X.L conducted fieldwork. X.Z processed and analysed the data. X.Z., H.Z., Y.L., P.M. and B.Z created all figures. J.P., Y.W., P.W., J.Z. and K.L. assisted the zircon U-Pb dating. All authors contributed to the discussion and writing of the manuscript.

## Competing interests
The authors declare no competing interests.
