## [Peer Review File · Nature Communications]

Existence of a continental-scale river system in eastern Tibet during the late Cretaceous–early PalaeogeneReviewers' Comments:

Reviewer #1:

Remarks to the Author:

The evolution of large river is always a hot topic in earth science and attract huge interest. This study provide the traditional provenance results of the Upper Cretaceous to Paleocene in four basins, eastern Tibet, including petrographic modal counting and heavy mineral analysis for 22 samples and detrital zircon U-Pb ages for 16 samples. The results show the provenance similarities between the samples, the Songpan-Ganzi, Yidun, and Yangtze terranes, and the upper Jinsha, Min and Yalong rivers sand. Based on the provenance interpretation, the authors believed that there was a long-lived continental-scaled river system flowed into the Neo-Tethyan Ocean which generated a low-relief landscape in eastern Tibet. However, the provenance analysis and drainage reconstruction is questionable and thus conclusions in the manuscript are not reasonable. Moreover, there are several similar studies published before which eclipsed the significance of this study, not to mention several big weaknesses. Thus, I have to reject the manuscript for publication in Nature Communications.

Major Comments:

(1) To track the provenance across continents for a specific geological time period (e.g., late Cretaceous to Paleocene), comparisons have to be made amongst such equivalent-aged sedimentary strata. The authors should give the age constraints of the strata as accurate as possible. We all know that the Upper Cretaceous to Paleocene in these studied basins are based on mostly by the ostracods that identified about 40 years ago, which is not robust age constraint. Although the authors provide the youngest single detrital zircon ages (80, 90, 76 Ma for Sample CX-34, CX-25, CX-36, respectively) of the Chuxiong Basin, the single zircon age is insufficient to tell the maximum depositional age (MDA). About the MDA, pls refers to the following paper.

Sharman and Malkowski, 2020. Needles in a haystack: Detrital zircon U-Pb ages and the maximum depositional age of modern global sediment. *Earth-Science Reviews* 203, 103109.

<https://doi.org/10.1016/j.earscirev.2020.103109>

Moreover, the samples of the Simao Basin are from the Denghei Formation, which is believed to be deposited during Paleocene to late Eocene. The Upper Cretaceous to Lower Paleocene in the Simao Basin is the Mengyejing Formation which have robust radioactive and magnetostratigraphic constraints (112~>63 Ma, Yan M. et al., 2021, *Science China Earth Science*; Wang L. et al., 2015, *Cretaceous Research*). Please use provenance similarities to link continents and restore paleo-drainage systems only when the sedimentary strata are formed within a same time. Without correlation between the coeval strata, it is not convincible that the large-scale river can be flow into Simao and further south.

(2) The depositional environment is essential to reconstruct the paleo-river system. In Supplementary Information Xichang Basin section, for the Xiaoba and Leidashu Formations, Deng et al. (2018, ref. 16) never proposed a meandering river but a fluvial and shallow-lacustrine environment. Also, ref. 25 indicate a lacustrine environment. Together with the occurrence of evaporites, I think the lacustrine environment is more reasonable. In Chuxiong Basin section, I strongly doubt the interpretation of the exorheic lake because the thick evaporites are occurred in the Jiangdihe Formation. Similarly, the Upper Cretaceous Mengyejing Formation developed several hundred meters of evaporites, indicative of an endorheic basin. How to understand the drainage pattern when the large river flow into the Chuxiong and Simao Lake?

(3) Provenance interpretation

a, About the potential sources. I doubt if you correctly plot the age distribution of the Yangtze terrane. Because all 1191 zircon ages in figure 3 and extended data figures 4&6 don't show a prominent age peak of 750-1000 Ma which is believed to be a diagnostic feature of the Yangtze. I tried to find the paper you cited (refs. 14& 30) and found that the zircon ages in these papers are from the Early Paleozoic. So why the age peaks at ca. 150 Ma and 220 Ma occur? I don't know the specific samples of the 1191 ages although you mentioned they were from the pre-late Cretaceous strata. Thus, the relationship between the ages<200 Ma and the Yangtze terrane need to be further proved.

b, line 171-172, We cannot interpret the provenance just based on the age peaks. Why the Lhasa Terrane just simply provide the late Cretaceous zircons to the Chuxiong Basin, not conclude the other age populations? If you plot the Lhasa in the MDS diagram, you will find the Lhasa plot apart from

your samples.

c, In the MDS plot (Figure 5b), clearly, the Min River and Yalong River plot apart from the samples. The age peaks of Yalong River sand are different from the samples and Songpan-Ganzi. Only Upper Jinsha river plot closer to the samples. So how do you explain your drainage reconstruction?

(4) Drainage system

As mentioned above, for the reasons of the depositional environment and provenance interpretation, I disagree with the idea that a large-scale river system that connect the all the so-called exorheic basins although I agree somehow a south-flowing river exists. Specifically, the Chuxiong and Simao are two endorheic basins with several hundred meters of evaporites. Perhaps, the river just stop when it flowed into these two basins. Even if a trans-continental river system flowed to the Neo-Tethyan Ocean, about the river course more evidence should be provided, e.g., provenance correlation to the west Burma? And to Khorat Basin? Cai F.L. et al. (2020, GSAB) indicated that the Upper Cretaceous-Eocene strata are mainly sourced from the western Myanmar Arc with detrital zircon age peaks of 100-60 Ma. This is totally different with the coeval strata of the eastern Tibetan basins in this study. Thus, in the reconstruction map Figure. 4, I strongly doubt the river flowed Simao via Myanmar to the Neo-Tethyan Ocean. Actually, the paleocurrent of the Simao Basin would suggest a connection with the Khorat basin (See Yan M.D., et al., 2021, and references therein). More importantly, large river system in the Late Cretaceous flowed to the Neo-Tethyan Ocean had been proposed by Yan Maodu et al., 2021, Science China Earth Science. So it is not the authors who claimed that they proposed it for the first time. I found that there are several papers proposed a large river system prior to the collision between India and Asia, e.g., Deng et al. (2018, ref. 16); Wang L.C., et al. 2020, Palaeo-3; Yan M., et al., 2021.

(5) Data and methodology

1), Petrography part. The authors should tell the readers what kind of method you use when you do modal analysis. Usually, sedimentary geologists use the Gazzi-Dickinson method (Ingersoll et al., 1984). And all ternary diagrams should cite the original references.

In Extended data Figure 2, you use Lc to represent the carbonatite lithic, however, the Ls was used in the Table S2. Moreover, I strongly doubt the high percentage of carbonatite lithic in the samples, if yes, why don't you count these into the Lv?

2), For all 22 samples for petrographic and heavy mineral analysis, I suggest the authors give a table that contain the GPS and stratigraphic information (I can't tell which formation and basin of some samples belong to, e.g., CX-29, CX-01, CX08). In Extended data Figure 1, the authors should locate the heavy mineral samples in the stratigraphic columns of the Xichang, Huili, and Chuxiong basins.

3) Detrital zircon geochronology part. More information should be provided to let the readers examine the reliability of your data. What are the dating results of your age external standards? And how do these results compared to the suggested age values? So you should provide the dating results the standard zircons. In addition, relevant citations should be given regarding external standards. The representative CL images of detrital zircons especially the ones with young ages (<200 Ma in this area) is very important. Because in my opinion, these young zircons are mostly from local source.

4) The method of MDS is not sufficiently explained. (a) How the MDS map was generated? (b) Which metric do you choose when you plot the MDS, e.g., likeness, similarity.....?

5) Line 461-464, the K-S test p-value is not recommended to use. Use of the K-S or Kuiper test p-values for quantitative similarity analysis of detrital geochronological data sets is likely to lead to incorrect conclusions (Satkoski et al., 2013, GSA Bulletin; Vermeesch, 2013; Saylor and Sundell, 2016, Geosphere). As noted by Vermeesch (2013), the D or V values provide more robust assessment of the dissimilarity between samples than do p-values. Thus, the D or V values of K-S or Kuiper tests are suggested to use.

Minor Comments:

1, Line 103, Yangtze is believed to be a part of South China block, why did you show a different concept? The same as in Figure 1a.

2, Line 125-128, I can't understand why the late Cretaceous-early Paleogene strata could represent the youngest terrestrial clastic deposits? At least the Lower Cretaceous in these basins are terrestrial clastic rocks.

3, Line 131, the same as the previous comment, actually the Upper Cretaceous evaporites developed

in the SW Sichuan, Xichang, Chuxiong, and Simao Basins (Liu Shugen et al., 2019, Journal of Chengdu University of Technology, v.46, No. 1, 1-28; and references therein). Especially, thick evaporites were developed in the Chuxiong and Simao Basin (Liu Chenglin et al., 2018, Ore Geology Reviews).

4, Line 135-137, about the depositional environment interpretation, pls see my comments above.

5, Line 204-205, during K2-E1, the global sea level was gradually rose from ca.80 Ma to 60-50 Ma, and fell since 50 Ma (Miller et al., 2005, Science, 10.1126/science.1116412). Thus, it is not stable. How to understand?

6, Actually, Deng B. et al.(2018, GSAB) published many detrital zircon ages of the Upper Cretaceous-Paleocene in Sichuan, Chuxiong basins. I suggest the authors should compile all published detrital zircon ages together with this study to do provenance analysis.

7, Figure. 1, where is the Sichuan Basin?

8, Supplementary Information line 41-42, very thick evaporites were developed in the Upper Cretaceous Chuxiong and Simao Basin. Thus, it is not an indication of river discharge but an endorheic lake.

9, Extended figure 4a and 4b, all the references are not incorrect. I can't find any mentioned samples in these cited papers (ref 31, 47, and 25). For the ref. 25, maybe you want to refer to ref. 14 (Chen Y., et al., 2017, EPSL). However, the 272 zircon ages from ref. 14 belong to the Denghe Formation, which was assigned a Paleocene-late Eocene in age based on the fossils. Thus, I doubt whether it is reasonable to correlate to the Simao basin in this manuscript.

The part of Landscape evolution simulation is beyond my expertise, so I can't give any comments about it.

Reviewer #2:

Remarks to the Author:

Dear Authors and Editor(s),

Thank you for the opportunity to review "Existence of a continental-scale river system in eastern Tibet during the late Cretaceous-early Palaeogene" by Zhao and co-authors. This study reports provenance data (detrital zircon and sandstone petrography) from Cretaceous-Eocene sedimentary basins that are located along the southeast margin of the Tibetan Plateau. The authors observe that the provenance data is very consistent between basins and back up these observations with statistical methods that are mainly presented in the supplementary figures. To explain these data, the authors argue that there was likely a continent-scale river that connected the basins and drained to the Neo-Tethyan Ocean, which separated India from Eurasia prior to India-Asia collision. The authors note that this interpretation also has relevance to the debate over the low-relief, incised landscapes that are located along the southeast margin of the Tibetan Plateau. This aspect of the research will likely be the most broadly relevant and controversial aspect. Several dynamic models for Tibetan Plateau growth invoke different mechanisms for formation of the low-relief landscapes. This, coupled with the inherent interestingness of ancient, continental-scale rivers, make this research exciting and broadly relevant. As I am an expert on detrital zircon geochronology and the tectonics of the Tibetan Plateau, I chose to focus on these aspects for my review. In my opinion, the provenance data are robust and well supported by statistical methods. I would encourage the authors to include some more discussion of their comparison between source terrane signatures and basin signatures. A key point that must be proven is that the source areas are distinguishable based on their detrital zircon age spectra whereas all the sampled basins are not, indicating mixing by a large river system. The supplementary figures were very helpful to convince me of the validity of the argument, yet they are sparsely discussed or referenced in the main text.

The authors also present a landscape evolution model that they link with their thermal modeling results. They claim that it illustrates the plausibility of the continent-scale river along the southeastern Tibetan Plateau because faster exhumation rates would be expected in the upstream reaches of the drainages. This seems plausible to me, but the authors should also emphasize that the Lhasa Terrane hosted an active, Cordilleran orogenic system at the time of deposition of the samples. It has

previously been likened to the modern Andes. It should not be implied that fluvial drainage networks were solely responsible for the cooling of samples further toward the interior of the Tibetan Plateau, as the tectonic activity in this region undoubtedly affected these results.

Respectfully,

Dr. Andrew Laskowski

Assistant Professor

Department of Earth Sciences

Montana State University

Reviewer #3:

Remarks to the Author:

Comments to 'Existence of a continental-scale river system in eastern Tibet during the late Cretaceous–early Paleogene' By Zhao et al.

Based on new petro-stratigraphy, heavy-mineral analysis, and detrital zircon U-Pb studies on late Cretaceous–early Paleogene sediments from the east margin of Tibet, Zhao et al. proposed a novel continental-scale river flowing southwestward to the Neo-Tethyan Ocean along the eastern Tibet in the late Cretaceous–early Paleogene, and used this river to explain the formation of low-relief in the east margin of Tibet. The topic of this study is of broad interest for geologists, and the proposed model is significantly different with previous models, which seems valuable for publication in *Nature Communications*. However, there are some important unclears in the MS, which should be addressed before acceptance.

1. The authors claimed that the sediments they studied are Late Cretaceous–Early Paleogene, but did not provide solid evidences to support, although I noticed in the Supplementary Information they have shown some fossils evidence, but these are not enough. Some recent studies have shown that the age of sediments in this area is significantly older than the traditional fossils suggested, e.g., Gourbet et al. (2017). Therefore, if the ages of these sediments are wrong, the river story could be changed.

2. The authors argued that the K2–E1 sediments were not deposited in spatially separated endorheic basins based on the lack of coarse-grained sediments and thick evaporites. However, these are not strong evidence to preclude this possibility. It is very likely that these sediments were derived locally by recycling from surrounding older rocks. For example, the Nangqian and Gonjo basins in eastern Tibet show similar lithologies as the K2–E1 sediments by the authors, but studies have shown that the two basins were sourced locally (Horton et al., 2002). The authors also suggested that the low-relief could not be severed as physiographic barriers for endorheic basins. However, endorheic basin can be formed in any landscape with local structures, e.g., normal fault.

3. The evidence of lack K2–E1 terrestrial deposits in the South China Sea to against a paleo-river flowed to the Proto-Pacific Ocean is not the truth. For example, as shown in the Fig. 3 of Clift et al., 2006 *EPSL*, the Paleogene sediments are very thick in the South China Sea. So the authors have to find strong evidences to support why a continental-scale river was flowed to the Neo-Tethyan Ocean rather than the Proto-Pacific Ocean.

4. The biggest problem of this MS is the Fig. 4, which is inconsistent with the tectonic background. The authors refer the reference of Muller et al. (2016) for the late Cretaceous paleogeography reconstruction, but the map shown in Fig. 4a is inconsistent with geological evidence: The Indochina and Sibumasu, which locate southwest of the South China Block, have amalgamated to South China in Late Triassic, and the Lhasa has collided with Qiangtang in the early Cretaceous, so it is impossible that a Neo-Tethyan Ocean was still existed between South China and Indochina as shown in Fig. 4a. Therefore, if a southwest flow river to the Neo-Tethyan Ocean existed in the Late Cretaceous, more evidence from the Indochina and Sibumasu terranes must be shown, currently the westernmost basin

shown in the MS is the Simao Basin, which cannot preclude the possibility that the river flowed to the South China Sea as previous model suggested.

**Response letter**

In this letter we will provide our detailed response (in blue text) to the comments of the editor
and the three reviewers (in black) and explain all changes performed on the manuscript.

**Response to the point raised by the Associate Editor**

*Regarding the "additional evidence that supports the proposed route (of the palaeo-river) to*
*the Neo-Tethyan Ocean", we would like to mention four lines of additional evidence:*

**Reply:**

(1) In the Simao–Khorat Basins (located south of the Chuxiong Basin in Indochina; Fig. 1),
published data sets on sedimentology proxies, biomarkers, element geochemistry, and
isotopic geochemistry consistently indicate that late Cretaceous to early Paleogene
evaporites are mainly of marine origin, which has generally been interpreted as the result
of a transgression of the Tethys ocean (e.g., Hite and Japakasetr, 1979; El Tabakh et al.,
1999; Zhang et al., 2013; Liu et al., 2018; Qin et al., 2020; Wang et al., 2021). This
suggests that the Simao–Khorat Basins was very close to the ocean at that time. Moreover,
these basins pertain to Tethyan Tectonic Domain during the late Cretaceous–early
Paleogene (Yan et al., 2021; Liu et al., 2018). Thus, a more reasonable interpretation is
that the proposed continental-scale river system discharged into the Neo-Tethyan Ocean.
We have supplemented this evidence on **lines 219–226** of the revised manuscript.

(2) A new compilation of detrital zircon ages from the studied basins and three other basins
farther south (Simao, Muang Xai, and Khorat) are very similar and provide additional
support for a through-going sediment transport system (see new Fig. S8 in Supplementary
Information) (Carter et al., 1999; Wang et al., 2014; Wang et al., 2017; Chen et al., 2017;
Wang et al., 2020; this study). Specifically, late Cretaceous samples from the Muang Xai
and Khorat basins show strikingly consistent Precambrian peaks at 2400–2600 Ma, 1900–
1600 Ma, and 900–600 Ma (Fig. S8), strongly suggesting the Songpan-Ganzi and Upper
Yangtze terranes as main source areas (revised manuscript, **lines 212–216**).

(3) In the current depositional area of the Red River (the Yinggehai-Song Hong Basin of the
South China Sea), many boreholes have revealed that the Cenozoic deposits at the bottom
of borehole are not older than late Eocene (~37 Ma) (see Figure 4 in papers of Clift et al.,
2006 (GRL) and Clift et al., 2008; Lei et al., 2011; Wang et al., 2019). Regionally, the
Proto-South China Sea during the late Cretaceous to early Cenozoic period is
characterized by a series of deep, rapidly-subsiding small-scale rift basins under back-arc
extension (see review of Morley et al., 2012). For example, in the Pearl River Basin in the
eastern South China Sea, the upper Cretaceous strata are characterized by a dominance of
zircons with ages clustering around the late Jurassic–Cretaceous, which has been
interpreted to be from nearby continental arcs (see Figure 6 in Shao et al., 2017, and
Figure 10 in He et al., 2020). These observations clearly imply that there was no
large-scale drainage system that linked eastern Tibet with the proto-South China Sea prior
to late Eocene time (revised manuscript, **lines 216–219**).

(4) Abundant low-relief landscape patches are preserved on both sides of the present-day
Ailaoshan-Red River shear zone (e.g., Schoenbohm et al., 2004; Clark et al., 2005; Fig.1;
see Figure 1 of Wang et al., 2017) suggesting that there was a regional low-relief surface
from the Tibetan hinterland to the sea, which partly supports our interpretation that a
paleo-river flowed southwards into the Neo-Tethyan Ocean before the fault system

became active ~35 Ma ago (e.g., Scharer et al., 1994; Gilley et al., 2003). In other words,
if there was a paleo-Red river connecting the Tibetan hinterland with the Proto-South
China Sea before Miocene surface uplift in eastern Tibet (as proposed by Clift et al., 2006,
GRL), its course must have been established after the formation of the regional low-relief
surface. We have included this evidence on **lines 250–254** of the revised manuscript.

(5) Recently, Cai et al. (2020) and Zhang et al. (2021) have clearly shown that the Upper
Cretaceous–Early Eocene deposits in the Sibumasu–Burmese region are of proximal
origin (see Figure 10 in Cai et al., 2020, and Figure 13b in Zhang et al., 2021) based on
detrital zircon data, thus excluding the possibility that there was a west-flowing
continental-scale river system to the Sibumasu–Burmese region during the late
Cretaceous–early Palaeogene.

**References:**

- Cai, F., Ding, L., Zhang, Q., Orme, D. A., & Sein, K. Initiation and evolution of forearc basins
in the central myanmar depression. *Geol Soc Am Bull.* **132**, 5–6 (2020).
- Carter, A. & Moss, S. J. Combined detrital-zircon fission-track and U-Pb dating: A new approach to
understanding hinterland evolution. *Geology* **27**, 235–238 (1999).
- Chen, Y., Yan, M., Fang, X., Song, C., Zhang, W., Zan, J. et al. Detrital zircon U-Pb geochronological
and sedimentological study of the Simao Basin, Yunnan: Implications for the early Cenozoic
evolution of the Red River. *Earth Planet. Sci. Lett.* **476**, 22–33 (2017).
- Clark, M. K., House, M. A., Royden, L. H., Whipple, K. X., Burchfiel, B. C., Zhang, X. & Tang, W.
Late Cenozoic uplift of south-eastern Tibet. *Geology* **33**, 525–528 (2005).
- Clift, P. D., Blusztajn, J. & Nguyen, A. D. Large-scale drainage capture and surface uplift in eastern
Tibet-SW China before 24 Ma inferred from sediments of the Hanoi Basin, Vietnam. *Geophys. Res.*
*Lett.* **33**, L19403 (2006)
- Clift, P. D., Long, H. V., Hinton, R., Ellam, R. M., Hannigan, R., Tan, M. T. et al. Evolving east Asian
river systems reconstructed by trace element and Pb and Nd isotope variations in modern and
ancient Red Rive-Song Hong sediments. *Geochem., Geophys., Geosyst.* **9**, Q04039 (2008).
- El Tabakh, M., Utha-Aroon, C. & Schreiber, B.C. Sedimentology of the Cretaceous Maha Sarakham
evaporites in the Khorat Plateau of northeastern Thailand. *Sediment. Geol.* **123**, 31–62 (1999).
- Gilley, L. D., T. M. Harrison, P. H. Leloup, F. J. Ryerson, O. M. Lovera, and J.-H. Wang, Direct dating
of left-lateral deformation along the Red River shear zone, China and Vietnam. *J. Geophys. Res.*
**108 (B2)**, 2127 (2003).
- He, J., Garzanti, E., Cao, L., & Wang, H. The zircon story of the Pearl river (China) from Cretaceous
to present. *Earth Sci Rev.* **201**, 103078 (2020).
- Hite, R.J. & Japakasetr, T. Potash deposits of Khorat Plateau, Thailand and Laos. *Econ. Geol.* **74**, 448–
458 (1979).
- Lei, C., Ren, J., Clift, P. D., Wang, Z., Li, X. & Tong, C. The structure and formation of diapirs in the
Yinggehai–SongHong basin, South China Sea. *Mar. Pet. Geol.* **28**, 980–991 (2011).
- Liu, C.L., Wang, L.C., Yan, M.D., Zhao, Y.J., Cao, Y.T., Fang, X.M., Shen, L.J., Wu, C.H., Lv, F.L. &
Ding, T. The Mesozoic-Cenozoic tectonic settings, paleogeography and evaoritc sedimentation of
Tethyan blocks within China: implications for potash formation. *Ore Geol. Rev.* **102**, 406–425
(2018).
- Morley, C.K. Late Cretaceous–Early Palaeogene tectonic development of SE Asia. *Earth Sci Rev.*
**115**, 27–75 (2012).
- Qin, Z., Li, Q., Zhang, X. G., Fan, Q. S., Wang, J. P., Du. Y. S., Ma, Y. Q., Wei, H. C., Yuan, Q. &
Shan, F. S. Origin and recharge model of the late cretaceous evaporites in the khorat plateau. *Ore*
*Geol Rev.* **116**, 103226 (2020).
- Scharer, U., Z. Lian-Sheng, and P. Tapponnier, Duration of strike-slip movements in large shear zones:

- The Red River belt, China. *Earth Planet. Sci. Lett.* **126**, 379–397 (1994).
- Schoenbohm, L., Whipple, K. & Burchfiel, B.C. River incision into a relict landscape along the Ailao
Shan shear zone and Red River fault in Yunnan Province, China. *Geol. Soc. Am. Bull.* **116**, 895–909
(2004).
- Shao, L., Cao, L., Qiao, P., Zhang, X., Li, Q., van Hinsbergen, D.J. Cretaceous–Eocene provenance
connections between the Palawan Continental Terrane and the northern South China Sea margin.
*Earth Planet. Sci. Lett.* **477**, 97–107 (2017).
- Wang, C., Liang, X., Foster, D. A., Liang, X., Zhang, L. & Su, M. Provenance and drainage evolution
of the Red River revealed by Pb isotopic analysis of detrital K-feldspar. *Geophys. Res. Lett.* **46**,
6415–6424 (2019).
- Wang, L. C., Liu, C., Gao, X. & Zhang H. Provenance and paleogeography of the Late Cretaceous
Mengyejing Formation, Simao Basin, southeastern Tibetan Plateau: Whole-rock geochemistry,
U-Pb geochronology, and Hf isotopic constraints. *Sedimentary Geol.* **304**, 44–58 (2014).
- Wang, L. C., Zhong, Y. S., Xi, D. P., Hu, J. F. & Ding, L. The Middle to Late Cretaceous marine
incursion of the Proto-Paratethys sea and Asian aridification: A case study from the Simao-Khorat
salt giant, southeast Asia. *Palaeogeogr Palaeoclimatol Palaeoecol.* **567**, 110300 (2021).
- Wang, Y. L., Wang, L. C., Wei, Y. S., Shen, L. J., Chen, K., Yu, X. C. & Liu, C. L. Provenance and
paleogeography of the Mesozoic strata in the Muang Xai basin, northern Laos: petrology,
whole-rock geochemistry, and U–Pb geochronology constraints. *Int J Earth Sci.* **106**, 1409–1427
(2017).
- Wang, Y., Schoenbohm, L. M., Zhang, B., Granger, D., Zhou, R.J., Zhang, J. J. & Hou, J. J. Late
Cenozoic landscape evolution along the Ailaoshan shear zone, SE Tibetan Plateau: Evidence
from fluvial longitudinal profiles and cosmogenic erosion rates. *Earth Planet. Sci. Lett.* **472**, 323–
333 (2017).
- Yan, M. D., Zhang, D. W., Fang, X. M., Zhang, W. L., Song, C. H et al. New Insights on the age of the
Mengyejing formation in the Simao basin, SE Tethyan domain and its geological implications.
*Sci China Earth Sci.* **64**, 231–252 (2021).
- Zhang, P., L. Mei, S-Y., Jiang, S., Xu, R. A., Donelick, R. et al. Erosion and sedimentation in
SE Tibet and Myanmar during the evolution of the Burmese continental margin from the
Late Cretaceous to Early Neogene. *Gondwana Res.* In press (2021) doi:
<https://doi.org/10.1016/j.gr.2021.04.005>.
- Zhang, X.Y., Ma, H.Z., Ma, Y.Q., Tang, Q.L. & Yuan, X.L. Origin of the late Cretaceous
potash-bearing evaporites in the Vientiane Basin of Laos: $\delta^{11}\text{B}$ evidence from borates. *J. Asian*
*Earth Sci.* **62**, 812–818 (2013).

**Response to common comments from Reviewer #1 and Reviewer #3**

**(1) Reviewer #1:** To track the provenance across continents for a specific geological time
period (e.g., late Cretaceous to Paleocene), comparisons have to be made amongst such
equivalent-aged sedimentary strata. The authors should give the age constraints of the strata
as accurate as possible. We all know that the Upper Cretaceous to Paleocene in these studied
basins are based on mostly by the ostracods that identified about 40 years ago, which is not
robust age constraint. Although the authors provide the youngest single detrital zircon ages
(80, 90, 76 Ma for Sample CX-34, CX-25, CX-36, respectively) of the Chuxiong Basin, the
single zircon age is insufficient to tell the maximum depositional age (MDA).

**Reviewer #3:** The authors claimed that the sediments they studied are Late
Cretaceous-Early Paleogene, but did not provide solid evidences to support, although I
noticed in the Supplementary Information they have shown some fossils evidence, but these
are not enough. Some recent studies have shown that the age of sediments in this area is
significant older than the traditional fossils suggested, e.g., Gourbet et al. (2017). Therefore, if

the ages of these sediments are wrong, the river story could be changed.

**Reply:** As the reviewer#1 points out correctly, the maximum depositional age (MDA) is
insufficient to constrain depositional ages with certainty. However, although the youngest
single-grain ages from our samples are not a robust indicator of the true depositional age, they
are consistent with the late Cretaceous–early Cenozoic biostratigraphic age of these deposits.
More importantly, younger (i.e. late Eocene) zircons are completely lacking in our samples.
Given that late Eocene plutons are common across southeastern Tibet (e.g., Lu et al., 2012;
Deng et al., 2014); the absence of late Eocene zircon age implies that the studied continental
red-beds are older than late Eocene.

We argue that the fossil assemblages provide reasonable information on the depositional
age of our studied sedimentary sections and similar deposits that occur throughout eastern
Tibet. The characteristic ostracods, charophyta, and few lamellibranchia are very common in
late Cretaceous–early Paleocene strata from other areas of China as shown by recent reviews
of Xi et al. (2019) and Wang et al. (2019), which provide further support for our age scheme.
We wish to add that a recent magnetostratigraphy study shows that the Guankou and
Mingshan Formations of the Shiyang section in the southwestern Sichuan Basin ranges in age
from ~84 to ~43 Ma (Shen et al., 2018, Master thesis). Also, the Mengyejing Formation
(which roughly correlates with the Guankou, Xiaoba, and Jiangdihe formation of this study)
of the Jiangcheng section in the Simao Basin has been dated at ~112–63 Ma (Yan et al., 2021),
largely in agreement with the palaeontological results. We added this information to the
**Supplementary information.**

The Fig. R1 below summarizes the existing age data, which support our interpretation of
a late Cretaceous to early Palaeocene age of the studied sediments.

**References**

- Deng, B., Chew, D., Jiang, L., Mark, C., Cogne, N., Wang, Z. J. & Liu, S. G. Heavy mineral analysis
and detrital U-Pb ages of the intracontinental Palaeo-Yangtze basin: Implications for a
transcontinental source-to-sink system during Late Cretaceous time. *Geol. Soc. Am. Bull.* **130**,
2087–2109 (2018).
- Deng, J., Wang, Q. F., Li, G. J. & Santosh, M. Cenozoic tectono-magmatic and metallogenic processes
in the Sanjiang region, southwestern China. *Earth-Science Reviews* **138**, 268–299 (2014).
- Feng, M.G., Wu, J., Han, R.S., Zhu, Y.Z. & Yan, C.M. The salt-bearing strata in Vientiane plain, Laos.
*Yunnan Geol.* **24**, 407–413 (2005).
- Gu, X. D. & Li, X. H. Lithostratigraphy in Sichuan Province 207–217 (China University of
Geosciences Press, 1997).
- Huang, R. J. Cretaceous to early Tertiary Charophytes in Sichuan Province. *Acta*
*Micropalaeontologica Sinica* **2**, 77–89 (1985).
- Lu, Y.J., Kerrich, R., Cawood, P.A., McCuaig, T.C., Hart, C.J.R., Li, Z.X., Hou, Z.Q., Bagas, L. Zircon
SHRIMP U–Pb geochronology of potassic felsic intrusions in western Yunnan, SW China:
constraints on the relationship of magmatism to the Jinsha su-ture. *Gondwana Res.* **22**, 737–747
(2012).
- Royden, L. H., Burchfiel, B. C., Robert, D. H. The Geological Evolution of the Tibetan Plateau. *Science*
**321**, 1155371 (2008).
- SBGMR (Sichuan Bureau of Geology and Mineral Resources). Geological Map of the Emei Sheet
(scale 1:50000) (Geological Publishing House Press, 1989).
- SBGMR (Sichuan Bureau of Geology and Mineral Resources). Regional Geology of Sichuan Province
192 h.11&12 (part I): 264–282 (Geological Publishing House, 1991).
- Shen, Q. Cretaceous–Cenozoic magnetostratigraphy study in southwestern Sichuan Basin and its

geological significance. Master thesis in the Northwest University (2018).
 Tang, Z. Biostratigraphic units in Mishi Basin in Panxi area. *Acta Geologica Sichuan*. **16**, 209–212
 (1996).
 Wang, Y. L., Wang, L. C., Wei, Y. S., Shen, L. J., Chen, K., Yu, X. C. & Liu, C. L. Provenance and
 paleogeography of the Mesozoic strata in the Muang Xai basin, northern Laos: petrology,
 whole-rock geochemistry, and U–Pb geochronology constraints. *Int J Earth Sci*. **106**, 1409–1427
 (2017).
 Wang, Y. Q., Li, Q., Bai, B., Jin, X., Mao, F. Y., Meng, J. Paleogene integrative stratigraphy and
 timescale of China. *Sci China Earth Sci*. **62**, 287–309 (2019).
 Xi, D. P., Wan, X. Q., Li, G. B., Li, G. Cretaceous integrative stratigraphy and timescale of China. *Sci*
 *China Earth Sci*. **62**, 256–286 (2019).
 Yan, M. D., Zhang, D. W., Fang, X. M., Zhang, W. L., Song, C. H et al. New Insights on the age of the
 Mengyejing formation in the Simao basin, SE Tethyan domain and its geological implications. *Sci*
 *China Earth Sci*. **64**, 231–252 (2021).
 YBGMR (Yunnan Bureau of Geology and Mineral Resources). Geological Map of the Chuxiong Sheet
 (scale 1:200000) (Geological Publishing House Press, 1966).
 YBGMR (Yunnan Bureau of Geology and Mineral Resources). Regional Geology of Yunnan Province
 Ch.11&12 (part D): 222–253 (Geological Publishing House, 1990).
 Zhang D., Yan M., Fang X., Yang Y., Zhang T., Zan J., Zhang W., Liu, C & Yang Q.
 Magnetostratigraphic study of the potash-bearing strata from drilling core ZK2893 in the Sakhon
 Nakhon Basin, eastern Khorat Plateau. *Palaeogeogr Palaeoclimatol Palaeoecol*. **489**, 40–51
 (2018).

 Figure R1. Summary of previously published sedimentary ages for upper Cretaceous to lower Palaeocene
 deposits in eastern Tibet. The early Cenozoic paleogeography is from Royden et al. (2008). Base map is
 from Google Earth.

**(2) Reviewer #1:** Even if a trans-continental river system flowed to the Neo-Tethyan Ocean,
about the river course more evidence should be provided, e.g., provenance correlation to the
west Burma? And to Khorat Basin? Cai F.L. et al. (2020, GSAB) indicated that the Upper
Cretaceous-Eocene strata are mainly sourced from the western Myanmar Arc with detrital
zircon age peaks of 100-60 Ma. This is totally different with the coeval strata of the eastern
Tibetan basins in this study. Thus, in the reconstruction map Figure. 4, I strongly doubt the
river flowed Simao via Myanmar to the Neo-Tethyan Ocean. Actually, the paleocurrent of the
Simao Basin would suggest a connection with the Khorat basin (See Yan M.D., et al., 2021,
and references therein).

**Reviewer #3:** The evidence of lack K2-E1 terrestrial deposits in the South China Sea to
against a paleo-river flowed to the Proto-Pacific Ocean is not the truth. For example, as
shown in the Fig. 3 of Clift et al., 2006 EPSL, the Paleogene sediments are very thick in the
South China Sea. So the authors have to find strong evidences to support why a
continental-scale river was flowed to the Neo-Tethyan Ocean rather than the Proto-Pacific
Ocean.

**Reviewer #3:** If a southwest flow river to the Neo-Tethyan Ocean existed in the Late
Cretaceous, more evidence from the Indochina and Sibumasu terranes must be shown,
currently the westernmost basin shown in the manuscript is the Simao Basin, which cannot
preclude the possibility that the river flowed to the South China Sea as previous model
suggested.

**Reply:** Please see our response to the comment by the Associate Editor above (**numbers 1–5;**
**Lines 8–80** in this letter). Also, please note that although Clift et al. (2006, EPSL) interpreted
sedimentary deposits in the South China Sea to be Palaeogene in age (see Figure 3 of Clift et
al., 2006, EPSL), borehole data have subsequently revealed that these terrestrial deposits are
not older than late Eocene (~37 Ma) (as shown the Figure 4 in Clift et al., 2006 (GRL) and
Clift et al., 2008, and Figure 2 in Wang et al., 2019).

**References**

- Clift, P. D. Controls on the erosion of Cenozoic Asia and the flux of clastic sediment to the ocean.
*Earth Planet. Sci. Lett.* **241**, 571–580 (2006).
- Clift, P. D., Blusztajn, J. & Nguyen, A. D. Large-scale drainage capture and surface uplift in eastern
Tibet-SW China before 24 Ma inferred from sediments of the Hanoi Basin, Vietnam. *Geophys. Res.*
*Lett.* **33**, L19403 (2006)
- Clift, P. D., Long, H. V., Hinton, R., Ellam, R. M., Hannigan, R., Tan, M. T. et al. Evolving east Asian
river systems reconstructed by trace element and Pb and Nd isotope variations in modern and
ancient Red Rive-Song Hong sediments. *Geochem., Geophys., Geosyst.* **9**, Q04039 (2008).
- Wang, C., Liang, X., Foster, D. A., Liang, X., Zhang, L. & Su, M. Provenance and drainage evolution
of the Red River revealed by Pb isotopic analysis of detrital K-feldspar. *Geophys. Res. Lett.* **46**,
6415–6424 (2019).

**Response to the comments of the three reviewers**

**Reviewer #1**

The evolution of large river is always a hot topic in earth science and attract huge interest.
This study provide the traditional provenance results of the Upper Cretaceous to Paleocene in
four basins, eastern Tibet, including petrographic modal counting and heavy mineral analysis
for 22 samples and detrital zircon U-Pb ages for 16 samples. The results show the provenance
similarities between the samples, the Songpan-Ganzi, Yidun, and Yangtze terranes, and the
upper Jinsha, Min and Yalong rivers sand. Based on the provenance interpretation, the authors
believed that there was a long-lived continental-scaled river system flowed into the
Neo-Tethyan Ocean which generated a low-relief landscape in eastern Tibet. However, the
provenance analysis and drainage reconstruction is questionable and thus conclusions in the
manuscript are not reasonable. Moreover, there are several similar studies published before
which eclipsed the significance of this study, not to mention several big weaknesses. Thus, I
have to reject the manuscript for publication in *Nature Communications*.

**Reply:** As acknowledged by the reviewer, the paleo-drainage evolution and low-relief
landscape formation in eastern Tibet have been debated for a long time. We acknowledge that
we are not the first to propose a large-scale south-flowing river system prior to the India-Asia
collision. However, the most significant finding/highlight of our manuscript is to link the
development of this long-lived paleo-river system to the landscape evolution and the
formation of the low-relief landscape in present-day eastern Tibet before Cenozoic uplift and
plateau growth. Therefore, we argue that our study is of great significance and warrants
publication in *Nature Communications*.

We have carefully considered the comments of reviewer #1. Below, we respond in detail
to his/her major comments and argue that our interpretation is justified and robust.

**>Major Comments**

(1) The samples of the Simao Basin are from the Denghei Formation, which is believed to be
deposited during Paleocene to late Eocene. The Upper Cretaceous to Lower Paleocene in the
Simao Basin is the Mengyejing Formation which have robust radioactive and
magnetostratigraphic constraints (112~>63 Ma, Yan M. et al., 2021, Science China Earth
Science; Wang L. et al., 2015, Cretaceous Research). Please use provenance similarities to
link continents and restore paleo-drainage systems only when the sedimentary strata are
formed within a same time. Without correlation between the coeval strata, it is not convincible
that the large-scale river can be flow into Simao and further south.

**Reply:** Although the Denghei Formation of the Simao Basin was assigned a Palaeocene–late
Eocene age by Chen et al. (2017), their age scheme is based on palaeontology only. Chen et al.
(2017) state that “*Paleocene to Eocene ostracods are present in the Denghei Fm., including*
*Pinnocypris, Limnocythere, Ilyocypris, Cyprinotus, together with typical Paleocene*
*charophytes of Gyrogona, Obtusochara, Peckichara*”). Since almost all of the above
ostracods also occur in the lower Paleogene strata presented in our study (see Supplementary
information) and because the magnetostratigraphic age of the Mengyejing Formation is
112~>63 Ma (Yan et al., 2021), we argue that the Mengyejing Formation and the overlying
Denghei Formation form a continuous sedimentary succession (which is older than late
Eocene). Furthermore, the stratigraphic units studied by us can be correlated well with the
Mengyejing Formation and Denghei Formations. For example, they all formed as red beds
during a long period with dry climate (Regional Geology of Sichuan Province, 1991;
Regional Geology of Yunnan Province, 1990) and were likely deposited during the same time
interval. In the revised manuscript, we have added previously reported detrital zircon data

from the late Cretaceous Mengyejing Formation in the Simao basin for comprehensive
provenance analysis (for details see lines 212–216, Fig. 3, and Figs. S4–6 in the revised
manuscript).

**References**

Chen, Y., Yan, M., Fang, X., Song, C., Zhang, W., Zan, J. et al. Detrital zircon U-Pb geochronological
and sedimentological study of the Simao Basin, Yunnan: Implications for the early Cenozoic
evolution of the Red River. *Earth Planet. Sci. Lett.* **476**, 22–33 (2017).

Sichuan Bureau of Geology and Mineral Resources. Regional Geology of Sichuan Province h.11&12
(part I): 264–282 (Geological Publishing House, 1991).

Yan, M. D., Zhang, D. W., Fang, X. M., Zhang, W. L., Song, C. H et al. New Insights on the age of the
Mengyejing formation in the Simao basin, SE Tethyan domain and its geological implications. *Sci*
*China Earth Sci.* **64**, 231–252 (2021).

Yunnan Bureau of Geology and Mineral Resources. Regional Geology of Yunnan Province Ch. 11&12
(part I): 222–253 (Geological Publishing House, 1990).

(2) The depositional environment is essential to reconstruct the paleo-river system. In
Supplementary Information Xichang Basin section, for the Xiaoba and Leidashu Formations,
Deng et al. (2018, ref. 16) never proposed a meandering river but a fluvial and
shallow-lacustrine environment. Also, ref. 25 indicates a lacustrine environment. Together
with the occurrence of evaporites, I think the lacustrine environment is more reasonable.

**Reply:** We have double-checked Deng et al. (2018, GSAB) and disagree with reviewer #1.
Deng et al. (2018) did not suggest a meandering river environment, but proposed a fluvial
environment for the Xiaoba and Leidashu Formations (page 9, lines 1–15). The lithological
assemblage of the Xiaoba and Leidashu Formations in the Xichang Basin is characterized by
alternating reddish sandstone, siltstone, and mudstone. Specifically, meter-thick sandstone
beds with sharp erosional bases (Fig. R2) are most likely the result of lateral fluvial erosion
and deposition (cf. Miall, 1996).

Figure R2. Typical fluvial sandstone bodies in Upper Cretaceous–lower Paleogene sedimentary rocks of the Xichang Basin (a), Huili Basin (b), and Chuxiong Basin (c). All photographs taken by Xudong Zhao.

Apart from lacustrine environments, siltstones and mudstones are also common in modern continental-scale river systems, especially in extensive and low-gradient floodplains of anastomosing or meandering rivers that provide accommodation space for fine-grained sediments (as shown the Figure 3b in Ashworth et al., 2012). Moreover, the present-day largest anastomosing rivers in the world often develop fine-grained sediments, with lakes and wetlands between levee-flanked channel branches and stable alluvial islands that divide flow up to bankfull (Knighton and Nanson, 1993; Abbado et al., 2005). In other words, the presence of fine-grained sediments in big river systems (as shown the Figure 3b in Ashworth et al., 2012) should not be taken as evidence for an overall lacustrine environment. Hence, we argue that our interpretation of a low-energy, muddy anastomosing or meandering river is reasonable for the Xichang Basin.

References

- Abbado, D., Slingerland, R. & Smith, N.D. Origin of anastomosis in the upper Columbia River, British Columbia, Canada. In: Blum, M.D., Marriot, S.B., Leclair, S.M. (Eds.), *Fluvial Sedimentology VII: Special Publication of the International Association of Sedimentologists*, 35. Blackwell, Oxford, UK, pp. 3–1 (2005).
- Ashworth, P. J. & Lewin, J. How do big rivers come to be different? *Earth Sci Rev.* **114**, 84–107 (2012).
- Chen, Z., Xu, K. & Watanabe, M. Dynamic hydrology and geomorphology of the Yangtze River. In: Gupta, A. (Ed.), *Large Rivers: Geomorphology and Management*. Wiley, Chichester, pp. 457–469 (2007).
- Deng, B., Chew, D., Jiang, L., Mark, C., Cogne, N., Wang, Z. J. & Liu, S. G. Heavy mineral analysis

and detrital U-Pb ages of the intracontinental Palaeo-Yangtze basin: Implications for a
transcontinental source-to-sink system during Late Cretaceous time. *Geol. Soc. Am. Bull.* **130**,
2087–2109 (2018).

Knighton, D. & Nanson, G. Anastomosis and the continuum of channel pattern. *Earth Surf. Process.*
*Landf.* **18**, 613–625 (1993).

Miall, A. D. The Geology of Fluvial Deposits: Sedimentary Facies, Basin Analysis, and Petroleum
Geology, 362–365, *Springer* (1996).

(2 continued): In Chuxiong Basin section, I strongly doubt the interpretation of the exorheic
lake because the thick evaporites are occurred in the Jiangdihe Formation. Similarly, the
Upper Cretaceous Mengyejing Formation developed several hundred meters of evaporites,
indicative of an endorheic basin. How to understand the drainage pattern when the large river
flow into the Chuxiong and Simao Lake?

**Reply:** With respect to this comment, we note that “*Evaporite minerals are deposited within*
*fluvial sub-strates by the evaporation of groundwaters and on the surface of playa mudflats*
*during the evaporation of sheet floods. Thin beds, laminae, nodules, and individual crystals*
*(or crystal casts) of evaporite, particularly gypsum and halite, are very common in the*
*deposits of arid fluvial systems, especially in the distal regions where the braidplain or*
*terminal fan merges imperceptibly into a playa lake or arid tidal flat (Smoot 1983; Glennie*
*1987; Mertz and Hubert 1990).” cited from Miall (1996, p. 441). This sentence from Miall*
*(1996) indicates that the presence of evaporites does not necessarily require an internal*
*drainage pattern. During our field investigations in the Chuxiong Basin, we did not observe*
*continuous and/or thick pure evaporites in the Jiangdihe Formation. The formation*
*mechanism of small-scale evaporite rhythms is that “inflowing runoff ‘freshens’ the brine*
*body and this, together with cooler air temperatures, causes either cessation of evaporite*
*precipitation or precipitation of a less undersaturated phase—the runoff also brings in the*
*suspended clastic sediment” (Leeder, 2011). Thus, to deny the existence of a through-going*
*fluvial system on the basis of local evaporites is unsound, especially when considering the*
*warm climate conditions during the Late Cretaceous to early Cenozoic.*

From the published literature it may indeed appear that several hundred meters of
evaporites occur in the Upper Cretaceous Mengyejing Formation of the Simao basin (e.g.,
Wang et al., 2020; Yan et al., 2021). However, many studies have demonstrated that these late
Cretaceous–early Paleogene evaporites in the Simao basin (and the Khorat basin farther south)
are mainly of **marine origin** and formed during incursions of the Tethys ocean (e.g., Hite and
Japakasetr, 1979; El Tabakh et al, 1999; Zhang et al., 2013; Liu et al., 2018; Wang et al., 2020;
Qin et al., 2020). As a consequence, the presence of these evaporites cannot be used as an
argument against a fluvial system that drained into the Neo-Tethyan Ocean. We have clarified
this issue in the revised **supplementary information (lines 203–214)**.

**References**

El Tabakh, M., Utha-Aroon, C. & Schreiber, B.C. Sedimentology of the Cretaceous Maha Sarakham
evaporites in the Khorat Plateau of northeastern Thailand. *Sediment. Geol.* **123**, 31–62 (1999).

Glennie, K. Desert sedimentary environments, present and past—a summary. *Sediment Geol.* **50**, 135–
165 (1987).

Hite, R.J. & Japakasetr, T. Potash deposits of Khorat Plateau, Thailand and Laos. *Econ. Geol.* 74, 448–
458 (1979).

Leeder, M. *Sedimentology and Sedimentary Basins from Turbulence to Tectonics*, 117.
*Wiley-Blackwell* (2011).

Liu, C.L., Wang, L.C., Yan, M.D., Zhao, Y.J., Cao, Y.T., Fang, X.M., Shen, L.J., Wu, C.H., Lv, F.L. &
Ding, T. The Mesozoic-Cenozoic tectonic settings, paleogeography and evaporitic sedimentation of
Tethyan blocks within China: implications for potash formation. *Ore Geol. Rev.* **102**, 406–425
(2018).

Mertz, K. J. & Hubert, J. Cycles of sand-flat sandstone and playa-lacustrine mudstone in the
Triassic-Jurassic Blomidon redbeds, Fundy rift basin, Nova Scotia; implications for tectonics and
climatic controls. *Can J Earth Sci.* **27**, 442–451 (1990).

Miall, A. D. *The Geology of Fluvial Deposits: Sedimentary Facies, Basin Analysis, and Petroleum*
*Geology*, 441, *Springer* (1996).

Qin, Z., Li, Q., Zhang, X. G., Fan, Q. S., Wang, J. P., Du, Y. S., Ma, Y. Q., Wei, H. C., Yuan, Q. &
Shan, F. S. Origin and recharge model of the late cretaceous evaporites in the khorat plateau. *Ore*
*Geol Rev.* **116**, 103226 (2020).

Smoot, J. Depositional subenvironments in an arid closed basin; the Wilkins Peak Member of the
Green River Formation (Eocene), Wyoming, USA. *Sedimentology* **30**, 801–827 (1983).

Wang, L. C., Zhong, Y. S., Xi, D. P., Hu, J. F. & Ding, L. The Middle to Late Cretaceous marine
incursion of the Proto-Paratethys sea and Asian aridification: A case study from the Simao-Khorat
salt giant, southeast Asia. *Palaeogeogr Palaeoclimatol Palaeoecol.* **567**, 110300 (2021).

Zhang, X.Y., Ma, H.Z., Ma, Y.Q., Tang, Q.L. & Yuan, X.L. Origin of the late Cretaceous
potash-bearing evaporites in the Vientiane Basin of Laos: $\delta^{11}\text{B}$ evidence from borates. *J. Asian*
*Earth Sci.* **62**, 812–818 (2013).

(3a) About the potential sources. I doubt if you correctly plot the age distribution of the
Yangtze terrane. Because all 1191 zircon ages in figure 3 and extended data figures 4&6 don't
show a prominent age peak of 750-1000 Ma which is believed to be a diagnostic feature of
the Yangtze. I tried to find the paper you cited (refs. 14& 30) and found that the zircon ages in
these papers are from the Early Paleozoic. So why the age peaks at ca. 150 Ma and 220 Ma
occur? I don't know the specific samples of the 1191 ages although you mentioned they were
from the pre-late Cretaceous strata. Thus, the relationship between the ages <200 Ma and the
Yangtze terrane need to be further proved.

**Reply:** We are sorry for the misunderstanding about the use of this name for the source area.
In the submitted manuscript, the term “Yangtze terrane” was meant to be the present-day
Sichuan Basin, to avoid the confusion between “Sedimentary Basin” and “Source Region”.
The 1191 zircon ages are all from the pre-late Cretaceous basement in the Sichuan Basin (Li
et al., 2018). The Jurassic–early Cretaceous zircons (peak at ca. ~150 Ma) in the pre-late
Cretaceous basement in the Sichuan Basin were likely derived from the southern margin of
the North China Block (e.g., the Qinling Belt, Dabie Belt) (Li et al., 2018). Regarding the
zircon age peak at ~220 Ma, there are several potential source areas, including the Songpan–
Ganzi and Yidun terranes to the west, the Qinling Belt to the north, and the Western Jiangnan
orogen to the east (Li et al., 2018).

It is likely that the source area with a prominent age peak of 750–1000 Ma mentioned by
the reviewer reflects the western South China block (see Fig. S4), where a variety of
Neoproterozoic strata/rocks along the western margin yield age peaks at 700–900 Ma (e.g., Li
et al., 2003; Sun et al., 2003). In the revised manuscript, we now use “Upper Yangtze terrane”

(cf. Huang et al., 2021), instead of “Yangtze terrane”, to refer to the generalized source region
for the late Cretaceous–early Palaeogene deposits in the “Sichuan Basin”. For the locations of
the Upper Yangtze terrane and the western South China, please refer to Fig. 1a. Corrections
have also been made in **line 103** of the text.

**References**

- Li, Y. Q., He, D. F., Li, D., Lu, R. Q., Fan, C., Sun, Y. P. & Huang, H. Y. Sedimentary provenance
constraints on the Jurassic to Cretaceous paleogeography of Sichuan Basin, SW China. *Gondwana*
*Res.* 60, 15–33 (2018).
- Li, Z. X., Li, X., Kinny, X., Wang, J., Zhang, S & Zhou, H. Geochronology of Neoproterozoic
syn-rift magmatism in the Yangtze Craton, South China and correlations with other continents:
Evidence for a mantle superplume that broke up Rodinia. *Precambrian Res.* **122**, 85–109 (2003).
- Sun, W. H., Zhou, M. F., Gao, J. F., Yang, Y. H., Zhao, X. F., & Zhao, J. H. Detrital zircon U-Pb
geochronological and Lu-Hf isotopic constraints on the Precambrian magmatic and crustal
evolution of the Western Yangtze block, SW China. *Precambrian Res.* **172**, 99–126 (2009).
- Huang, H. Y., He, D. F., Li, Y. Q., Li, D., Zhang, Y. Y., Chen, J. J. Late Permian tectono-sedimentary
setting and basin evolution in the Upper Yangtze region, South China: Implications for the
formation mechanism of intra-platform depressions. *Journal of Asian Earth Sciences*, **205**, 104599
(2021).

(3b) Line 171-172, we cannot interpret the provenance just based on the age peaks. Why the
Lhasa Terrane just simply provides the late Cretaceous zircons to the Chuxiong Basin, not
conclude the other age populations? If you plot the Lhasa in the MDS diagram, you will find
the Lhasa plot apart from your samples.

**Reply:** As discussed in the submitted manuscript, the Lhasa terrane was not a dominant
source area for the late Cretaceous–early Palaeogene strata, but possibly provided some
Cretaceous zircon component to these basins, because the provenance signal of the Lhasa
terrane is characterized by a single Cretaceous age-peak (see Figure 12 in Yan et al., 2021).
We interpret the sediment transport system from the Lhasa terrane as a small tributary of the
proposed south-flowing river system. This explains why the Lhasa source would plot apart
from our samples in the MDS diagram. Please note that we do not plot the data from the
Lhasa terrane in the MDS diagram (Fig. S6).

Except for the Chuxiong Basin, the late Cretaceous-early Paleogene strata in the Simao
Basin also contain appreciable Cretaceous zircons (Yan et al., 2021), which further supports
the existence of a river system connected to the eastern Lhasa terrane at that time. Consistent
with provenance evidence, previous thermochronologic studies have indicated that there was
most likely an externally drained river system from the Lhasa terrane to the ocean during the
late Cretaceous–early Palaeogene (Hetzl et al., 2011; Haider et al., 2013). Please note that we
cannot rule out the possibility that the externally drained river systems originating in the
Lhasa terrane were independent from our proposed continental-scale palaeo-drainage system
(see Fig. R3). We have explained this issue on **lines 186–191** of the revised manuscript.

Figure R3. Plate reconstruction of East Asia during the latest Cretaceous showing an alternative drainage
 model with several river systems draining the Lhasa or/and Qiangtang terranes to the Neo-Tethys (based on
 Hetzel et al., 2011; Haider et al., 2013; Gourbet et al., 2016). The map is generated by Xudong Zhao using
 open-access GPlates software (accessed through <https://www.gplates.org/>).

**References**

Gourbet, L., G Mahéo, Shuster, D. L., Tripathy-Lang, A., Leloup, P. H. & Paquette, J. L. River
 network evolution as a major control for orogenic exhumation: case study from the western tibetan
 plateau. *Earth Planet Sci Lett.* **456**, 168–181 (2016).
 Haider, V.L., Dunkl, I., Eynatten, H., Von Ding, L., Frei, D. & Zhang, L. Cretaceous to Cenozoic
 evolution of the northern Lhasa Terrane and the Early Paleogene development of peneplains at
 Nam Co, Tibetan Plateau. *J. Asian Earth Sci.* **70–71**, 79–98 (2013).
 Hetzel, R., Dunkl, I., Haider, V., Strobl, M., von Eynatten, H., Ding, L. & Frei, D. Peneplain formation
 in southern Tibet predates the India-Asia collision and plateau uplift. *Geology* **39**, 983–986 (2011).
 Yan, M. D., Zhang, D. W., Fang, X. M., Zhang, W. L., Song, C. H et al. New Insights on the age of the
 Mengyejing formation in the Simao basin, SE Tethyan domain and its geological implications. *Sci*
 *China Earth Sci.* **64**, 231–252 (2021).

(3c) In the MDS plot (Figure 5b), clearly, the Min River and Yalong River plot apart from the
 samples. The age peaks of Yalong River sand are different from the samples and
 Songpan-Ganzi. Only Upper Jinsha river plot closer to the samples. So how do you explain
 your drainage reconstruction?

**Reply:** The reviewer brings up an important point. For exploring this issue, we collected more
 detrital zircon data of the Yalong River from the literatures (Yang et al., 2012; He et al., 2013),
 and found that the detrital zircon age components of the Yalong River indeed differ from our
 late Cretaceous–early Palaeogene samples, because the Yalong River displays (1) a higher
 abundance of 600–1000 Ma zircon grains derived from the western margin of the South China
 block, and (2) shows a secondary age peak at 250–200 Ma that likely indicates a contribution
 from the Yidun Terrane (Fig. S4b). However, the sand sample from the Minjiang River that
 drains the Songpan–Ganzi region, plots closer to our K₂–E₁ samples in the MDS plot,

supporting that the Triassic flysch of the Songpan–Ganzi terrane is the primary source for
these K₂–E₁ strata. We have updated the revised manuscript to clarify this issue in **lines 170–**
**172** of the revised manuscript and in Figs. S4 and S6.

**References**

He, M., Zheng, H., Clift, P.D. Zircon U–Pb geochronology and Hf isotope data from the Yangtze River
sands: implications for major magmatic events and crustal evolution in central China. *Chem. Geol.*
360–361, 186–203 (2013).

Yang, S., Zhang, F., Wang, Z. Grain size distribution and age population of detrital zircons from the
Changjiang (Yangtze) River system, China. *Chem. Geol.* 296, 26–38 (2012).

(4) As mentioned above, for the reasons of the depositional environment and provenance
interpretation, I disagree with the idea that a large-scale river system that connect the all the
so-called exorheic basins although I agree somehow a south-flowing river exists. Specifically,
the Chuxiong and Simao are two endorheic basins with several hundred meters of evaporites.
Perhaps, the river just stop when it flowed into these two basins.

**Reply:** We have explained this issue above in our response to the reviewer’s major point 2.

(4 continued) More importantly, large river system in the Late Cretaceous flowed to the
Neo-Tethyan Ocean had been proposed by Yan Maodu et al., 2021, *Science China Earth*
*Science*. So it is not the authors who claimed that they proposed it for the first time. I found
that there are several papers proposed a large river system prior to the collision between India
and Asia, e.g., Deng et al. (2018, ref. 16); Wang L.C., et al. 2020, *Palaeo-3*; Yan M., et al.,
2021.

**Reply:** We acknowledge that we are not the first to propose a large-scale south-flowing
river system prior to the India and Asia collision. Our work does lend new support to these
previous assertions. Moreover, the most significant and novel finding of our study is to link
the development and extent of this long-lived paleo-river system to the formation of the
low-relief landscape in eastern Tibet before Cenozoic uplift and plateau growth.

**(5) Data and methodology**

(5-1) Petrography part. The authors should tell the readers what kind of method you use when
you do modal analysis. Usually, sedimentary geologists use the Gazzi-Dickinson method
(Ingersoll et al., 1984). And all ternary diagrams should cite the original references. >In
Extended data Figure 2, you use Lc to represent the carbonatite lithic, however, the Ls was
used in the Table S2. Moreover, I strongly doubt the high percentage of carbonatite lithic in
the samples, if yes, why don’t you count these into the Lv?

**Reply:** We appreciate the advice from the reviewer. We added the original references
(Dickinson et al., 1983; Ingersoll et al., 1984) to the revised manuscript. We have also made
the abbreviations for the different lithic components consistent throughout the revised
manuscript: Ls is terrestrial sedimentary lithic, Lc refers to carbonate lithic, and Lv is
volcanic lithic (the latter should not be combined with the carbonate lithic).

**References**

Dickinson, W.R., Beard, S.L., Brakenridge, G.R., Erjavec, J.L., Ferguson, R.C., Inman, K.F., Knepp,
R.A., Lindberg, F.A. & Ryberg, P.T. Provenance of North American Phanerozoic sandstones in
relation to tectonic setting. *Geol. Soc. Am. Bull.* **94**, 222–235 (1983).

Ingersoll, R. V., Bullard, T. F., Ford, R. L., Grimm, J. P., Pickle, J. D. & Sares, S. W. The effect of
grain size on detrital modes: A test of the Gazzi-Dickinson point-counting method. *J Sediment*
*Petrol.* **54**, 103–116 (1984).

(5-2) For all 22 samples for petrographic and heavy mineral analysis, I suggest the authors
give a table that contains the GPS and stratigraphic information (I can't tell which formation
and basin of some samples belong to, e.g., CX-29, CX-01, CX08). In Extended data Figure 1,
the authors should locate the heavy mineral samples in the stratigraphic columns of the
Xichang, Huili, and Chuxiong basins.

**Reply:** As requested by reviewer #1, we added a table (Table S3) that contains the GPS and
stratigraphic information. In Fig. S1, all detrital zircon samples from the Xichang, Huili, and
Chuxiong basins were also analyzed for heavy minerals, so locations of heavy mineral and
detrital zircon samples from the Xichang, Huili, and Chuxiong overlap (please see legend of
Fig. S1, where red stars refers to samples used for heavy mineral and detrital zircon analysis).

(5-3) Detrital zircon geochronology part. More information should be provided to let the
readers examine the reliability of your data. What are the dating results of your age external
standards? And how do these results compared to the suggested age values? So you should
provide the dating results the standard zircons. In addition, relevant citations should be given
regarding external standards. The representative CL images of detrital zircons especially the
ones with young ages (<200 Ma in this area) is very important. Because in my opinion, these
young zircons are mostly from local source.

**Reply:** To determine fractionation factors and correct for instrumental drift, two standards
(91500 and GJ-1) were analyzed every 10 grains; element content was determined by
NIST610 as external standard. We have revised this part in the manuscript (lines 298–304).
All dating results of the standard zircons are available and the zircon standards are described
in Yuan et al. (2004). We think that an extensive method description is not required, because
detrital zircon geochronology is a standard tool and detailed information on methodology, age
standards, and analytical procedures are available in the cited references (Andersen, 2002;
Yuan et al., 2004).

As zircon grains of different age were indistinguishable based solely on CL images, we
refrain from showing CL images. Ages <200 Ma most likely reflect recycled grains from
pre-late Cretaceous strata in the Sichuan Basin (Upper Yangtze terrane), as suggested by
petrographic and heavy mineral data.

**References**

Andersen, T. Correction of common lead in U-Pb analyses that do not report ²⁰⁴Pb. *Chem. Geol.* **192**,
59–79 (2002).

Yuan, H. L., Gao, S., Liu, X. M., Li, H. M., Günther, D. & Wu, F. Y. Accurate U-Pb age and trace
element determinations of zircon by laser ablation inductively coupled plasma mass
spectrometry. *Geostand. Geoanal. Res.* **28**, 335–370 (2004).

(5-4) The method of MDS is not sufficiently explained. (a) How the MDS map was generated?
(b) Which metric do you choose when you plot the MDS, e.g., likeness, similarity.....?

**Reply:** We added a method description on how we generated the MDS plot. Note that the
MDS method used (cf. Vermeesch, 2013) is based on ‘dissimilarities’ between samples.

**Reference:** Vermeesch, P. Multi-sample comparison of detrital age distributions. *Chem. Geol.* **341**,

140–146 (2013).

(5-5) Line 461-464, the K-S test p-value is not recommended to use. Use of the K-S or Kuiper
test p-values for quantitative similarity analysis of detrital geochronological data sets is likely
to lead to incorrect conclusions (Satkoski et al., 2013, GSA Bulletin; Vermeesch, 2013; Saylor
and Sundell, 2016, Geosphere). As noted by Vermeesch (2013), the D or V values provide
more robust assessment of the dissimilarity between samples than do p-values. Thus, the D or
V values of K-S or Kuiper tests are suggested to use.

**Reply:** Thank you for this helpful comment. As suggested, we replaced the K-S test p-value
by the *D* values of K-S and the *V* values of the Kuiper tests (please see improved Table S1).

**(6) Minor Comments:**

(6-1) Line 103, Yangtze is believed to be a part of South China block, why did you show a
different concept? The same as in Figure 1a.

**Reply:** We have explained this issue above in our response to the reviewer's major point 3a.

(6-2) Line 125-128, I can't understand why the late Cretaceous-early Paleogene strata could
represent the youngest terrestrial clastic deposits? At least the Lower Cretaceous in these
basins are terrestrial clastic rocks.

**Reply:** This appears to be a misunderstanding by the reviewer. Apart from very limited
Quaternary sediments (i.e. the Xigeda Formation), the late Cretaceous–early Paleogene strata
are the youngest terrestrial clastic deposits at the eastern margin of Tibet. Of course, there are
also older sedimentary strata of Lower Cretaceous age (but these are not the focus of our
study).

(6-3) Line 131, the same as the previous comment, actually the Upper Cretaceous evaporates
developed in the SW Sichuan, Xichang, Chuxiong, and Simao Basins (Liu Shugen et al., 2019,
Journal of Chengdu University of Technology, v.46, No. 1, 1-28; and references therein).
Especially, thick evaporites were developed in the Chuxiong and Simao Basin (Liu Chenglin
et al., 2018, Ore Geology Reviews).

**Reply:** We have explained this issue in our response to the previous comments above. Here,
we only add that Liu Chenglin et al. (2018, Ore Geology Reviews) did not propose that thick
evaporites occur in the Chuxiong basin, which is consistent with our field investigations.

**Reference:**

Liu, C.L., Wang, L.C., Yan, M.D., Zhao, Y.J., Cao, Y.T., Fang, X.M., Shen, L.J., Wu, C.H., Lv, F.L. &
Ding, T. The Mesozoic-Cenozoic tectonic settings, paleogeography and evaporitic sedimentation of
Tethyan blocks within China: implications for potash formation. *Ore Geol. Rev.* **102**, 406–425
(2018).

(6-4) Line 135-137, about the depositional environment interpretation, pls see my comments
above.

**Reply:** We have already addressed this issue in our response to the major comment 2 of
reviewer #1 above.

(6-5) Line 204-205, during K2-E1, the global sea level was gradually rose from ca. 80 Ma to
60-50 Ma, and fell since 50 Ma (Miller et al., 2005, Science, 10.1126/science.1116412). Thus,
it is not stable. How to understand?

**Reply:** We agree that sea level was not stable in a rigorous sense. Nevertheless, Figs. 2 and 3
in Miller et al. (2005) indicate that the global sea level rose very slowly (but did not change
significantly between ~92 Ma and ~55 Ma (as shown the Figure 3 in Miller et al., 2005). In
the revised manuscript, we explain that the sea level was slowly rising (line 233). Note that
the slowly rising sea level, coupled with an arid climate, could well be responsible for the
marine incursions to the Simao to Khorat basins and the formation of the marine evaporites
there.

**References**

Miller, K. G., Kominz, M. A., Browning, J. V., Wright J. D., Mountain, G. S., Katz, M. E, Sugarman, P.
679 J., Cramer B. S., Christie-Blick, N. & Pekar S. F. The Phanerozoic Record of Global Sea-Level
Change. *Science* **310**, 1293–1298 (2005).

(6-6) Actually, Deng B. et al. (2018, GSAB) published many detrital zircon ages of the Upper
Cretaceous-Paleocene in Sichuan, Chuxiong basins. I suggest the authors should compile all
published detrital zircon ages together with this study to do provenance analysis.

**Reply:** The error calculation of the detrital zircon data of Deng B. et al. (2018) and ours are
different. Moreover, zircon age spectra from Deng et al. (2018) are largely similar to our data,
thus compiling more detrital zircon ages would not change the conclusions of our study.

**References**

Deng, B., Chew, D., Jiang, L., Mark, C., Cogne, N., Wang, Z. J. & Liu, S. G. Heavy mineral analysis
and detrital U-Pb ages of the intracontinental Palaeo-Yangtze basin: Implications for a
transcontinental source-to-sink system during Late Cretaceous time. *Geol. Soc. Am. Bull.* **130**,
2087–2109 (2018).

(6-7) Figure. 1, where is the Sichuan Basin?

**Reply:** We added the term "Sichuan Basin" (also called Upper Yangtze terrane as explained
above) in Fig. 1.

(6-8) Supplementary Information line 41-42, very thick evaporites were developed in the
Upper Cretaceous Chuxiong and Simao Basin. Thus, it is not an indication of river discharge
but an endorheic lake.

**Reply:** We have addressed this issue in detail in our response above.

(6-9) Extended figure 4a and 4b, all the references are not incorrect. I can't find any
mentioned samples in these cited papers (ref 31, 47, and 25). For the ref. 25, maybe you want
to refer to ref. 14 (Chen Y., et al., 2017, EPSL).

**Reply:** We are sorry that the numbers of the cited reference were incorrect. In the revised
manuscript, the reference numbering has been corrected.

(6-9 continued) However, the 272 zircon ages from ref. 14 belong to the Denghei Formation,
which was assigned a Paleocene-late Eocene in age based on the fossils. Thus, I doubt
whether it is reasonable to correlate to the Simao basin in this manuscript.

**Reply:** We have addressed this issue in our response to the second part of main point 1 above.

**Reviewer #2 (Andrew Laskowski)**

This study reports provenance data (detrital zircon and sandstone petrography) from
Cretaceous- Eocene sedimentary basins that are located along the southeast margin of the
Tibetan Plateau. The authors observe that the provenance data is very consistent between
basins and back up these observations with statistical methods that are mainly presented in the
supplementary figures. To explain these data, the authors argue that there was likely a
continent-scale river that connected the basins and drained to the Neo-Tethyan Ocean, which
separated India from Eurasia prior to India-Asia collision. The authors note that this
interpretation also has relevance to the debate over the low-relief, incised landscapes that are
located along the southeast margin of the Tibetan Plateau. This aspect of the research will
likely be the most broadly relevant and controversial aspect. Several dynamic models for
Tibetan Plateau growth invoke different mechanisms for formation of the low-relief
landscapes. This, coupled with the inherent interestingness of ancient, continental-scale rivers,
make this research exciting and broadly relevant.

As I am an expert on detrital zircon geochronology and the tectonics of the Tibetan
Plateau, I chose to focus on these aspects for my review. In my opinion, the provenance data
are robust and well supported by statistical methods. I would encourage the authors to include
some more discussion of their comparison between source terrane signatures and basin
signatures. A key point that must be proven is that the source areas are distinguishable based
on their detrital zircon age spectra whereas all the sampled basins are not, indicating mixing
by a large river system. The supplementary figures were very helpful to convince me of the
validity of the argument, yet they are sparsely discussed or referenced in the main text.

**Reply:** We sincerely thank Prof. Andrew Laskowski for his positive and constructive
comments. We applied multiple statistical methods to the zircon U-Pb age distributions
including probability density function plots, multidimensional scaling, DZStats, and DZMix
modeling. These different methods yielded consistent results.

As requested, we first emphasize the application of multiple methods and their consistent
results at the beginning of the provenance analysis section (**lines 155–158**). More importantly,
we also added more comparison and discussion details for each statistical method in the
revised manuscript (please see **lines 162–169**).

The authors also present a landscape evolution model that they link with their thermal
modeling results. They claim that it illustrates the plausibility of the continent-scale river
along the southeastern Tibetan Plateau because faster exhumation rates would be expected in
the upstream reaches of the drainages. This seems plausible to me, but the authors should also

emphasize that the Lhasa Terrane hosted an active, Cordilleran orogenic system at the time of
deposition of the samples. It has previously been likened to the modern Andes. It should not
be implied that fluvial drainage networks were solely responsible for the cooling of samples
further toward the interior of the Tibetan Plateau, as the tectonic activity in this region
undoubtedly affected these results.

**Reply:** We agree with the comment on the tectonically active Lhasa terrane during K₂–E₁ and
mention this issue in the revised manuscript (**lines 186–191**). Previous studies revealed
widespread uplift in the Lhasa area during the late Cretaceous to early Cenozoic (e.g.,
Rohrmann et al., 2012; Kapp and DeCelles, 2019), and argued that there was an externally
drained river system that connected the Lhasa terrane to the Neo-Tethys Ocean (Hetzl et al.,
2011; Haider et al., 2013). Thus, a combination of surface uplift due to crustal shortening and
fluvial erosion likely caused more rapid cooling in the Tibetan hinterland as revealed by
thermochronologic data (see Fig. 1). These observations indicate that the spatial trend in
exhumation rates from central to eastern Tibet created a gently topographic slope in the
eastern part of our envisaged paleo-river system.

**References**

- Haider, V.L., Dunkl, I., Eynatten, H., Von Ding, L., Frei, D. & Zhang, L. Cretaceous to Cenozoic
evolution of the northern Lhasa Terrane and the Early Paleogene development of peneplains at
Nam Co, Tibetan Plateau. *J. Asian Earth Sci.* **70–71**, 79–98 (2013).
- Hetzel, R., Dunkl, I., Haider, V., Strobl, M., von Eynatten, H., Ding, L. & Frei, D. Peneplain formation
in southern Tibet predates the India-Asia collision and plateau uplift. *Geology* **39**, 983–986 (2011).
- Kapp, P. & DeCelles, P.G. Mesozoic–Cenozoic geological evolution of the Himalayan-Tibetan orogen
and working tectonic hypotheses. *American Journal of Science.* **319**, 159–254 (2019).
- Rohrmann, A., Kapp, P., Carrapa, B., Reiners, P. W., Guynn, J., Ding, L. & Heizler, M.
Thermochronologic evidence for plateau formation in central Tibet by 45 Ma. *Geology* **40**, 187–190
(2012).

**Reviewer #3**

Comments to ‘Existence of a continental-scale river system in eastern Tibet during the late
Cretaceous–early Paleogene’ By Zhao et al. Based on new petro-stratigraphy, heavy-mineral
analysis, and detrital zircon U-Pb studies on late Cretaceous–early Paleogene sediments from
the east margin of Tibet, Zhao et al. proposed a novel continental-scale river flowing
southwestward to the Neo-Tethyan Ocean along the eastern Tibet in the late Cretaceous–early
Paleogene, and used this river to explain the formation of low-relief in the east margin of
Tibet. The topic of this study is of broad interest for geologists, and the proposed model is
significant different with previous models, which seems valuable for publication in *Nature*
*Communications*. However, there are some important unclears in the manuscript, which
should be addressed before acceptance.

(1) The authors argued that the K₂–E₁ sediments were not deposited in spatially separated
endorheic basins based on the lack of coarse-grained sediments and thick evaporites. However,
these are not strong evidence to preclude this possibility. It is very likely that these sediments
were derived locally by recycling from surrounding older rocks. For example, the Nangqian
and Gonjo basins in eastern Tibet show similar lithologies as the K₂-E₁ sediments by the

authors, but studies have shown that the two basins were sourced locally (Horton et al., 2002).
The authors also suggested that the low-relief could not be severed as physiographic barriers
for endorheic basins. However, endorheic basin can be formed in any landscape with local
structures, e.g., normal fault.

**Reply:** The sedimentary facies in the Xichang, Huili, Chuxiong, and Simao basins are
dominated by fluvial, lacustrine, and floodplain, lacking proximal facies (e.g., alluvial-fan)
and do not resemble the basins in the Nangqian-Yushu region argued to be internally drained.
Horton et al. (2002) suggested an internal drainage for basins in the Nangqian-Yushu region
of east-central Tibet based on centrally directed paleocurrents, dominantly lacustrine
depositional conditions, and a lack of single lithostratigraphic units that can be correlated
regionally among the basins. More importantly, preserved proximal facies (i.e., alluvial-fan)
were limited to basin margins, and fine-grained lacustrine deposition was mainly developed in
the present-day basin interior. Such distinct lateral facies evolution, together with growth
strata along the basin margins, indicates that the basins developed as distinct, isolated features
with dimensions approximately similar to their present-day outcrop areas.

If the K₂–E₁ sediments in this study had formed in endorheic basins, analogous to the
Nangqian basin, thick gravel or gravelly sandstone deposits or growth strata (near
basin-controlling faults) would be expected; however, such sedimentary deposits were not
identified by us.

From the perspective of provenance data, if the K₂–E₁ sediments were derived locally by
recycling from surrounding older rocks, we would expect the widespread Neoproterozoic
metamorphic rocks surrounding these basins (at the western margin of South China Block)
(e.g., Li et al., 2005) to provide a major source of detrital material and a unique prominent
peak at 600–900 Ma (similar to the provenance signal of the current Anning river, which
drains the western margin of the South China block; see Figure 2 in Yang et al., 2020).
However, our data clearly show that zircon age populations from all studied basins fall mainly
into five different groups of 200–300 Ma, 390–480 Ma, 700–900 Ma, 1700–2000 Ma, and
2300–2600 Ma. Furthermore, a slow long-term exhumation of the areas around these basins
from late Cretaceous to early Palaeogene is clearly documented by our compilation of
thermochronological data (Fig. 1b), indicating there was no significant local erosion occurring
around these basins at that time. In summary, we favor that all studied basins along eastern
margin of the Tibetan Plateau were characterized by externally drained rivers from late
Cretaceous to early Paleogene. We added this discussion to the revised manuscript (**lines 197–**
**207**).

Although we agree with reviewer #3 that endorheic basins can be formed in any
landscape with local structures, the combination of the axial distribution of the southwestern
Sichuan, Xichang, Huili, and Chuxiong basins along the foredeep depozone between eastern
Tibet and South China (Fig. 1) and their provenance interpretations, fits well with the model
of **Lateral tributary-dominated trans-continental river system** (e.g., Ganges, Mississippi,
Paraná) proposed by Ashworth et al. (2012) (as shown the Figure 2 in Ashworth et al., 2012).
“Big rivers may extend into sedimentary basins, but they also cross them (e.g. Danube,
Yangtze), and have longitudinally-extensive depositional zones” (Ashworth et al., 2012).

The identification of ancient big rivers would follow three elements based on the
suggestion from Miall. (2006), including (1) Prediction from plate-tectonic setting, (2)
Analysis of the scale of depositional elements, and (3) Study of sedimentary provenance. If it
is a case, a combination of long-term tectonic stability, several meters to tens of meters thick
channel beds, and common provenance signals presented our study, leads us to more strongly
believe the existence of a large-scale river system.

**References**

Ashworth, P. J. & Lewin, J. How do big rivers come to be different? *Earth Sci Rev.* **114**, 84–107
(2012).

Horton, B. K., Yin, A., Spurlin, M. S., Zhou, J. Y. & Wang, J. H. Paleocene-Eocene syncontractional
sedimentation in narrow, lacustrine-dominated basins of east-central Tibet. *Geol Soc Am Bull.*
**114**, 771–786 (2002).

Li, Z. X., Li, X. H., Kinny, P. D., Wang, J., Zhang, S., & Zhou, H. Geochronology of Neoproterozoic
syn-rift magmatism in the Yangtze Craton, South China and correlations with other continents:
evidence for a mantle superplume that broke up Rodinia. *Precambrian Res.* **122**, 85–109 (2005).

Miall, A. D. How do we identify big rivers? And how big is big? *Sediment Geol.* **186**, 39–50 (2006).

Yang, R., Suhail, H. A., Gourbet, L., Willett, S. D., Fellin, M. G., Lin, X. B., Gong, J. F., Wei, X. C.,
Maden, C., Jiao, R. H. & Chen, H. L. Early Pleistocene drainage pattern changes in Eastern
Tibet: Constraints from provenance analysis, thermochronometry, and numerical modeling.
*Earth Planet. Sci. Lett.* **531**, 115955 (2020).

(2) The biggest problem of this manuscript is the Fig. 4, which is inconsistent with the
tectonic background. The authors refer the reference of Muller et al. (2016) for the late
Cretaceous paleogeography reconstruction, but the map shown in Fig. 4a is inconsistent with
geological evidence: The Indochina and Sibumasu, which locate southwest of the South China
Block, have amalgamated to South China in Late Triassic, and the Lhasa has collided with
Qiangtang in the early Cretaceous, so it is impossible that a Neo-Tethyan Ocean was still
existed between South China and Indochina as shown in Fig. 4a.

**Reply:** We agree with this comment regarding the late Cretaceous paleogeography pattern
and acknowledge our mistake. We have now corrected Fig. 4 by integrating the Indochina and
Sibumasu terranes with the South China Block, and by showing the collided/amalgamated
Lhasa and Qiangtang terranes north of the Neotethys ocean.

Reviewers' Comments:

Reviewer #1:

Remarks to the Author:

The authors had a positive feedback to most reviews. However, I still see the insufficient interpretation or ignored reviews that I and editor gave.

The authors give four lines of additional evidence to support the river route to the Neo-Tethys Ocean. However, unfortunately, some evidence was incorrectly used by the authors. Firstly, the marine evaporites in the Simao-Khorat Basins reach a consensus, but discharge model in the cited paper is totally misunderstood by the authors. Recent publications (in cited papers) proposed that seawater incursions were from the Meso-Tethys Ocean or proto-Paratethys Sea which came from northern part of the Simao-Khorat Basins, not from the southern Neo-Tethys Ocean. Moreover, as the authors stated that the Khorat Basin was the last stop to the Neo-Tethyan Ocean, it is absolutely impossible to form such a salt giant in the Khorat Plateau Basins in an open water mass. The Khorat Plateau Basins cover an area of ca. 247,000 km² with evaporite thickness up to 1000 m (Hite and Japakasetr, 1979, Economic Geology). Another evidence to deny the river court is that all publications and geological observations tell us the evaporite-bearing Maha Sarakham Formation in the Khorat Basin was lacustrine environment. So when we considered the direction of marine incursion and salt giant, it would be reasonable why the Simao-Khorat are endorheic lakes.

Secondly, in Figure S8 (Supplementary Figure 8), the authors cited the data of Carter and Moss (1999, Geology) and show the age distribution of the Khorat Basin from so-called late Late Cretaceous and early Paleogene. However, I checked the paper of Carter and Moss (1999) and found the youngest strata in their study is the Early Cretaceous Khok Khruat Formation. There was totally no any data of the late Cretaceous to early Paleocene.

Based on these two evidence, the river court don't convince me.

(1) Age constraint

The age and depositional environment cannot convince me in the revised manuscript and that is what I am concerned about. Why the age is so important and I repeatedly stress it? K2-E1 is a large time range from 100 Ma to 56 Ma and you mentioned that it was a long-term tectonic stability. In this region, at least two tectonic events identified by many geologists: 1, the mid-Cretaceous (ca. 100 Ma) tectonic event (e.g., Lovatt-Smith et al., 1996) caused by collision between Qiangtang and Lhasa; 2, the India-Asia collision at ca. 65 Ma. The first event caused unconformity in the studied basins. It's hard to imagine the existence of such a long-term tectonic stable environment.

Of course, it's hard to do the geochronology work in thick red bed basins. But in my opinion, the charophyta and ostracod fossils would give wrong age constraints compared with the U-Pb ages. This is true in the Jianchuan Basin and Simao Basin (See Gourbet et al., 2017; Yan et al., 2021). So age issue is the first priority, or it would be another story.

About the MDA, the reason that you got so few youngest zircons is inadequate zircon tests. Most samples for the U-Pb analysis in the study is n=100/150. The 'large-n' datasets (e.g., > > 300 analyses per sample; Pullen et al., 2014; Daniels et al., 2018; Sundell et al., 2019a, 2019b) in order to increase the probability of analyzing young grains should be used.

(2) Depositional environment

The author's reply is actually not convinced me. We all know that the meandering river system typically consists of lag and sand bar deposits in the bottom and flood plain deposits showing a upward fining sequence. However, from your supplementary field photo Figure R6a and stratigraphic column Supplementary Figure 1, I cannot tell these features in the Xichang Basin. Moreover, I don't think a meandering environment that consist of a thickness of about 4,000 m of sandstone and claystones (Supplementary Figure 1) in the Xichang Basin.

For the Chuxiong Basin, I stated that "thick evaporite are occurred in the Jiangdihe Formation ". Yes, thin layers or nodules or crystals of evaporite can be formed in the arid fluvial environment. Of course, we cannot see thick evaporites in the field in the Chuxiong Basin, even in the Simao Basin and Khorat Basin since that the evaporite is easily dissolved under the tropical monsoon climate in these basins. Previous publication showed that several medium-scale salt mine of the Jiangdihe Formation were found (ref. 77). I am very curious about what the authors stated in the Supplementary Note line 211-2113 "Thus, the presence of thin evaporites is in contradiction with the existence of a low-gradient and continental-scale river system". So, what is exactly the authors' opinion?

The authors acknowledged that all stratigraphic and sedimentary work are from previous publication (in supplementary note). For the section in the Xichang Basin, the stratigraphic column (Supplementary Figure 1) is totally copied from Deng et al. (2018). Deng et al. (2018) state that "the basal member of the Xiaoba formation having 1400 m in thick changes from a fluvial environment at the base to a shallow-lacustrine facies at the top". "The second member of the Xiaoba Formation is composed of ~100 m of lacustrine red sandstone, calcareous siltstone, calcareous mudstone, and limestone (mainly at the top of the section), interbedded with gray-purple silt-stone and calcareous lenses". The third member is over 700 m thick and is primarily composed of calcareous siltstone and calcareous mudstone, interbedded with gray-purple quartz siltstone and gypsum layers, and represents a fluvial and shallow-lacustrine facies. The Paleogene Lei-dashu Formation is over 1300 m thick changes from a fluvial environment at the base to a shallow-lacustrine facies at the top. Therefore, the authors did not do sedimentary work, but they denied the lacustrine interpretation of Deng et al. (2018) even if they copied Deng's section.

Similarly, the authors copy the stratigraphic column of Deng's section in the Chuxiong Basin. However, they wrongly place the basal conglomerates of the Matoushan Formation in Deng's paper as the Jiangdihe Formation in Supplementary Figure 1. Deng et al. (2018) stated that "The Jiangdihe Formation represents a fluvial and shallow-lacustrine facies and The Zhaojiadian Formation fines upwards and represents a shallow-to-marginal lacustrine facies". However, the authors proposed fluvial+lake and floodplain+fluvial environment for the Jiangdihe and Zhaojiadian formation, respectively, without any sedimentary work.

Therefore, I easily found that the authors just simply copied two sections in the Xichang and Chuxiong basins of Deng et al. (2018). Based on my personal opinion and authors' cited publication (ref 16, Deng et al., 2018), I do prefer a fluvial and lacustrine environment during that time. It also indicates that the sedimentary environment fluctuate between fluvial and lacustrine and is not stable in such a long time range as the authors' claimed.

As for the Simao and Khorat basins, no matter the evaporite is marine or non-marine, it is impossible to form such a salt giant in an open environment (Please see Warren, 2016, Evaporites-A Geological Compendium). The authors claimed in the response letter line 417 to 422 that marine incursion evaporites cannot be used as an argument against a fluvial system that drained into the Neo-Tethyan Ocean. I guess the authors argued the seawater may be intruded from the Neo-Tethyan Ocean (from south to north) and thus claimed like that. However, marine incursions were not from the Neo-Tethyan Ocean, but rather from the Meso-Tethys Ocean or proto-Paratethys Sea as the cited publications (in Response letter line 14 to 15) argued. Therefore, the authors used the marine incursion evaporite to support the existence of fluvial system is unreasonable and untenable. Moreover, the lithological and sedimentary features is indicative of a typical saline lake by many publications (Hite and Japakesetr, 1979; El Tabakh et al., 1999; Zhang et al., 2013; Liu et al., 2018; Qin et al., 2020; Wang et al., 2021).

As such, the Xichang, Chuxiong, Simao, and Khorat Basins during the so called late Cretaceous to Early Paleogene is mostly lacustrine. And Simao and Khorat saline lakes were not exorheic.

(3) Provenance analysis

I suggest you put the newly added data of Simao and Khorat in the MDS plot in Supplementary Figure 6. And it is obvious that the Indochina plot so far away from your samples, indicating there was no provenance connection between them. So why do the authors' believe the fluvial system can flow to the Indochina?

(4) Landscape and large river system

Such a long and large river system existed in the eastern Tibetan Plateau maybe need a higher elevation and large erosion, and thus will result in enough clastic materials to transport into the Neo-Tethys Ocean. Together with the authors' claimed that there were no proximal sources, I have three concerns. (1) How can a low-relief landscape with slow exhumation and erosion (slow cooling) supply so many clastic materials to the Neo-Tethys Ocean in such a long distance (at least 3000 km)? (2) If the river do flow into the Neo-Tethys Ocean, do the authors have any evidence of sedimentary sequence in delta and offshore Thailand?

(5) Sections and samples

Authors provide the GPS of all samples, however I found sample distance in the Dujiangyan section is nearly up to 30 km from Google Earth. That is absolutely not the sampling action in the same section because the Dujiangyan section only have ca. 800m in thickness. The same as the Leshan section, two sample distance is up to 21 km. So it is absolutely questionable of the sampling.

The authors acknowledged that all stratigraphic and sedimentary work in the Xichang and Chuxiong basins are from previous publication. And as I mentioned above, the stratigraphic columns in these two basins are totally copied from Deng et al. (2018).

So I doubt that the authors' sampling is consistent with the section description.

Therefore, from the authors' sampling and stratigraphic work, I have to trace back to the age of the section again. All age constraints from this study is the fossils which is done about 40 years ago.

Previous fossil results were from regional mapping at scale of 1:200,000 or type section. The studied sections are obviously not the type section, so previous results should not be applied directly. How can the regional fossils be used in whole basin?

(6) Standard zircons

I commented the dating results of the Standard zircons, but I cannot see correct or appropriate response. Yes, as the authors replied, the detrital zircon dating is a usual way. But for each test, your 91500 and GJ-1 zircons should generate their ages. What I asked is you should provide all the ages of 91500 and GJ-1 during your experiment, so that I can determine your deviation between the test values and recommended ages. And then I will have an idea about the reliability of all ages of the samples. So you cannot use the dating results in Yuan et al.(2004) to tell me that the ages in your study is reliable.

Reviewer #2:
Remarks to the Author:
Greetings,

I reviewed the revision submitted by Huiping Zhang and co-authors. All of my comments were sufficiently addressed. I support publication of this manuscript.

Andrew Laskowski

Reviewer #3:
Remarks to the Author:
Dear Editor and Authors,

Thanks for the detailed response to my previous comments. After reading the manuscript, I thought there are still a few problems that need to be addressed before publication.

First, the authors present a few pictures to prove a continental scale river through the basins in SE Tibet, but I suspect this. As shown in the pictures, the laminated and fine grained sandstones and mudstones are better to be interpreted in lacustrine environment, as suggested by the Reviewer 1. If the sediments were transported by a large river, we should observe some cross beddings, which are scarce in these basins.

I am also disagree with the statement that the Late Cretaceous-Paleogene sediments in Simao Basin is still marine facies. The sediments have been terrestrial since late Jurassic.

Second, I am not expert to modeling, so cannot judge the reliability of the modeling results to support a low relief at sea level before Eocene. But I am curious even there is a continental scale river flow to the Neo-Tethys Ocean, why must be a low relief existed?

Third, in recent studies, Clift et al. (2020) and Zheng et al. (2021) shown that, the Gonjo, Jianchuan, Yuanjiang, and the Northern Vietnam basins have similar provenance in the early Cenozoic, which supporting a southward-flowing river from eastern Tibet to the South China Sea. This is contrast with the figure as suggested in this manuscript. I am wondering how the authors reconcile this inconsistency.

Reviewer #4:

Review of “*Existence of a continental-scale river system in eastern Tibet during the late Cretaceous–early Palaeogene*”

I am assessing this paper on the basis of its integrated regional data, its innovative interpretations, and the potential impact of those interpretations on our understanding of some large-scale geomorphic anomalies along the eastern and SE margin of the Tibetan Plateau. This approach likely stands in contrast to some other reviewers who are more familiar with details of the local geologic-stratigraphic-petrographic-chronologic data sets that this submission exploits and builds upon. The key scientific question that this paper addresses is the origin of the widespread, high-altitude, low-relief surfaces that characterize much of the SE Tibetan Plateau: a region where local relief at present is typically quite high, with deeply incised river gorges, and rock-uplift rates are rapid in a global context. The presence of these long-lived, low-relief surfaces at rather high altitudes has long piqued our curiosity: why are they there; how did they form; what are modern analogues of their formative sequence; what data can be used to test various hypotheses?

The authors argue that, despite some coeval tectonic uplift, a long-lived, ~north-to-south river system in eastern Tibet during Late Cretaceous to Paleogene times created abundant, low-relief surfaces during a time of relative stability (or in the face of ongoing, but slow rock uplift) prior to the Indo-Aisan collision and the main Himalayan orogeny. The authors support this scenario by comparing different data sets from these terranes: contrasts in cooling histories from the proposed river corridor (versus the bounding terranes); contrasts in detrital mineral compositional abundances and U/Pb zircon cooling ages within the “drainage corridor” versus outside of it; mixing models that optimize inputs from diverse source areas in order to “match” the observed age abundances; etc.

The cooling histories of the compiled thermochronological records (Figure 1b) make a rather persuasive case that slow Late Cretaceous to Early Tertiary cooling in Songpan-Garzi and Yidun terranes contrasts markedly with the regions of significantly more rapid Late Cretaceous-Early Tertiary cooling to the east and west of these terranes. Hence, while considerable rock uplift, erosion, and bedrock cooling was going on to the east and west during Late Cretaceous to Paleogene times, this north-south corridor in eastern Tibet appears quite stable. To support their interpretation of an integrated fluvial system draining southward to the Neotethyan ocean, the authors combine paleocurrent analysis with detrital mineralogy and detrital zircon U/Pb cooling ages to show a noteworthy consistency among dated sampling sites spanning ~600 km from north to south along the proposed fluvial corridor. For me, the match between (i) the detrital zircon ages from the Songpan-Ganzi and Yidun terranes (proposed source areas in the north) with (ii) the suite of consistent detrital zircon ages from depositional basins spanning 600-750 km from north to south provides critical support to their hypothesized drainage basin geometry and the proposed timing of its existence as an integrated depositional system. To me, this spatial-temporal consistency is a key factor supporting the interpretation offered by these authors.

I suspect that, for some readers, examination of the extensive supplementary data will be needed to convince them of the validity of the authors’ hypotheses. I find both (1) their multi-dimensional scaling plots of detrital data sets and potential source areas and (2) their modeled relative contributions from potential source areas quite persuasive.

I note that previous reviews brought up many specific issues and questions, commonly related to the characteristics of a given source area and an alternative interpretation. I am not

qualified to judge the merits (or validity) of these objections. But, I did find that this contribution's authors gave quite convincing justifications for their choices and interpretations.

Overall, this provocative, innovative synthesis and interpretation provides a potential resolution to a long-standing problem related to how these low-relief, high-elevation surfaces in SE and Eastern Tibet developed. I believe it is worthwhile to get this data set and interpretation "out there" for the interested audience to contemplate and to try to test with new data or re-analysis. I also think that the "problem" that this paper addresses is a long-standing and puzzling one: a problem that has come into clearer focus in recent decades as (i) high-resolution digital topography has become available of even remote or restricted areas (thereby enabling clear topographic syntheses, comparisons across regions, and identification of "anomalies") and (ii) as high-resolution, low-cost, and high-throughput analytical techniques have enabled thousands of analyses to be made and synthesized, and (iii) as improved and diversified numerical modeling approaches have enabled more rigorous evaluation of hypotheses. This contribution from the edge of Tibet exploits all of these technologies in a creative synthesis that is sure to inspire (and provoke) further research focused on the evolution of large-scale dynamic orogens.

Note to editors/authors: I show my "linguistic/grammatical/clarification" suggestions below in red text.

Note that the Yangtze River is not identified/labelled in any figure that I could find!

"**comprise**" means "to be composed of" So eastern Tibet comprises these provinces, not the other way around.

with sustained ~~topography~~ **topographic relief**

**SUCH** regional differences in erosion/exhumation rates could be explained by a

along the foredeep depozone (i.e., SW Sichuan, Xichang, Huili, and Chuxiong basins: **Fig. 1a**)

139-41 ~~Several meters to tens of meters~~ "Thick, cross-stratified sandstone beds ~~with cross-~~

~~stratification~~ represent channel deposits of southward-flowing, low-**energy** ("energy" or "gradient"?) rivers ~~and associated floodplains, and/or exorheic lakes~~ I don't think that

crevasse splays or lake deposits should be cited as indicative of overall paleocurrent directions for large river systems, especially for thick sandstone deposits. What indicates that these rivers are "low-energy"? Are there complete channel cross sections and longitudinal sections to enable you to deduce "energy" versus gradient or simply associated grain size?

"**Leshan section**" in the Sichuan Basin, (given that the Leshan section is not identified in the figure.

"**genuine**"?? Does this mean "**statistically significant**"?

173-5 "The consistent provenance signal from the different basins **requires** the existence of a continuous fluvial system during the late Cretaceous–early Palaeogene." Does it truly "**require**"? That may well be the most likely scenario, but it doesn't "require" this scenario, in my opinion.

367 showing main tectonic units "**(red text)**" (in reference to what represents these units in the figure) and major river systems (except for the Yangtze! **Add a label for this river!**)

370 low-relief plateau areas^{24,31–34,36,58}. New suggested text: **Hexagons indicate sites with Cretaceous-Tertiary cooling histories (shown in 1B)** Dashed

Comments on Figures

Figure 1a. Nowhere is the Yangtse River labeled on this figure. Its name should be clearly identified. Note that in the figure caption, no description is given of the light blue vertical band from 40-50 Ma in Fig 1b. What is that? I presume it's a proposed "boundary" between rapid L Cret-Paleocene cooling versus post-50-Ma rapid cooling. Why not add blue or red labels for rapid cooling pre-50 Ma and post-40 Ma, respectively? I also note that the chosen level of transparency of some of the "yellow" low-relief surfaces makes these surfaces appear to be a different color (more orange than yellow, where superimposed above the orange swath), and that there is a strange (inconsistent?) mix of yellow and orange surfaces just above the red "5" in the figure. These issues should all be readily corrected. Could a label be added to the orange region so that it's more self-explanatory? Similarly, a label on the blue-dashed line ("hypothesized drainage divide") would make its significance more obvious.

Figure 1b. Add a legend indicating blue lines for rapid cooling prior to 50 Ma, versus red lines for rapid cooling since 50 Ma.

Figure 2. How about helping your readers along with a title box, "Drainage Scenarios" and cryptic summary titles for each scenario's panel?

Figure 3. The rationale is unstated for the red and blue lines for the probability density functions. Please make that clear!

Comments on Supplemental Figures

Supp Figure 3. Illustrative and quite compelling figure!! Spell out the names of the sections (CX, HL, etc) for each group of samples.

**Response letter**

In this letter we will provide our detailed response (in blue text) to the comments of the editor
and the four reviewers (in black) and explain all changes performed on the manuscript.

**Response to the comments of the four reviewers**

**Reviewer #1**

The authors had a positive feedback to most reviews. However, I still see the insufficient
interpretation or ignored reviews that I and editor gave.

(1) The authors give four lines of additional evidence to support the river route to the
Neo-Tethys Ocean. However, unfortunately, some evidence was incorrectly used by the
authors. Firstly, the marine evaporites in the Simao-Khorat Basins reach a consensus, but
discharge model in the cited paper is totally misunderstood by the authors. Recent
publications (in cited papers) proposed that seawater incursions were from the Meso-Tethys
Ocean or proto-Paratethys Sea which came from northern part of the Simao-Khorat Basins,
not from the southern Neo-Tethys Ocean.

**Reply:** As the reviewer#1 points out correctly, the marine origin of Late Cretaceous–early
Palaeogene evaporite in the Simao-Khorat Basins has reached a consensus. This clearly
indicates that, as we have emphasized in the main text, these areas were close to sea level at
that time. Although Wang et al., (2021) has inferred that the seawater likely originated from
the proto-Paratethys Sea to the northwest, these authors also proposed an alternative
hypothesis that marine incursions were from the Neo-Tethyan Ocean to the southwest (Wang
et al., 2015). In addition, the proto-Paratethys Sea-derived model is subject to debate, because
it cannot explain the present perception that the Cretaceous marine strata were disrupted
between the Qiangtang and the Simao-Khorat Basins (Qin et al., 2020). About the model of
“Meso-Tethys Ocean-derived” proposed by Qin et al. (2020), we infer this just reflects
ambiguity about the use of this name for “Tethys Ocean” because it is generally believed that
the Meso-Tethys Ocean had closed before the late Cretaceous (see Kapp et al., 2019, Li et al.,
2019 for reviews; as shown the Figure 38 in Metcalfe, 2021). Overall, we originally showed
the existence of marine evaporates in the Simao-Khorat Basins through the two studies (Qin
et al., 2020; Wang et al., 2021).

About the source of paleo-seawater, Qu (1997) proposed that paleo-seawater recharged
from the south to the north based on the similar stratigraphic ages, sedimentary features,
mineral sequences, and solute sources of the evaporites between the Simao and Khorat Basin.
Moreover, the thickness of the late Cretaceous evaporate formation in the Khorat Basin is
apparently thicker than that in the Simao basin (e.g., Zhang et al., 2018, Yan et al., 2021a, Yan
et al., 2021b). These observations, coupled with the reconstructed block-tectonic pattern
during the late Cretaceous–early Paleogene (e.g., Boucot et al., 2009; Poblete et al., 2021;
Metcalfe, 2021), it is proposed that that the seawater originated from the Neo-Tethys Ocean to
the south or southwest during sea intrusion episodes (Qu, 1997; El Tabakh et al., 1999; Wang
et al., 2015; Rattana et al., 2021). We have clarified and improved our discussion of this issue
in the revised manuscript (see lines 233–243).

**References:**

- Boucot A J, Chen X, Scotese C R, Fan J X. 2009. Phanerozoic Paleoclimate: An Atlas of Lithologic
Indicators of Climate. Beijing: Science Press. 173.
El Tabakh, M., Utha-Aroon, C. & Schreiber, B.C. Sedimentology of the Cretaceous Maha Sarakham
evaporites in the Khorat Plateau of northeastern Thailand. *Sediment. Geol.* **123**, 31–62 (1999).
Metcalfe, I. Multiple Tethyan ocean basins and orogenic belts in Asia. *Gondwana Research*, **in press**,
(2021).
Li, S., Yin, C., Guilmette, C., Ding, L., & Zhang, J. Birth and demise of the Bangong-Nujiang Tethyan

- ocean: a review from the Gerze area of central Tibet. *Earth-Science Reviews*, **198**, 102907 (2019).
- Kapp, P., Decelles, P. G. Mesozoic-Cenozoic geological evolution of the Himalayan-Tibetan orogen and working tectonic hypotheses. *American Journal of Science*, **319**, 159–254 (2019).
- Poblete, F., Dupont-Nivet, G., Licht, A., Hinsbergen, D., & Baatsen, M. Towards interactive global paleogeographic maps, new reconstructions at 60, 40 and 20 Ma. *Earth-Science Reviews*, **214**, 103508 (2021).
- Qin, Z., Li, Q., Zhang, X. G., Fan, Q. S., Wang, J. P., Du, Y. S., Ma, Y. Q., Wei, H. C., Yuan, Q. & Shan, F. S. Origin and recharge model of the Late Cretaceous evaporites in the Khorat Plateau. *Ore Geol Rev.* **116**, 103226 (2020).
- Qu, Y.H. On affinity of potassium bearing brine in Lanping-Simao basin, China to that in Ale basin, Thailand, and location of target areas for potassium hunting in former basin. *Geol. Chem. Miner.* **19** (2), 81–85 (1997) (in Chinese with English abstract).
- Rattana, P., Choowong, M., He, M.-Y., Tan, L., Lan, J., Bissen, R., Chawchai, S., Geochemistry of evaporitic deposits from the Cenomanian (Upper Cretaceous) Maha Sarakham Formation in the Khorat Basin, northeastern Thailand. *Cretaceous Research*, in press (2021)
- Wang, L. C., Zhong, Y. S., Xi, D. P., Hu, J. F. & Ding, L. The Middle to Late Cretaceous marine incursion of the Proto-Paratethys sea and Asian aridification: A case study from the Simao-Khorat salt giant, southeast Asia. *Palaeogeogr Palaeoclimatol Palaeoecol.* **567**, 110300 (2021).
- Wang, L. C., Liu, C, Fei, M, Shen, L, Zhang, H, Zhao, Y. First shrimp U-Pb zircon ages of the potash-bearing Mengyejing Formation, Simao Basin, southwestern Yunnan, China. *Cretac Res.* **52**, 238–250 (2015).
- Yan, M. D., Zhang, D. W., Fang, X. M., Zhang, W. L., Song, C. H et al. New Insights on the age of the Mengyejing formation in the Simao basin, SE Tethyan domain and its geological implications. *Sci China Earth Sci.* **64**, 231–252 (2021a).
- Yan, M. D., Zhang, D. W., Li, M. H. New Progresses on the Potash Deposits in the Simao and Khorat Basins: A Synthesis. *Earth Science Frontier*, in press, (2021b).
- Zhang D., Yan M., Fang X., Yang Y., Zhang T., Zan J., Zhang W., Liu, C & Yang Q. Magnetostratigraphic study of the potash-bearing strata from drilling core ZK2893 in the Sakon Nakhon Basin, eastern Khorat Plateau. *Palaeogeogr Palaeoclimatol Palaeoecol.* **489**, 40–51 (2018).

(2) Moreover, as the authors stated that the Khorat Basin was the last stop to the Neo-Tethyan Ocean, it is absolutely impossible to form such a salt giant in the Khorat Plateau Basins in an open water mass. The Khorat Plateau Basins cover an area of ca. 247,000 km² with evaporite thickness up to 1000 m (Hite and Japakasetr, 1979, Economic Geology). Another evidence to deny the river court is that all publications and geological observations tell us the evaporite-bearing Maha Sarakham Formation in the Khorat Basin was lacustrine environment. So when we considered the direction of marine incursion and salt giant, it would be reasonable why the Simao-Khorat are endorheic lakes.

Reply: We respectfully disagree with the referee on this as they appear to have misinterpreted Hite and Japakasetr (1979). Hite and Japakasetr, (1979) definitely stated that “*The potash deposits of the Khorat Plateau are in the Maha Sarakham Formation of Cretaceous age. This formation is present only on the Khorat Plateau. North, in the Sakon Nakhon Basin, the formation extends over an area of about 21,000 km². South, in the Khorat Basin, the formation covers a slightly larger area of about 36,000 km². The maximum thickness of the formation in either basin is unknown, but it could exceed 1,000 m*”. This sentence just shows that the thickness of the Maha Sarakham Formation is up to 1000 m, and does not mention the thickness of evaporate deposits. In fact, the evaporite thickness from the late Cretaceous–early Paleogene Maha Sarakham Formation in most areas of the Khorat Plateau is only tens to hundred of meters according to a collection of bore documents (see Hite and Japakasetr, 1979; Rattana et al., 2021, in press).

In terms of the formation mechanism of evaporites in the Khorat Plateau Basins, in addition to hydrological conditions, the coupling of regional aridity and global

high-temperature events, along with eustatic change would have further promoted the
formation of potash salts (Liu et al., 2018; Yan et al., 2021). According to previous studies, if
this sea is located in a low-latitude setting and especially if it straddles the belt of
high-pressure dry air, evaporation can result in the formation of a major evaporite deposits in
continental margin/shelf settings (e.g., narrow seas, tidal flats, lagoons) (Reynolds and Johnson,
2019; Miall, 2016; as shown the figure below in p. 240 of Reynolds and Johnson, 2019). Thus,
it is likely that major salt-bearing sub-basins / depressions in the Khorat Plateau represent
several lagoons or tidal flats during low sea level episodes, and the large-scale river system
may flow to the sea near these tidal flats or lagoons (resembling landscape referred to the
figure in p. 76-77 of Reynolds and Johnson, 2019; modified Fig. 4b).

We have never denied the occurrence of lacustrine environments. Rather, several lines of
evidence below clearly corroborate a co-existence of fluvial and/or delta-plain environments
in the Khorat basins during the late Cretaceous–early Paleogene: (i) we note that “*The*
*non-marine red beds interbedded with the evaporites are fluvial or alluvial deposits and*
*include displacive anhydrite nodules and beds and displacive halite in cubic forms*”; and
“*Siliciclastics from the Maha Sarakham Formation in the Khorat Basin are composed of*
*alternating cross-bedded siltstones and massive mudstones of fluvial origin, suggesting fluvial*
*deposition in the basin province*” cited from El Tabakh et al., (1999, p. 54, 59). The two
sentences from El Tabakh et al., (1999) support the existence of a fluvial depositional
environment for upper Cretaceous–lower Paleogene strata in the Khorat basins, even though
lacustrine environments may be more extensive. Similarly, the late Cretaceous Maha
Sarakham Formation in northeastern Thailand consists of red to reddish-brown, fine-grained,
laminated and small-scale cross-bedded sandstones interbedded with siltstones and mudstones
with disseminated salts and gypsum (Meesook, 2000), which was interpreted as a meandering
river system, even if evaporitic conditions also prevailed (Meesook, 2000). (ii) Sedimentary
structures in mudstones such as mudcracks, chaotic textures, root structures and burrows
suggest a subaerial setting (El Tabakh et al., 1999). (iii) A widespread and dominantly
south-directed paleocurrent from the alternating thin mudstone and sandstone beds of the late
Cretaceous units in the Khorat basins provides additional physical evidence corroborating the
existence of a south-flowing sediment routing system (Heggemann et al., 1992; Singsoupho et
al., 2015). (iv) Isotope measurement results of potash deposits suggest that a terrestrial input
or riverine water had entered into the Khorat Plateau when the potash deposits were
precipitating (El Tabakh et al., 1999; Zhang et al., 2013), which likely has a close relationship
with the influence of fluvial influx (Zhang et al., 2013). Of course, our interpretation of a
hydrologic connection between the Khorat Basin and the Neo-Tethyan Ocean does not
preclude a more complex drainage configuration, involving the assembly of multiple channels
along a near south-to-north major route. More detailed provenance analyses of Upper
Cretaceous samples from the Khorat basin may further test this hypothesis to constrain a more
complete drainage configuration of the proposed large river. We have included part of this
evidence in the revised manuscript (lines 228–233).

From the perspective of provenance data, the late Cretaceous–early Paleogene deposits
in the Simao Basin and Muang Xai Basin to the south have widely been proposed to be
mainly from the Songpan-Ganzi Yidun, and Qiangtang terranes, rather than proximal source
areas (e.g., Indosinian terrane) (Wang et al., 2014; Wang et al., 2017; Chen et al., 2017; Yan et
al., 2021). In other words, if the Simao and Muang Xai Basins were endorheic basins during
the late Cretaceous–early Paleogene as suggested by reviewer #1, K₂–E₁ sediments in the two

basins must be derived from surrounding older rocks, and we would expect the widespread
Triassic metamorphic rocks surrounding these basins to provide a major source of detrital
material and a unique prominent peak at 200–250 Ma (Fig. S4). However, previously
published detrital zircon data from late Cretaceous–early Paleogene samples of the two basins
show strikingly consistent peaks at 200–300 Ma, 390–480 Ma, 700–900 Ma, 1700–2000 Ma,
and 2300–2600 Ma (Wang et al., 2014; Wang et al., 2017; Chen et al., 2017; Yan et al., 2021;
Fig. S8). This strongly suggests the Simao and Muang Xai Basins were not endorheic basins
in the late Cretaceous–early Paleogene, but were connected to a transcontinental drainage
system that linked central–eastern Tibet with the two basins (Wang et al., 2014; Wang et al.,
2017; Chen et al., 2017; Yan et al., 2021). Although no detrital zircon data from K₂–E₁
sediments in the Khorat basin have been reported to date, a dominantly south-directed
paleocurrent from the sandstone beds in the late Cretaceous Maha Sarakham Formation in the
Khorat Basin provides additional physical evidence for a south-flowing sediment transport
pathway (Heggemann et al., 1992; Singsoupho et al., 2015).

Based on the collective weight of stratigraphic and sedimentologic observations
described above, we contend that the deposits in the Khorat Basin do not preclude our
hypothesis that a continental-scale river system discharged into the the Neo-Tethyan Ocean
near what is now the Khorat Basin (see Figure 4), even though the major paleo-channels
cannot be traced due to post-depositional modification and later erosion.

**References:**

- Chen, Y., Yan, M., Fang, X., Song, C., Zhang, W., Zan, J. et al. Detrital zircon U-Pb geochronological
and sedimentological study of the Simao Basin, Yunnan: Implications for the early Cenozoic
evolution of the Red River. *Earth Planet. Sci. Lett.* **476**, 22–33 (2017).
Heggemann, H., Helmcke, D., & Tietze, K. W. Sedimentary evolution of the Mesozoic Khorat basin in
Thailand. *Zentralblatt für Geologie und Paläontologie*, 11–12, 1267–1285 (1992).
Hite, R.J. & Japakasetr, T. Potash deposits of Khorat Plateau, Thailand and Laos. *Econ. Geol.* **74**, 448–
458 (1979).
Liu, C.L., Wang, L.C., Yan, M.D., Zhao, Y.J., Cao, Y.T., Fang, X.M., Shen, L.J., Wu, C.H., Lv, F.L. &
Ding, T. The Mesozoic–Cenozoic tectonic settings, paleogeography and evaporitic sedimentation of
Tethyan blocks within China: implications for potash formation. *Ore Geol. Rev.* **102**, 406–425
(2018).
Meesook, A. Cretaceous environments of northeastern Thailand. *Developments in Palaeontology &*
*Stratigraphy*, **17** (2000).
Miall, A. D. (2016). *Facies Models*. Switzerland: Springer International Publishing, p206.
Wang, L. C., Liu, C., Gao, X. & Zhang H. Provenance and paleogeography of the Late Cretaceous
Mengyejing Formation, Simao Basin, southeastern Tibetan Plateau: Whole-rock geochemistry,
U-Pb geochronology, and Hf isotopic constraints. *Sedimentary Geol.* **304**, 44–58 (2014).
Wang, Y. L., Wang, L. C., Wei, Y. S., Shen, L. J., Chen, K., Yu, X. C. & Liu, C. L. Provenance and
paleogeography of the Mesozoic strata in the Muang Xai basin, northern Laos: petrology,
whole-rock geochemistry, and U–Pb geochronology constraints. *Int J Earth Sci.* **106**, 1409–1427
(2017).
Rattana, P., Choowong, M., He, M.-Y., Tan, L., Lan, J., Bissen, R., Chawchai, S., Geochemistry of
evaporitic deposits from the Cenomanian (Upper Cretaceous) Maha Sarakham Formation in the
Khorat Basin, northeastern Thailand. *Cretaceous Research*, in press (2021).
Reynolds and Johnson (2019). *Exploring earth science*. New York: McGraw-Hill Education,
p.76-77, 240.
Singsoupho S, Bhongsuwan T, Elming S Å. Palaeocurrent direction estimated in Mesozoic redbeds of
the Khorat Plateau, Lao PDR, Indochina Block using anisotropy of magnetic susceptibility.
*Journal of Asian Earth Sciences*, **106**, 1–18 (2015).

Yan, M. D., Zhang, D. W., Fang, X. M., Zhang, W. L., Song, C. H et al. New Insights on the age of the
Mengyejing formation in the Simao basin, SE Tethyan domain and its geological implications.
*Sci China Earth Sci.* **64**, 231–252 (2021).

Zhang, X.Y., Ma, H.Z., Ma, Y.Q., Tang, Q.L. & Yuan, X.L. Origin of the late Cretaceous
potash-bearing evaporites in the Vientiane Basin of Laos: $\delta^{11}\text{B}$ evidence from borates. *J. Asian*
*Earth Sci.* **62**, 812–818 (2013).

(3) Secondly, in Figure S8 (Supplementary Figure 8), the authors cited the data of Carter and
Moss (1999, Geology) and show the age distribution of the Khorat Basin from so-called late
Late Cretaceous and early Paleogene. However, I checked the paper of Carter and Moss (1999)
and found the youngest strata in their study is the Early Cretaceous Khok Khruat Formation.
There was totally no any data of the late Cretaceous to early Paleocene. Based on these two
evidence, the river court don't convince me.

**Reply:** We thank the reviewer for catching this oversight regarding the age of the Khorat
Group (Carter and Moss, 1999). We have removed this from our compilation of detrital zircon
in a modified Fig. S8. Fortunately, this does not impact our interpretations.

(4) The age and depositional environment cannot convince me in the revised manuscript and
that is what I am concerned about. Why the age is so important and I repeatedly stress it?
K2-E1 is a large time range from 100 Ma to 56 Ma and you mentioned that it was a long-term
tectonic stability. In this region, at least two tectonic events identified by many geologists: 1,
the mid-Cretaceous (ca. 100 Ma) tectonic event (e.g., Lovatt-Smith et al., 1996) caused by
collision between Qiangtang and Lhasa; 2, the India-Asia collision at ca. 65 Ma. The first
event caused unconformity in the studied basins. It's hard to imagine the existence of such a
long-term tectonic stable environment.

**Reply:** We must emphasize that it has been widely appreciated that the final collision between
the Qiangtang and Lhasa terranes occurred in the Early Cretaceous (~130–120 Ma) (e.g.,
Kapp et al., 2005; Li et al., 2019; also see Figure 7(G), 7 (H) in Kapp et al., 2019). Moreover,
Cretaceous shortening, basin deformation, and magmatism activities mainly occurred in the
central Tibetan plateau due to northward underthrusting of the Lhasa terrane beneath the
Qiangtang terrane along the Bangong suture during low-angle subduction of Neo-Tethys
oceanic lithosphere (e.g., Kapp et al., 2005, 2019; Rohrmann et al., 2012; Li et al., 2019). This
region is well to the west of our study area. With respect to eastern Tibet, there is no clear
evidence for an obvious erosion surface between the early and late Cretaceous horizons in the
studied basins (~100 Ma), and contacts are likely conformable. More importantly, as reviewer
#1 acknowledged, we proposed a prolonged tectonic stability in eastern Tibet from late
Cretaceous to early Paleogene (<100 Ma–50 Ma). Thus, even if there was a transient tectonic
event in the mid-Cretaceous (>100 Ma), this would be fully compatible with our
interpretation and conclusions.

Although discussion still persists, a growing number studies have consistently agreed
that the initial timing of the India-Asia collision (or collision of India with an intra-Tethyan
island arc) is between ~60 Ma and 50 Ma (e.g., Tapponnier et al., 2001; Najman et al., 2010;
DeCelles et al., 2014; Martin et al., 2020; An et al., 2021; Yuan et al., 2021). But please note
that the convergence rate began to rapidly decrease after the onset of the India-Asia collision
(Copley et al., 2010; van Hinsbergen et al., 2011), thus the main crustal thickening of a large
part of Tibet occurred later (e.g., Tapponnier et al., 2001; Cao et al., 2020). For example, the
strong south-north compression in the central TP and the continuous north-ward subduction of
the Indian lithosphere mainly occurred between 50 and 34 Ma (Kapp et al., 2019; Lin et al.,
2020; Wang et al., 2008; Wang et al., 2014). Back to our study, the large-scale tectonic

deformation of eastern Tibet occurred during the post-collision of India with Asia. Based on
low-temperature thermochronology (e.g., Zhang et al., 2016; Wang et al., 2012),
paleoaltimetry (e.g., Hoke et al., 2014; Xiong et al., 2020), and recent paleomagnetism studies
(Tong et al., 2017; Zhang et al., 2020), significant tectonic deformation and uplift on the
eastern margin of the Tibetan plateau and the western margin of the South China block have
widely been acknowledged to occur later than the middle Eocene. In addition, the widespread
magmatic activities at ~40–30 Ma in eastern Tibet also suggest a diachronous uplift history
for the Tibetan plateau (Chung et al., 1998). Thus, the initial India-Asia collision at ca. 60–50
254 Ma did not immediately affect the region of eastern Tibet studied here.

**References:**

- An, W., Hu, X., Garzanti, E., Wang, J. G., & Liu, Q. New precise dating of the India-Asia collision in
the Tibetan Himalaya at 61 Ma. *Geophysical Research Letters*, **48**, e2020GL090641 (2021).
- Cao, K., Leloup, P. H., Wang, G. C., Liu, W., Mahéo, G., Shen T. Y., Xu, Y.D., Sorrel, P. & Zhang, K.
X. Thrusting, exhumation, and basin fill on the western margin of the South China block during
the India-Asia collision. *Geol. Soc. Am. Bull.* **133**, 74–90 (2020).
- Chung, S., Lo, C. H., Lee, T. Y., Zhang, Y. Q., Xie, Y. W., Li, X. H., et al. Diachronous uplift of the
Tibetan plateau starting 40 Myr ago. *Nature*, **394**, 769–773 (1998).
- Copley, A., Avouac, J. P., & Royer, J. Y. India-Asia collision and the Cenozoic slow-down of the
Indian plate: Implications for the forces driving plate motions. *Journal of Geophysical Research*,
**115**, B03410 (2010).
- DeCelles, P., Kapp, P., Gehrels, G., & Ding, L. Paleocene-Eocene foreland basin evolution in the
Himalaya of southern Tibet and Nepal: Implications for the age of initial India-Asia collision.
*Tectonics*, **33**, 824–849 (2014).
- Hoke, G. D., Liu-Zeng, J., Hren, M. T., et al. Stable isotopes reveal high southeast Tibetan Plateau
margin since the Palaeogene. *Earth and Planetary Science Letters*, **394**, 270–278 (2014).
- Kapp, P., Yin, A., Harrison, T. M., & Ding, L. Cretaceous-Tertiary shortening, basin development, and
volcanism in Central Tibet. *Geological Society of America Bulletin*, **117**(7–8), 865–878 (2005).
- Kapp, P., Decelles, P. G. Mesozoic-Cenozoic geological evolution of the Himalayan-Tibetan orogen
and working tectonic hypotheses. *American Journal of Science*, **319**, 159–254 (2019).
- Li, S., Yin, C., Guilmette, C., Ding, L., & Zhang, J. Birth and demise of the Bangong-Nujiang Tethyan
ocean: a review from the Gerze area of central Tibet. *Earth-Science Reviews*, **198**, 102907 (2019).
- Lin, J., Dai, J. G., Zhuang, G. S., Jia, G. D., Zhang, L. M., Ning, Z., et al. Late Eocene–Oligocene high
relief paleotopography in the north central Tibetan Plateau: Insights from detrital zircon U–Pb
geochronology and leaf wax hydrogen isotope studies. *Tectonics*, **39**, e2019TC005815(2020).
- Martin, C. R., Jagoutz, O., Upadhyay, R., Royden, L. H., Eddy, M. P., Bailey, E., et al. Paleocene
latitude of the Kohistan–Ladakh arc indicates multistage India–Eurasia collision. *PNAS*,
11729487–29494 (2020).
- Najman, Y., Appel, E., Boudagher-Fadel, M., Bown, P., Carter, A., Garzanti, E., et al. Timing of
India-Asia collision: Geological, biostratigraphic, and paleomagnetic constraints. *Journal of*
*Geophysical Research*, **115**, B12416 (2010).
- Rohrmann, A., Kapp, P., Carrapa, B., Reiners, P. W., Guynn, J., Ding, L. & Heizler, M.
Thermochronologic evidence for plateau formation in central Tibet by 45 Ma. *Geology* **40**, 187–
190 (2012).
- Tapponnier, P., Zhiqin, X., Roger, F., Meyer, B., Arnaud, N., Wittlinger, G., & Jingsui, Y. Oblique
stepwise rise and growth of the Tibet plateau. *Science*, **294**, 1671–1677 (2001).
- Tong, Y., Yang, Z., Mao, C., Pei, J., Pu, Z., Xu, Y., 2017. Paleomagnetism of Eocene red-beds in the
eastern part of the Qiangtang Terrane and its implications for uplift and southward crustal
extrusion in the southeastern edge of the Tibetan Plateau. *Earth Planet. Sci. Lett.* **475**, 1–14.
- Van Hinsbergen, D. J. J., Steinberger, B., Doubrovine, P. V., & Gassmoller, R. Acceleration and
deceleration of India-Asia convergence since the Cretaceous: Roles of mantle plumes and
continental collision. *Journal of Geophysical Research*, **116**, B06101 (2011).
- Wang, E., Meng, K., Su, Z., Meng, Q., Chu, J. J., Chen, Z., et al. Block rotation: Tectonic response of
the Sichuan basin to the southeastward growth of the Tibetan plateau along the Xianshuihe-

Xiaojiang fault. *Tectonics*, **33**, 686–718 (2014).
Wang, C. S., Zhao, X. X., Liu, Z. F., Lippert, P. C., Graham, S. A., Coe, R. S., et al. Constraints on the
early uplift history of the Tibetan Plateau. *Proceedings of the National Academy of Sciences*, **105**,
4987–4992 (2008).
Xiong, Z. Y., Ding, L., Spicer, R. A., et al. The early Eocene rise of the Gonjo Basin, SE Tibet: From
low desert to high forest. *Earth and Planetary Science Letters*, **543**, 116312 (2020).
Yuan, J., Yang, Z., Deng, C., Krijgsman, W., Hu, X., Li, S., et al. Rapid drift of the Tethyan Himalaya
terrane before two-stage India-Asia collision. *National Science Review*, **8**, 173 (2021).
Zhang, H. P., Oskin, M. E., Liu-Zeng, J., Zhang, P. Z., Reiners, P. W. & Xiao, P. Pulsed exhumation of
interior eastern Tibet: implications for relief generation mechanisms and the origin of
high-elevation planation surfaces. *Earth Planet. Sci. Lett.* **449**, 176–185 (2016).
Zhang, W., Fang, X., Zhang, T., Song, C., & Yan, M. Eocene rotation of the northeastern central
Tibetan Plateau indicating stepwise compressions and eastward extrusions. *Geophysical Research*
*Letters*, **47**, e2020GL088989 (2020).

(5) Of course, it's hard to do the geochronology work in thick red bed basins. But in my
opinion, the charophyta and ostracod fossils would give wrong age constraints compared with
the U-Pb ages. This is true in the Jianchuan Basin and Simao Basin (See Gourbet et al., 2017;
Yan et al., 2021). So age issue is the first priority, or it would be another story.

**Reply:** The Jianchuan Basin is a middle-late Eocene intracontinental basin in eastern Tibet,
whose basin property, depositional age, lithologic and facies features, and subsidence
mechanism are totally different from those of the basins we study (Gourbet et al., 2017;
Zheng et al., 2020; Feng et al., 2021). More significantly, please note that a dispute regarding
age scheme of the Cenozoic sedimentary sequences in the Jianchuan Basin stems from
absolute ages of the Shuanghe, Jianchuan, and Jiuziyuan Formations. Previously estimated
ages of the three formations were mainly based on plant fossils, bivalves, gastropods, and
nannofossils, rather than charophyte, or ostracod. Instead, ages of the underlying Paleocene
Yunlong and Eocene Baoxiangsi Formations in the Jianchuan Basin were initially estimated
by characteristic charophyte, and ostracod assemblages, which is well in agreement with
recent magnetostratigraphic and radiochronologic dating results (Fang et al., 2021). Thus,
taking an example of the Jianchuan Basin to question the age scheme presented this study is
untenable.

The Mengyejing Formation in the Simao Basin has been dated at ~112–63 Ma using
magnetostratigraphic method (Yan et al., 2021), largely in agreement with the previously
reported palaeontological results (i.e., Ostracod, Charophyte fossils, and Sporopollen) (as
shown the Figure 10 of Yan et al., 2021). Similarly, the Guankou and Mingshan Formations
of the Shiyang section in the southwestern Sichuan Basin has been dated at ~84–43 Ma using
the magnetostratigraphic method (Shen et al., 2018, Master thesis), which is also largely
consistent with the biostratigraphic age of these deposits. Taken together, we argue that the
charophyta and ostracod assemblages provide reasonable information on the depositional age
of our studied sedimentary sections, especially given that most geochronology methods are
impracticable for dating these deposits, as acknowledged by reviewer #1.

**References:**

Fang, X. M., Yan, M. D., Zhang, W. L., et al. Paleogeography control of Indian monsoon
intensification and expansion at 41 Ma. *Science Bulletin*, in press (2021).
Gourbet, L., Leloup, P.H., Paquette, J., et al. Reappraisal of the Jianchuan Cenozoic basin stratigraphy
and its implications on the SE Tibetan Plateau evolution. *Tectonophysics* **700**, 162–179 (2017).
Shen, Q. Cretaceous–Cenozoic magnetostratigraphy study in southwestern Sichuan Basin and its

geological significance. Master thesis in the Northwest University (2018).
Yan, M. D., Zhang, D. W., Fang, X. M., Zhang, W. L., Song, C. H et al. New Insights on the age of the
Mengyejing formation in the Simao basin, SE Tethyan domain and its geological implications.
*Sci China Earth Sci.* **64**, 231–252 (2021).
Zheng, H. B., Clift, P. D., He, M. Y., et al. Formation of the First Bend in the late Eocene gave birth to
the modern Yangtze River, China. *Geology* **49**, 35–39 (2020).

(6) About the MDA, the reason that you got so few youngest zircons is inadequate zircon tests.
Most samples for the U-Pb analysis in the study is n=100/150. The ‘large-n’ datasets (e.g., >>
analyses per sample; Pullen et al., 2014; Daniels et al., 2018; Sundell et al., 2019a, 2019b)
in order to increase the probability of analyzing young grains should be used.

**Reply:** As we mentioned in our Response letter from the first round of reviews, younger (i.e.
late Cretaceous–early Paleogene) zircons are completely lacking in our samples from the
Sichuan, Xichang, and Huili Basins. Thus, the maximum depositional age (MDA) is
insufficient to constrain depositional ages with certainty for these basins. Although the
youngest single-grain ages in samples from the Chuxiong Basin are too few to be a robust
indicator of the true depositional age, they are consistent with the late Cretaceous–early
Cenozoic biostratigraphic age of these deposits. Detrital zircon “large-n” (n ≈ 1000) results
from samples are generally used to assess dissimilarities amongst their age spectra during
provenance analysis (Ibañez-Mejia et al., 2018), but samples from our study consistently
show similarities (five age peaks, see Figs. 3 and S4). Moreover, we note that only one young
zircon grain was observed from individual samples in the Chuxiong Basin. Thus, even if
“large-n” is applied, the MDA calculated from several young zircons cannot reflect the exact
age of each sampled horizon because of the unknown lag time between erosion and
sedimentation (especially given that Cretaceous zircons likely experienced long-distance
transport from the Lhasa terrane). Given this, we contend that the number of samples in our
study area, in fact, sufficient to support the conclusions

**References:**

Ibañez-Mejia, M., Alex Pullen., Martin Pepper., Franco Urbani., Gourab Ghoshal., Juan C. Use and
abuse of detrital zircon U-Pb geochronology—A case from the Río Orinoco delta, eastern Venezuela.
*Geology* **46** (11), 1019–1022 (2018).

(7) The author’s reply is actually not convinced me. We all know that the meandering river
system typically consists of lag and sand bar deposits in the bottom and flood plain deposits
showing an upward fining sequence. However, from your supplementary field photo Figure
R6a and stratigraphic column Supplementary Figure 1, I cannot tell these features in the
Xichang Basin. Moreover, I don’t think a meandering environment that consist of a thickness
of about 4,000 m of sandstone and claystones (Supplementary Figure 1) in the Xichang Basin.
The authors acknowledged that all stratigraphic and sedimentary work are from previous
publication (in supplementary note). For the section in the Xichang Basin, the stratigraphic
column (Supplementary Figure 1) is totally copied from Deng et al. (2018). Deng et al. (2018)
state that “the basal member of the Xiaoba formation having 1400 m in thick changes from a
fluvial environment at the base to a shallow-lacustrine facies at the top”. “The second member
of the Xiaoba Formation is composed of ~100 m of lacustrine red sandstone, calcareous
siltstone, calcareous mudstone, and limestone (mainly at the top of the section), interbedded
with gray purple siltstone and calcareous lenses”. The third member is over 700 m thick and
is primarily composed of calcareous siltstone and calcareous mudstone, interbedded with
gray-purple quartz siltstone and gypsum layers, and represents a fluvial and

shallow-lacustrine facies. The Paleogene Leidashu Formation is over 1300 m thick changes
from a fluvial environment at the base to a shallow-lacustrine facies at the top. Therefore, the
authors did not do sedimentary work, but they denied the lacustrine interpretation of Deng et
al. (2018) even if they copied Deng's section.

**Reply:** As acknowledged by the reviewer #1, fluvial environments do exist in the late
Cretaceous Xiaoba Formation and the early Paleogene Leidashu Formation, even if lacustrine
facies is also common. Meanwhile, we aim to emphasize the occurrence of fluvial
environment in these deposits, thus lacustrine/lake interpretation was clearly not expressed in
the earlier Supplementary Note. This has been clarified in the revised manuscript, and please
noted that we have never denied the lacustrine interpretation of Deng et al. (2018), the
existence of (exorheic) lake environment was mentioned in Supplementary Figure 1 and main
text of the earlier manuscript.

More importantly, the sedimentary facies results in Deng et al., (2018) were also taken
from previous province-scale or regional-scale geology reports, and they only provided a
typical section (i.e., Mishi section) in the Xichang Basin. In the submitted Supplementary
Note, we collected more sedimentological descriptions and depositional environment
interpretations from more detailed geological mapping surveys of scale 1:50000 (Sichuan
Bureau of Geology and Mineral Resources, 1990), and more recent geological mapping
surveys of scale 1:250000 (Sichuan Geological Survey Institute, 2013). This literature clearly
indicates that the late Cretaceous Xiaoba Formation and early Paleogene Leidashu Formation
are dominated by alternating fluvial and lacustrine environments, based on detailed
stratigraphic and sedimentological analysis (see revised Supplementary Note for details).
Typical fluvial-lacustrine sequences of the Xiaoba Formation and Leidashu Formation in the
Xichang Basin were presented in geological mapping survey report of scale 1:250000 (see
Figures 1-56 and 1-57 in the Sichuan Geological Survey Institute, 2013). Overall, the
fluvial-lacustrine facies association in the basin interpreted from the literature most likely
records a hydrologically open lake or exorheic lake according to Carroll and Bohacs, (1999)
and Bohacs et al., (2000).

The identification of ancient large-scale drainage systems should be fully related to the
understanding of the depositional environments associated with modern large rivers.
According to a review for the world's continental-scale rivers by Ashworth et al. (2012), the
different combinations of sedimentation types lead to floodplain morphologies for big rivers
that can be classified into four main types, and Type 1 is a lacustrine-dominated river system
(as shown Figure 8A in Ashworth et al., 2012). The present-day largest anastomosing or
meandering rivers in the world often develop lakes between trunk streams and their major
tributaries (Maill et al., 1996; Ashworth et al., 2012). In other words, lake environments are a
common component of large river systems (Ashworth et al., 2012), especially in extensive
and low-gradient catchment areas, as in the Dongting Lake of the middle Yangtze (Chen et al.,
2007).

**References:**

- Ashworth, P. J. & Lewin, J. How do big rivers come to be different? *Earth Sci Rev.* **114**, 84–107
(2012).
Bohacs, K. M., A. R. Carroll, J. E. Neal, P. J. Mankiewicz. Lake-basin type, source potential, and
hydrocarbon character: an integrated-sequence-stratigraphic–geochemical framework, in E. H.
Gierlowski-Kordesch and K. R. Kelts, eds., *Lake basins through space and time: AAPG Studies in*

Geology 46, p. 3–34 (2000).
Carroll, A. R., Bohacs, K.M. Stratigraphic classification of ancient lakes: balancing tectonic and climatic
controls. *Geology* 27, 99–102 (1999).
Chen, Z., Xu, K., Watanabe, M. Dynamic hydrology and geomorphology of the Yangtze River. In:
Gupta, A. (Ed.), *Large Rivers: Geomorphology and Management*. Wiley, Chichester, pp. 457–469,
(2007).
Miall, A. D. The Geology of Fluvial Deposits: Sedimentary Facies, Basin Analysis, and Petroleum
Geology, 362–365, *Springer* (1996).
Sichuan Geological Survey Institute. The Report of Regional Geological Survey of Xichang at Scale 1:
250000 (China Industry Press, 2013).
Sichuan Bureau of Geology and Mineral Resources. Geological Map of the Xichang, Hexi, Xincun,
and Guolianggai Sheet (scale 1:50000) (Geological Publishing House Press, 1990).

(8) For the Chuxiong Basin, I stated that “thick evaporite are occurred in the Jiangdihe
Formation “. Yes, thin layers or nodules or crystals of evaporite can be formed in the arid
fluvial environment. Of course, we cannot see thick evaporites in the field in the Chuxiong
Basin, even in the Simao Basin and Khorat Basin since that the evaporite is easily dissolved
under the tropical monsoon climate in these basins. Previous publication showed that several
medium-scale salt mine of the Jiangdihe Formation were found (ref. 77). I am very curious
about what the authors stated in the Supplementary Note line 211-2113 “Thus, the presence of
thin evaporites is in contradiction with the existence of a low-gradient and continental-scale
river system”. So, what is exactly the authors’ opinion?

Similarly, the authors copy the stratigraphic column of Deng’s section in the Chuxiong Basin.
However, they wrongly place the basal conglomerates of the Matoushan Formation in Deng’s
paper as the Jiangdihe Formation in Supplementary Figure 1. Deng et al. (2018) stated that
“The Jiangdihe Formation represents a fluvial and shallow-lacustrine facies and the
Zhaojiadian Formation fines upwards and represents a shallow-to-marginal lacustrine facies”.
However, the authors proposed fluvial+lake and floodplain+fluvial environment for the
Jiangdihe and Zhaojiadian formation, respectively, without any sedimentary work.

**Reply:** From the published literature (ref.⁷⁷ (i.e., Yunnan Bureau of Geology and Mineral
Resources, 1965)) it may indeed appear that several sets of salt-bearing sequences occur in the
Jiangdihe Formation. However, these sporadic salt-bearing sequences are still dominated by
terrigenous clastic materials (e.g., 50% – 70% of siltstone and glutenite; ref.⁷⁷), implying the
lacustrine-dominated depositional systems were not persistently stable. In addition, deposition
of the salt-rich calcareous mudstone facies only developed in the middle-upper part of the
Jiangdihe Formation in local areas, which may have resulted from local lowlands (e.g., mire,
pond, and oxbow lakes), or short-term climatic fluctuations. Please note that the current
drainage basins of the world’s largest rivers also contain some small lakes (e.g., Amazon,
Congo, Orinoco, and Mississippi), providing modern examples consistent with our idea.

More significantly, all Late Cretaceous–early Palaeogene stratigraphic sections in the
Chuxiong Basin did not record prolonged lacustrine deposition. From a sedimentological
perspective at the basin-scale, all literature consistently suggests that fluvial and shallow
lacustrine depositional environments were predominant during deposition of the Upper
Cretaceous Matoushan and Jiangdihe Formations (Yunnan Bureau of Geology and Mineral
Resources, 1965, 1966, 1996; Xue et al., 2019). For example, a thick wedge-shaped sandstone
body, known as “Fangjiahe wedge-sandstone body”, was found in the lower part of the
Jiangdihe Formation (Yunnan Bureau of Geology and Mineral Resources, 1996). This

sandstone body is characterized by gray purple and dark purple thick lithic quartz sandstone
interbedded with purplish red siltstone and mudstone, which are laterally continuous over
scales of hundreds of meters, and longitudinally spread about 4.3 km. Single bed thickness is
commonly >2 m, and sandstones show large-scale cross stratification, consistent with
deposition by a large-scale fluvial system. For the early Paleogene Zhaojiadian Formation,
Deng et al., (2018) interpreted this unit in the Yijiu section as dominated by a
shallow-to-marginal lacustrine system, but approximately synchronous lateral facies changes
may exist across the basin. The Zhaojiadian Formation in the Guatang-Sanzhi section (near
the Yijiu section) clearly contains fluvial deposits (Yunnan Bureau of Geology and Mineral
Resources, 1965), which shows typical fluvial sedimentary features including: (i) purplish
thickly bedded red sandstone interbedded with siltstone and mudstone beds; (ii) individual
sandstone beds with basal granule lags and multi-scale cross stratifications, which probably
resulted from bar migration and channel migration and filling processes; (iii) Mud cracks,
calcareous nodules, and bioturbation structures are common in finer-grained deposits, likely
representing floodplain or overbank depositional environments.

Biofacies records in the Chuxiong Basin provide further evidence. Here the fossil record
is dominated by ostracodes, freshwater lamellibranchia, fish species, and plant fossils which
are typical of the paleontological assemblages of many other hydrologically open lakes
(Carroll and Bohacs, 1999; see Figure 1 in Bohacs et al., 2000). Taken together, these findings
strongly suggest that the Chuxiong Basin was not a protracted endorheic basin, but instead a
hydrologically open basin over the time span of accumulation of depositional sequences
during the late Cretaceous–early Paleogene. Such overfilled lake setting has been believed to
be very closely related to perennial river systems (Carroll and Bohacs, 1999; Bohacs et al.,
2000). We have added these significant supplements to the revised manuscript and
Supplementary Note. Finally, errors in placement of the boundary between the Matoushan and
Jiangdihe Formations in Supplementary Figure 1, and inappropriate sedimentary environment
assemblages of Deng et al., (2018) have been corrected.

**References:**

- Bohacs, K. M., A. R. Carroll, J. E. Neal, P. J. Mankiewicz. Lake-basin type, source potential, and
hydrocarbon character: an integrated-sequence-stratigraphic–geochemical framework, in E. H.
Gierlowski-Kordesch and K. R. Kelts, eds., Lake basins through space and time: AAPG Studies in
Geology 46, p. 3–34 (2000).
- Carroll, A. R., Bohacs, K.M. Stratigraphic classification of ancient lakes: balancing tectonic and climatic
controls. *Geology* 27, 99–102 (1999).
- Xue C, D., Xiang K., Hu, T. Y., et al. Sedimentary Environments of Late Cretaceous Ore-bearing
Sequences at the Guihua Copper Ore Field in the Northern Chuxiong Basin, Yunnan Province, SW
China. *Acta Sedimentologica Sinica*, 37, 491-501 (2019). doi: 10.14027/j.issn.1000-0550.2018.153
- Yunnan Bureau of Geology and Mineral Resources. Geological Map of the Chuxiong Sheet (scale
1:200000) (Geological Publishing House Press, 1966).
- Yunnan Bureau of Geology and Mineral Resources. Geological Map of the Chuxiong Sheet (scale
1:50000) (Geological Publishing House Press, 1996).
- Yunnan Bureau of Geology and Mineral Resources. Geological Map of the Dayao Sheet (scale
1:200000) (Geological Publishing House Press, 1965).

(9) Therefore, I easily found that the authors just simply copied two sections in the Xichang
and Chuxiong basins of Deng et al. (2018). Based on my personal opinion and authors' cited
publication (ref 16, Deng et al., 2018), I do prefer a fluvial and lacustrine environment during
that time. It also indicates that the sedimentary environment fluctuate between fluvial and
lacustrine and is not stable in such a long time range as the authors' claimed.

**Reply:** We agree the reviewer that fluvial and lacustrine sedimentary environments were
dominant in the studied basins during the late Cretaceous–early Palaeogene. As mentioned in
points 6 and 7 above, such fluvial-lacustrine facies associations together with characteristic
freshwater fossil assemblages demonstrates that these late Cretaceous–early Palaeogene
basins are best interpreted as typical overfilled lake basins or hydrologically open lakes based
on Carroll and Bohacs (1999) and Bohacs et al. (2000). This suggests the influx rate of water
+ sediment fill generally exceeds potential accommodation for these studied basins, which
would allow a development of through-going river systems (Carroll and Bohacs, 1999).

Although the two typical stratigraphic columns of the Xichang and Chuxiong basins in
Fig. S1 were cited from Deng et al., (2018), we have included more detailed sedimentological
information from a basin-scale perspective in the revised manuscript (lines 138–150) and
Supplementary Note. All evidence and/or observations do not support the interpretation of
endorheic lakes for late Cretaceous–early Palaeogene basins envisaged by the reviewer (See
revised Supplementary Note for more details).

We have emphasized a prolonged period of regional tectonic stability, rather than the
stability of sedimentary environments. Temporal and spatial shifts of sedimentary
environments between fluvial and lacustrine are very common in large-scale drainage systems
(e.g., the Yinchuan rift basin through which the Yellow River flows; see Figure 5 in Ma et al.,
2021), possibly due to channel migration, climate fluctuations, differential
subsidence/compaction and local tectonic perturbation. Especially in a large-scale drainage
system that includes several overflow lakes as interpreted in this study, shifts between fluvial
and lacustrine conditions would be frequent.

**References:**

566 Ma, X. D., Yin, G. M., Wei, C. Y., et al. High-resolution late Pliocene-quaternary magnetostratigraphy
of the Yinchuan Basin, NE Tibetan Plateau. *Quaternary International*, **607**, 120-127 (2021)

(10) As for the Simao and Khorat basins, no matter the evaporite is marine or non-marine, it is
impossible to form such a salt giant in an open environment (Please see Warren, 2016,
*Evaporites-A Geological Compendium*). The authors claimed in the response letter line 417 to
422 that marine incursion evaporites cannot be used as an argument against a fluvial system
that drained into the Neo-Tethyan Ocean. I guess the authors argued the seawater may be
intruded from the Neo-Tethyan Ocean (from south to north) and thus claimed like that.
However, marine incursions were not from the Neo-Tethyan Ocean, but rather from the
Meso-Tethys Ocean or proto-Paratethys Sea as the cited publications (in Response letter
line 14 to 15) argued. Therefore, the authors used the marine incursion evaporite to support the
existence of fluvial system is unreasonable and untenable. Moreover, the lithological and
sedimentary features is indicative of a typical saline lake by many publications (Hite and
Japakasetr, 1979; El Tabakh et al., 1999; Zhang et al., 2013; Liu et al., 2018; Qin et al., 2020;
Wang et al., 2021). As such, the Xichang, Chuxiong, Simao, and Khorat Basins during the so

called late Cretaceous to Early Paleogene are mostly lacustrine. And Simao and Khorat saline
lakes were not exorheic.

**Reply:** We have explained these issues in our response to the reviewer's major points (1) and
(2) above. Here we need to add that we have never considered the use of the marine incursion
evaporite to directly support the existence of a large-scale fluvial system that presented this
study. Instead, Late Cretaceous–early Palaeogene sea transgression in the Simao–Khorat
Basins strongly suggests these areas were very close to the ocean and at a relatively low
elevation (near sea level) at that time, which allows the more reasonable interpretation –
advocated in our paper – that the proposed continental-scale river system discharged
southward into the Neo-Tethyan Ocean (see Fig. 4 for an illustration of our inferred
paleogeographic position).

(11) Provenance analysis

I suggest you put the newly added data of Simao and Khorat in the MDS plot in
Supplementary Figure 6. And it is obvious that the Indochina plot so far away from your
samples, indicating there was no provenance connection between them. So why do the
authors' believe the fluvial system can flow to the Indochina?

**Reply:** Thanks, we had put the newly added data of the Simao basin in the MDS plot in the
last submitted version. The Simao Basin and Muang Xai Basin are located on the Indochina
terrane. Detrital zircon data from late Cretaceous–early Paleogene samples of the two basins
show strikingly consistent peaks at 200–300 Ma, 390–480 Ma, 700–900 Ma, 1700–2000 Ma,
and 2300–2600 Ma (Wang et al., 2014; Wang et al., 2017; Chen et al., 2017; Yan et al., 2021;
Fig. S8), which has been interpreted to be sourced mainly from the Songpan-Ganzi, Yidun,
and Qiangtang terranes, rather than proximal source area (i.e., Indosinian terrane) (Wang et al.,
2014; Wang et al., 2017; Chen et al., 2017; Yan et al., 2021). In other words, if the Simao and
Muang Xai Basins were endorheic basins during the late Cretaceous–early Paleogene,
sediments in the two basins must be derived from surrounding older rocks in the Indosinian
terrane, and we would expect the widespread Triassic metamorphic rocks surrounding these
basins to provide a major source of detrital material and a unique prominent peak at 200–
250 Ma. But this is not the case. This is why we have believed there was a large-scale
paleohydraulic system connected the current Tibetan Plateau and the Simao-Muang Xai
basins during the late Cretaceous–early Paleogene, as discussed in the main text (lines 224–
227).

The dominance of upper reaches/headwaters-derived provenance is also commonly
observed in the middle and lower reaches of most today's large river systems. For example,
southeastern Tibet (the upper reaches of the Yangtze River) has been determined to be the
dominant sediment contributor for stored sediments in the middle-lower reaches of the
Yangtze River (Zhang et al., 2020); Similarly, the drainage area of the upper Yellow River
(i.e., northeastern Tibetan Plateau) has been interpreted as a major sediment source for the
Yinchuan Basin (approximately 1500 km northeast of the headwater area) (Wang et al., 2019).

**References:**

Wang, Z., Nie, J., Wang, J., Zhang, H., Peng, W., Garzanti, E., et al. Testing contrasting models of the
formation of the upper Yellow River using heavy-mineral data from the Yinchuan Basin drill
cores. *Geophysical Research Letters*, **46**, 10338–10345. (2019).

Zhang, Z., Daly J. S., Li C., et al. Southeastern Tibetan Plateau serves as the dominant sand contributor
to the Yangtze River: Evidence from Pb isotopic compositions of detrital K-feldspar. *Terra Nova*,

33, 195–207 (2020).

(12) Landscape and large river system.

Such a long and large river system existed in the eastern Tibetan Plateau maybe need a higher
elevation and large erosion, and thus will results in enough clastic materials to transport into
the Neo-Tethys Ocean. Together with the authors' claimed that there were no proximal
sources, I have three concerns. (1) How can a low-relief landscape with slow exhumation and
erosion (slow cooling) can supply so many clastic materials to the Neo-Tethys Ocean in such
a long distance (at least 3000 km)?

**Reply:** We do not argue that a higher elevation is a prerequisite for the establishment of large
river systems, because a stable long-wavelength topographic slope is sufficient to sustain the
longevity of a continental-scale drainage system. The combination of depositional
environments, thermochronological results, landscape modeling, available palaeo-altimetric
data, and tectonic setting discussed in this study shows that the proposed large river was
low-gradient, low-energy, and long-lived, and also was accompanied by extensive floodplains,
wetlands, and exorheic lakes that are similar to modern continental, lowland rivers. This
implies this river system was likely dominated by lateral planation over most of the region,
rather than vertical incision. Concurrently, a prolonged Late Cretaceous–early Palaeogene
tectonic stability in eastern Tibet would not have provided sufficient relief to generate
significant amounts of denudation. Last but not least, sediment budget and provenance studies
widely demonstrate that the majority of the eroded sediments of large rivers are stored in
terrestrial basins (Potter, 1978; Nie et al., 2015) which resemble the Sichuan, Xichang, Huili,
Chuiong, Simao basins in our fluvial model. In this case, those pre-existing upland reliefs
were progressively smoothed away from the main riverine network as individual negative
relief elements were filled with fine-materials over tens of millions of years. Thus, the detrital
materials flowing into the Neo-Tethys Ocean may be only a small fraction of the total
sediment load; however, more thorough examination of Late Cretaceous–early Palaeogene in
the Khorat Basin and the area to the south, especially adjacent sea area, will be required to
fully ascertain sediment budget delivered by the proposed transcontinental river during that
time period, using offshore drilling and seismic investigations.

With respect to the ~3000 km long axis of the river envisaged by reviewer #1, this
appears to be overestimated. Because both the Simao and Khorat Basins were located at
paleolatitudes of ~20°–30° N during the late Cretaceous to early Paleocene based on available
palaeomagnetic results (e.g., Charusiri et al., 2006; Singsoupho et al., 2014; Zhang et al.,
2018; Yan et al., 2021), which means the paleo-position of the Khorat Plateau during the late
Cretaceous was located to the northwest of the present position, at least ~750 km apart
(Singsoupho et al., 2014).

**References:**

Charusiri, P., Imsamut, S., Zhuang, Z., Ampaiwan, T., Xu, X. Paleomagnetism of the earliest
Cretaceous to early late Cretaceous sandstones, Khorat Group, Northeast Thailand: Implications for
tectonic plate movement of the Indochina block. *Gondwana Res.* **9**, 310–325 (2006).
Nie, J. S., Stevens, T., Rittner, M., Stockli, D., Garzanti, E., Limonta, M., et al. Loess Plateau storage
of northeastern Tibetan Plateau-derived Yellow River sediment. *Nature Communications*, **6**, 8511
(2015).
Singsoupho S, Bhongsuwan T, Elming S Å. Tectonic evaluation of the Indochina Block during
Jurassic-Cretaceous from palaeomagnetic results of Mesozoic redbeds in central and southern Lao
PDR. *Journal of Asian Earth Sciences*, 2014, **92**, 18–35 (2014).

Potter, P. E. Significance and origin of big rivers. *J. Geol.* 13–33 (1978).
Yan, M. D., Zhang, D. W., Fang, X. M., Zhang, W. L., Song, C. H et al. New Insights on the age of the
Mengyejing formation in the Simao basin, SE Tethyan domain and its geological implications. *Sci*
*China Earth Sci.* **64**, 231–252 (2021).
Zhang D., Yan M., Fang X., Yang Y., Zhang T., Zan J., Zhang W., Liu, C & Yang Q.
Magnetostratigraphic study of the potash-bearing strata from drilling core ZK2893 in the Sakhon
Nakhon Basin, eastern Khorat Plateau. *Palaeogeogr Palaeoclimatol Palaeoecol.* **489**, 40–51
(2018).

(13) If the river do flow into the Neo-Tethys Ocean, do the authors have any evidence of
sedimentary sequence in delta and offshore Thailand?

**Reply:** On the basis of frequent marine incursions and a paleo-elevation of near sea level in
the Simao–Khorat areas, detrital zircon provenance results from the Muang Xai and Khorat
Basins, a dominant south-directed paleocurrent from the Khorat Basin, and reconstructed
paleo-continental configuration of East Asia (see revised manuscript for details), we interpret
that the Late Cretaceous–early Palaeogene large river system discharged into the Neo-Tethys
Ocean. However, it is very difficult to reconstruct basin paleogeography, and to exactly
determine where this river discharged into the Neo-Tethys Ocean at that time, due to
significant paleo-position change resulting from extrusion of Indochina, complex block
rotation, and intense uplift-erosion since that time. For example, 1000–1250 m of Late
Cretaceous deposits once covered the whole of the Khorat Plateau area, and has subsequently
been eroded (Booth and Sattayarak, 2011), which explains why remnants of the Late
Cretaceous–early Palaeogene Maha Sarakham Formation are mainly exposed in low-lying
areas, particularly in the central part of the Khorat Plateau (Meesook, 2000). Moreover, a
latitudinal movement of the Indochina Block of about 5–11° (translation of about 750–1700
700 km in the southeastward direction along the Red River Fault) and clockwise rotation of 13–18°
with respect to the South China Block have occurred since the Mesozoic (Charusiri et al.,
2006; Singsoupho et al., 2014).

Nonetheless, we still try to explore this issue from a broader context. In the Borneo area
south of the Indochina terrane, there are very thick, late Cretaceous to early Cenozoic deep
water sub-marine fan deposits, known as the Rajang Group and Kayan Group. Field
observations and sedimentological analyses indicate such very thick deep-water sequence
must have been deposited in one of the world's largest ancient submarine fans and transported
by a large river system (Galín et al., 2017). Galín et al. (2017) identified the late Cretaceous–
early Paleogene Rajang Group as deposits of a sub-marine fan similar in size to some modern
river fans like the Amazonas or the Mississippi, remarkably mismatching present-day smaller
river systems in Borneo. Detrital zircon results from late Cretaceous–early Paleogene samples
have shown that these deposits were likely from SW Borneo, Sibumasu, and Indochina based
on a dominance of Mesozoic zircon ages that derived proximally from widespread Mesozoic
magmatic arcs (Galín et al., 2017; Breitfeld and Hall, 2018). However, some distal
provenance signals were diluted or even overridden by a robust proximal source given that the
zircon yield of magmatic rocks is significantly higher than that of sedimentary rocks. We note
that late Cretaceous–early Paleogene samples from the giant marine fan distinctly show three
Precambrian peaks at 2400–2600 Ma, 2000–1600 Ma, and 1100–800 Ma (Fig. S8; also see
Figure 11 in Galín et al., 2017, and Figure 12 in Breitfeld and Hall, 2018), suggesting the
Songpan-Ganzi and Upper Yangtze terranes as possible source areas. Moreover, Paleozoic and

Precambrian zircons are usually subrounded to rounded and also indicate recycling from older
sediments as important source (Galín et al., 2017). Thus, the Borneo fan could be a potential
sink of the large river presented this study, but a further investigation of late Cretaceous–early
Paleogene strata from the Khorat to Borneo area is needed to test this hypothesis (revised
manuscript, lines 252–256; Fig. S8).

In any case, it is clear from the above review that the Simao–Khorat area was very close
to the ocean at that time, and even a marginal marine or sea/land facies has been identified
within the Late Cretaceous strata (Wang et al., 2021). Therefore, a more reasonable
interpretation is that the proposed continental-scale river system discharged into the
Neo-Tethyan Ocean.

**References:**

- Breitfeld, H. T., and Hall, R. The eastern Sundaland margin in the latest Cretaceous to Late Eocene:
Sediment provenance and depositional setting of the Kuching and Sibuluan Zones of Borneo.
*Gondwana Research* **56**, 34–64 (2018).
- Booth, J. E., Sattayarak, N., 2011. Subsurface Carboniferous–Cretaceous geology of Northeast
Thailand. In: Ridd, M.F., Barber, A.J., Crow, M.J. (Eds.), *The Geology of Thailand*. *Geological*
*Society London*, pp. 184–222.
- Charusiri, P., Imsamut, S., Zhuang, Z., Ampaiwan, T., Xu, X. Paleomagnetism of the earliest
Cretaceous to early late Cretaceous sandstones, Khorat Group, Northeast Thailand: Implications for
tectonic plate movement of the Indochina block. *Gondwana Research* **9**, 310–325 (2006).
- Galín, T., Breitfeld, H. T., Hall, R., Sevastjanova, I. Provenance of the Cretaceous–Eocene
Rajang Group submarine fan, Sarawak, Malaysia from light and heavy mineral assemblages and U–Pb
zircon geochronology. *Gondwana Research* **51**, 209–233 (2017).
- Singsoopho S, Bhongsuwan T, Elming S Å. Tectonic evaluation of the Indochina Block during
Jurassic–Cretaceous from palaeomagnetic results of Mesozoic redbeds in central and southern Lao
PDR. *Journal of Asian Earth Sciences*, 2014, **92**, 18–35 (2014).
- Wang, L. C., Zhong, Y. S., Xi, D. P., Hu, J. F. & Ding, L. The Middle to Late Cretaceous marine
incursion of the Proto-Paratethys sea and Asian aridification: A case study from the Simao–Khorat
salt giant, southeast Asia. *Palaeogeogr Palaeoclimatol Palaeoecol.* **567**, 110300 (2021)

(14) Sections and samples Authors provide the GPS of all samples, however I found sample
distance in the Dujiangyan section is nearly up to 30 km from Google Earth. That is
absolutely not the sampling action in the same section because the Dujiangyan section only
have ca. 800 m in thickness. The same as the Leshan section, two sample distance is up to 21
756 km. So it is absolutely questionable of the sampling.

The authors acknowledged that all stratigraphic and sedimentary work in the Xichang and
Chuxiong basins are from previous publication. And as I mentioned above, the stratigraphic
columns in these two basins are totally copied from Deng et al. (2018). So I doubt that the
authors' sampling is consistent with the section description. Therefore, from the authors'
sampling and stratigraphic work, I have to trace back to the age of the section again. All age
constraints from this study is the fossils which is done about 40 years ago. Previous fossil
results were from regional mapping at scale of 1:200,000 or type section. The studied sections
are obviously not the type section, so previous results should not be applied directly. How can
the regional fossils be used in whole basin?

**Reply:** This has been clarified in the revised text. Unfavorable outcrop conditions of late
Cretaceous–early Palaeogene deposits exposed in the studied basins preclude the possibility

of dating every sample horizon. Field investigations have revealed that Upper Cretaceous to
lower Palaeocene outcrops in the southwestern Sichuan Basin were either partially covered by
vegetation or slightly deformed. The observed outcrops are restricted to road cuts, coast and
valleys. Thus, sampling locations in the Dujiangyan and Leshan areas were not from a single
cross section, but an integrated stratigraphic section of a gently folded anticline or syncline.
Stratigraphic horizons of samples were based on mapped geological cross-section, lithofacies
associations, depositional systems, strata thickness, and deformation patterns that described
from detailed 1: 200000 and 1: 50000 regional-scale geological maps. In order to avoid
misunderstanding, we used the Dujiangyan *area* and Leshan *area* instead of the Dujiangyan
*section* and Leshan *section* in the revised manuscript.

The section sampled in the Xichang basin in this study is the Mishi Section of Deng et al.
(2018), thus the sampling horizons in Fig. S1 are correct. In fact, late Cretaceous–early
Palaeogene stratigraphic successions in the Xichang basin have only been exposed in a single
syncline including the type section, which is conducive to spatiotemporal correlation of
limited sedimentary sequences of late Cretaceous–early Palaeogene. The extent of late
Cretaceous–early Palaeogene strata in the Chuxiong Basin is larger than that in Xichang Basin,
but it is also exposed in two adjacent synclines based on regional mapping at scale of
1:200000 and geological cross-sections. Moreover, multiple late Cretaceous–early Palaeogene
stratigraphic sections including the Yijiu section from Deng et al. (2018) in the Chuxiong
Basin show very similar stratigraphic cycles, lithofacies and facies associations, and marked
beds, despite slightly varying lateral facies. Combined with well-mapped geological
cross-sections, this allows the sampling horizons in different sites to be easily set on
corresponding positions of typical sections. We admit that the sampling horizons of this study
may not be very exact due to unavoidable uncertainties, but adequate sedimentation interval
between samples from one basin suggests the provenance signal has been stable over tens of
millions of years. Just as the comment from Reviewer #4, *“the suite of consistent detrital*
*zircon ages from depositional basins spanning 600-750 km from north to south provides*
*critical support to their hypothesized drainage basin geometry and the proposed timing of its*
*existence as an integrated depositional system. To me, this spatial-temporal consistency is a*
*key factor supporting the interpretation offered by these authors.”*

About depositional age constrained by palaeontological results, abundant ostracods and
charophyta presented this study were obtained from the whole basin, rather than a type
section. Characteristic ostracods and/or charophyta assemblage is also one of justifications of
sequence correlation. Also, please note that palaeontological types, formation boundaries, and
even sedimentary facies results in Deng et al., (2018) were mainly taken from previous
province-scale or regional-scale geology reports. By comparison, we have added more
detailed materials on the basis of research of Deng et al., (2018) from smaller-scale geology
reports (e.g., Yunnan Bureau of Geology and Mineral Resources, 1996; Sichuan Bureau of
Geology and Mineral Resources, 1990)

**References:**

Sichuan Bureau of Geology and Mineral Resources. Geological Map of the Xichang, Hexi, Xincun,
and Guolianggai Sheet (scale 1:50000) (Geological Publishing House Press, 1990).
Yunnan Bureau of Geology and Mineral Resources. Geological Map of the Chuxiong Sheet (scale
1:50000) (Geological Publishing House Press, 1996).

(15) Standard zircons

I commented the dating results of the Standard zircons, but I cannot see correct or appropriate
response. Yes, as the authors replied, the detrital zircon dating is a usual way. But for each test,
your 91500 and GJ-1 zircons should generate their ages. What I asked is you should provide
all the ages of 91500 and GJ-1 during your experiment, so that I can determine your deviation
between the test values and recommended ages. And then I will have an idea about the
reliability of all ages of the samples. So you cannot use the dating results in Yuan et al. (2004)
to tell me that the ages in your study is reliable.

**Reply:** As requested by reviewer #1, we added a Table that contains dating results of the
standard zircons (available from <https://figshare.com/s/a6f79d6f03b18aec1d44>). In our study,
the external standard zircons of 91500, GJ-1 and Qinghu yielded $^{206}\text{Pb}/^{238}\text{U}$ weighted ages of
**1062.5 ± 1 Ma (n = 212), 604.6 ± 2 Ma (n = 137), and 159.9 ± 0.83 Ma (n = 34)**,
respectively. These are consistent with reference ages of $1063.1 ± 8.1$ Ma (91500, Yuan et al.,
2004), $599.8 ± 4.5$ Ma– $610 ± 1.7$ Ma (GJ-1, Jackson et al., 2004 and Eelhlou et al., 2006), and
$159 ± 0.2$ Ma (Qinghu, Li et al. 2013). This indicates our detrital zircon data are credible
(revised manuscript, lines 339–343).

**References**

Eelhlou S, Belousova E, Griffin W L, et al. Trace element and isotopic composition of GJ red zircon
standard by laser ablation. *Geochimica et Cosmochimica Acta*, 70 (18), A158 (2006).

Jackson, S.E., Pearson, N.J., Griffin, W.L., Belousova, E.A. The application of laser
ablation-inductively coupled plasma-mass spectrometry to in situ U-Pb zircon geochronology.
*Chemical Geology*, 211, 47–69 (2004). <https://doi.org/10.1016/j.chemgeo.2004.06.017>.

Li, X., Tang, G., Gong, B., Yang, Y.H., Hou, K., Hu, Z., Li, Q., Liu, Y., Li, W. Qinghu zircon: A
working reference for microbeam analysis of U-Pb age and Hf and O isotopes. *Chinese Science*
*Bulletin*, 58, 4647–465 4 (2013). <https://doi.org/10.1007/s11434-013-5932-x>.

Yuan, H. L., Gao, S., Liu, X. M., Li, H. M., Günther, D. & Wu, F. Y. Accurate U-Pb age and trace
element determinations of zircon by laser ablation inductively coupled plasma mass spectrometry.
*Geostand. Geoanal. Res.* 28, 335–370 (2004).

**Reviewer #2**

I reviewed the revision submitted by Huiping Zhang and co-authors. All of my comments
were sufficiently addressed. I support publication of this manuscript.

Andrew Laskowski

**Reply:** We sincerely appreciate Dr. Andrew Laskowski for accepting our corrections in
previous manuscript.

**Reviewer #3**

Dear Editor and Authors,

Thanks for the detailed response to my previous comments. After reading the manuscript, I
thought there are still a few problems that need to be addressed before publication.

(1) First, the authors present a few pictures to prove a continental scale river through the
basins in SE Tibet, but I suspect this. As shown in the pictures, the laminated and fine grained
sandstones and mudstones are better to be interpreted in lacustrine environment, as

suggested by the Reviewer 1. If the sediments were transported by a large river, we should
observe some cross beddings, which are scarce in these basins.

**Reply:** Thank you for this comment. Apart from lacustrine facies, late Cretaceous–early
Paleogene strata from these studies basins also contain significant fluvial deposits, which
show typical fluvial sedimentary features, including various-scale cross stratifications, upward
fining sequences, basal granule lags, lenticular or tabular sandstone beds with erosional
contacts, and crevasse splay deposits, as widely described by many studies (e.g., Yunnan
Bureau of Geology and Mineral Resources, 1965, 1966, 1996; Sichuan Bureau of Geology
and Mineral Resources, 1990; Sichuan Geological Survey Institute, 2013; Deng et al., 2018;
see revised Supplementary Note for details; Fig. R1). It is noteworthy that the type of sheet
and ribbon sandstone interbedded with the mudstone beds has commonly been formed by
aggradation of fixed channels and simple fills (Miall, 1996), rather than lacustrine facies due
to the prevalence of mud cracks, calcareous nodules, and bioturbation structures within
siltstone and mudstone interlayers.

Fig. R1 Representative sedimentary structures of fluvial architecture in studies late Cretaceous–early
Palaeogene basins. (a) Lenticular sand body of channel in fluvial deposits (Dujiangyan area). (b)
Clast-supported conglomerates and coarse-grained sandstone showing upward fining trend (Dujiangyan
area). (c) Reddish sandstone with large-scale tabular-cross bedding (Leshan area). (d) Planar cross-bedded
sand overlying suspected rippled cross-bedded sand (Xichang Basin). (e) Rippled cross-bedded and
climbing cross-bedded sandstone (Xichang Basin). (f) Floodplain and crevasse splay deposits;
fine-grained deposits are floodplain fines, lenticular sandstones are interpreted as crevasse splays (Huili
Basin). All photographs taken by Xudong Zhao.

For authenticating the occurrence of fluvial environments, we collected detailed
sedimentological descriptions and depositional environment interpretations from more
detailed or more recent geological mapping surveys in the submitted Supplementary Note.
Again, this literature clearly indicates that late Cretaceous–early Paleogene sediments from
these basins were dominated by alternating fluvial and lacustrine environments (e.g., Sichuan
Geological Survey Institute, 2013; Sichuan Bureau of Geology and Mineral Resources, 1990;
Yunnan Bureau of Geology and Mineral Resources, 1965, 1966, 1996; Xue et al., 2019; see
revised Supplementary Note for details). More importantly, based on the identifying criterion
of lake types from Carroll and Bohacs (1999) and Bohacs et al. (2000), such fluvial-lacustrine
facies association has been ascribed to be typical of the deposits of hydrologically open lakes
or exorheic lakes associated with perennial river systems (see Figure 7 in Bohacs et al., 2000).

The biofacies records provide further evidence. In sequences exposed in these late
Cretaceous–early Paleogene basins, ostracods, freshwater lamellibranchia, fish species, and
plant fossils have been widely found — typical paleontological assemblages of many other
hydrologically open lakes (Carroll and Bohacs, 1999; Bohacs et al., 2000; Figure 1 in Bohacs
et al., 2000). These findings strongly suggest that these studied basins are best interpreted as
hydrologically open basins/overflow basins associated with a large-scale river system during
the late Cretaceous–early Paleogene (Carroll and Bohacs, 1999; Bohacs et al., 2000). We have
added this clarification and additional documentation to the revised manuscript (lines 146–
153) and revised Supplementary Note.

**References**

- Bohacs, K. M., A. R. Carroll, J. E. Neal, P. J. Mankiewicz. Lake-basin type, source potential, and
hydrocarbon character: an integrated-sequence-stratigraphic–geochemical framework, in E. H.
Gierlowski-Kordesch and K. R. Kelts, eds., *Lake basins through space and time: AAPG Studies in
Geology* 46, p. 3–34 (2000).
- Carroll, A. R., Bohacs, K.M. Stratigraphic classification of ancient lakes: balancing tectonic and climatic
controls. *Geology* **27**, 99–102 (1999).
- Deng, B., Chew, D., Jiang, L., Mark, C., Cogne, N., Wang, Z. J. & Liu, S. G. Heavy mineral analysis
and detrital U-Pb ages of the intracontinental Palaeo-Yangtze basin: Implications for a
transcontinental source-to-sink system during Late Cretaceous time. *Geol. Soc. Am. Bull.* **130**,
2087–2109 (2018).
- Miall, A. D. *The Geology of Fluvial Deposits: Sedimentary Facies, Basin Analysis, and Petroleum
Geology*, 362–365, *Springer* (1996).
- Sichuan Geological Survey Institute. *The Report of Regional Geological Survey of Xichang at Scale 1:
250000* (China Industry Press, 2013).
- Sichuan Bureau of Geology and Mineral Resources. *Geological Map of the Xichang, Hexi, Xincun,
and Guolianggai Sheet (scale 1:50000)* (Geological Publishing House Press, 1990).
- Yunnan Bureau of Geology and Mineral Resources. *Geological Map of the Chuxiong Sheet (scale
1:200000)* (Geological Publishing House Press, 1966).
- Yunnan Bureau of Geology and Mineral Resources. *Geological Map of the Chuxiong Sheet (scale
1:50000)* (Geological Publishing House Press, 1996).
- Yunnan Bureau of Geology and Mineral Resources. *Geological Map of the Dayao Sheet (scale
1:200000)* (Geological Publishing House Press, 1965).
- Xue C, D., Xiang K., Hu, T. Y., et al. Sedimentary Environments of Late Cretaceous Ore-bearing
Sequences at the Guihua Copper Ore Field in the Northern Chuxiong Basin, Yunnan Province, SW
China. *Acta Sedimentologica Sinica*, **37**, 491-501 (2019).

(2) I am also disagree with the statement that the Late Cretaceous-Paleogene sediments in
Simao Basin is still marine facies. The sediments have been terrestrial since late Jurassic.

**Reply:** Thanks again, this appears to be a misunderstanding. As mentioned by reviewer, late
Cretaceous–early Paleogene red beds in the Simao–Khorat Basins have been consistently
interpreted as deposits of the alternating fluvial–lacustrine environments under continental
setting (e.g., Chen et al., 2017; Yan et al., 2021). However, variable-thickness evaporate
intervals (e.g., anhydrite, gypsum, sylvite, and halite) are commonly sandwiched in these late
Cretaceous–early Paleogene terrestrial red beds in these basins. Although discussion still
persists, published datasets on sedimentology proxies, biomarkers, element geochemistry, and
isotopic geochemistry consistently indicate that late Cretaceous to early Paleogene evaporites
are mainly of marine origin (see Figure 6 in Zhang et al., 2013), which has generally been
interpreted as the result of a transgression of the Tethys ocean (e.g., Hite and Japakasetr, 1979;
El Tabakh et al., 1999; Zhang et al., 2013; Liu et al., 2018; Qin et al., 2020; Wang et al., 2021;
Rattana et al., 2021). This suggests that the Simao–Khorat Basins was very close to the ocean
and at a relatively low elevation (near sea level) at that time.

In Borneo south of Khorat, a Mississippi- or Amazon- scale submarine fan system
persisted from the late Cretaceous to early Cenozoic, a requirement that can be satisfied by an
ancient transcontinental river. For details see our response to the comment by the Reviewer #1
above (**Lines 748–779** in this letter).

**References**

- Chen, Y., Yan, M., Fang, X., Song, C., Zhang, W., Zan, J. et al. Detrital zircon U-Pb geochronological
and sedimentological study of the Simao Basin, Yunnan: Implications for the early Cenozoic
evolution of the Red River. *Earth Planet. Sci. Lett.* **476**, 22–33 (2017).
- El Tabakh, M., Utha-Aroon, C. & Schreiber, B.C. Sedimentology of the Cretaceous Maha Sarakham
evaporites in the Khorat Plateau of northeastern Thailand. *Sediment. Geol.* **123**, 31–62 (1999).
- Galin, T., Breittfeld, H. T., Hall, R., Sevastjanova, I. Provenance of the Cretaceous–Eocene
RajangGroup submarine fan, Sarawak, Malaysia from light and heavy mineral assemblages and U-Pb
zircon geochronology. *Gondwana Research* **51**, 209–233 (2017).
- Hite, R.J. & Japakasetr, T. Potash deposits of Khorat Plateau, Thailand and Laos. *Econ. Geol.* **74**, 448–
458 (1979).
- Liu, C.L., Wang, L.C., Yan, M.D., Zhao, Y.J., Cao, Y.T., Fang, X.M., Shen, L.J., Wu, C.H., Lv, F.L. &
Ding, T. The Mesozoic–Cenozoic tectonic settings, paleogeography and evaporitic sedimentation of
Tethyan blocks within China: implications for potash formation. *Ore Geol. Rev.* **102**, 406–425
(2018).
- Rattana, P., Choowong, M., He, M.-Y., Tan, L., Lan, J., Bissen, R., Chawchai, S., Geochemistry of
evaporitic deposits from the Cenomanian (Upper Cretaceous) Maha Sarakham Formation in the
Khorat Basin, northeastern Thailand. *Cretaceous Research*, in press (2021).
- Qin, Z., Li, Q., Zhang, X. G., Fan, Q. S., Wang, J. P., Du, Y. S., Ma, Y. Q., Wei, H. C., Yuan, Q. &
Shan, F. S. Origin and recharge model of the late cretaceous evaporites in the khorat plateau. *Ore*
*Geol Rev.* **116**, 103226 (2020).
- Wang, L. C., Zhong, Y. S., Xi, D. P., Hu, J. F. & Ding, L. The Middle to Late Cretaceous marine
incursion of the Proto-Paratethys sea and Asian aridification: A case study from the Simao-Khorat
salt giant, southeast Asia. *Palaeogeogr Palaeoclimatol Palaeoecol.* **567**, 110300 (2021).
- Yan, M. D., Zhang, D. W., Fang, X. M., Zhang, W. L., Song, C. H et al. New Insights on the age of the
Mengyejing formation in the Simao basin, SE Tethyan domain and its geological implications.
*Sci China Earth Sci.* **64**, 231–252 (2021).
- Zhang, X.Y., Ma, H.Z., Ma, Y.Q., Tang, Q.L. & Yuan, X.L. Origin of the late Cretaceous
potash-bearing evaporites in the Vientiane Basin of Laos: $\delta^{11}\text{B}$ evidence from borates. *J. Asian*
*Earth Sci.* **62**, 812–818 (2013).

(3) Second, I am not expert to modeling, so cannot judge the reliability of the modeling results to support a low relief at sea level before Eocene. But I am curious even there is a continental scale river flow to the Neo-Tethys Ocean, why must be a low relief existed?

Reply: Continental-scale drainage system’s longevity in essentially the same path is sustained by the persistence of a stable topographic gradient related to intercontinental setting and lithospheric-scale tectonics (Shephard et al., 2010; Faccenna et al., 2019; Morón et al., 2019). Long-lived large-scale river systems such as the Mississippi and Amazon are major agents for the formation of low-relief landscapes in stable continental interiors. How such low-elevation landscapes with their embedded rivers respond to tectonic deformation and mountain building is of crucial importance for the evolution of continental drainage networks and for reconstructing uplift patterns during continental collision and plateau formation.

Just as the comment from Reviewer #4, “*the key scientific question that this paper addresses is the origin of the widespread, high-altitude, low-relief surfaces that characterize much of the SE Tibetan Plateau: a region where local relief at present is typically quite high, with deeply incised river gorges, and rock-uplift rates are rapid in a global context. The presence of these long-lived, low-relief surfaces at rather high altitudes has long piqued our curiosity: why are they there; how did they form; what are modern analogues of their formative sequence; what data can be used to test various hypotheses?*” And the most significant and novel finding of our study is to link the development and extent of this long-lived paleo-river system to the formation of the low-relief landscape in eastern Tibet before Cenozoic uplift and plateau growth. Specifically, the existence of a large river system that persisted for tens of millions of years intrinsically reflects a prolonged period of tectonic stability, a long-wavelength dynamic topography, and a relatively stable base level. More significantly, the relatively stable base level and long-term tectonic stability required for the maintenance of this large-scale river played a central role in the development of a low-relief region in the continental interior, similar to the formation of modern low-relief landscape in central Australia and Mongolia (Stewart et al., 1986; Jolivet et al., 2007).

Again, as commended by Reviewer #4, “*this provocative, innovative synthesis and interpretation provides a potential resolution to a long-standing problems related to how there low-relief, high-elevation surfaces in SE and Eastern Tibet developed. I believe it is worthwhile to get this data set and interpretation “out there” for the interested audience to contemplate and to try to test with new data or reanalysis.*” Moreover, our findings are also of fundamental importance for understanding the relationship between fluvial morphology and topographic signatures and bear implications for future studies investigating the overarching mechanism of surface uplift and landscape evolution.

References

Faccenna, C., Glišović, P., Forte, A. et al. Role of dynamic topography in sustaining the Nile River over 30 million years. *Nature Geosci.* **12**, 1012–1017 (2019).
Jolivet, M., Ritz, J. F., Vassallo, R., Larroque, C., Braucher, R., Todbileg, M. et al. Mongolian summits: an uplifted, flat, old but still preserved erosion surface. *Geology* **35**, 871–874 (2007).
Morón, S., Cawood, P. A., et al. Long-lived transcontinental sediment transport pathways of East Gondwana. *Geology* **47**, 513–516 (2019).
Shephard, G., Müller, R., Liu, L. et al. Miocene drainage reversal of the Amazon River driven by plate–mantle interaction. *Nature Geosci.* **3**, 870–875 (2010).
Stewart, A. J., Blake, D. H. & Ollier, C. D. Cambrian river terraces and ridgetops in central Australia: Oldest persisting landforms? *Science* **233**, 758–761 (1986).

(4) Third, in recent studies, Clift et al. (2020) and Zheng et al. (2021) shown that, the Gonjo,
Jianchuan, Yuanjiang, and the Northern Vietnam basins have similar provenance in the early
Cenozoic, which supporting a southward-flowing river from eastern Tibet to the South China
Sea. This is contrast with the figure as suggested in this manuscript. I am wandering how the
authors reconcile this inconsistency.

**Reply:** As reviewer #3 points out correctly, based on regional similarities of provenance
signature from a series of sedimentary basins, several recent studies including Clift et al.,
(2020), Zheng et al., (2021), and He & Zheng et al., (2021) consistently suggest that a
paleo-Jinsha River flowed south to the South China Sea from the SE Tibetan Plateau
following the initial surface uplift of eastern Tibet in the late Eocene (ca. 35 Ma) (see Figure
14 in Clift et al., 2020). Yet our proposed continental-scale fluvial system of this study existed
during late Cretaceous to early Palaeogene (ca. 100–50 Ma), predating the India–Eurasia
collision and plateau uplift. Thus, there is no contradiction between the two asynchronous
drainage models.

If these interpretations hold, the discrepancy in the drainage pattern before and after
plateau uplift would require a river reorganization event that synchronized with a regional
tectonic-geomorphological transformation that involved synchronous basin development
(Jackson et al., 2018; Li et al., 2020), tectonic uplift (e.g., Hoke et al., 2014; Li et al., 2015),
latitudinal crustal shortening (Tong et al., 2017; Todrani et al., 2020), and faulting activities
during the late Eocene (e.g., Wang et al., 2012; Zhang et al., 2016). Also, this reorganization
event may be related to a change in deformation rate and style along the Ailao Shan–Red
River fault system at ~35 Ma (Scharer, 1994; Gilley et al., 2013), which could set up the
template for sediment routing that along the Ailao Shan–Red River fault system as shown by
the Figure 14 in Clift et al., (2020).

**References**

- Clift, P. D., Carter, A., Wysocka, A., Van Hoang, L., Zheng, H., & Neubeck, N. A Late Eocene-
Oligocene through-flowing river between the Upper Yangtze and South China Sea. *Geochemistry,*
*Geophysics, Geosystems*, **21**, e2020GC009046 (2020).
- He, M., Zheng, H., Clift, P. D., Bian, Z., Yang, Q., Zhang, B., & Xia, L. Paleogene sedimentary
records of the paleo-Jinshajiang (Upper Yangtze) in the Jianchuan Basin, Yunnan, SW China.
*Geochemistry, Geophysics, Geosystems*, **22**, e2020GC009500 (2021).
- Hoke, G. D., Liu-Zeng, J., Hren, M. T., Wissink, G. K. & Garzzone, C. N. Stable isotopes reveal high
southeast Tibetan Plateau margin since the Palaeogene. *Earth Planet. Sci. Lett.* **394**, 270–278
(2014).
- Gilley, L. D., T. M. Harrison, P. H. Leloup, F. J. Ryerson, O. M. Lovera, and J.-H. Wang, Direct dating
of left-lateral deformation along the Red River shear zone, China and Vietnam. *J. Geophys. Res.*
**108 (B2)**, 2127 (2003).
- Jackson, W. T., Jr., Robinson, D. M., Weislogel, A. L., Jian, X., & McKay, M. P. Cenozoic
development of the nonmarine Mula basin in the Southern Yidun terrane: Deposition and
deformation in the eastern Tibetan Plateau associated with the India-Asia collision. *Tectonics* **37**,
2446–2465 (2018).
- Li, S., Su, T., Spicer, R. A., Xu, C., Sherlock, S., Halton, A., et al. Oligocene deformation of the
Chuandian terrane in the SE margin of the Tibetan Plateau related to the extrusion of Indochina.
*Tectonics* **39**, e2019TC005974 (2020).
- Li, S., Currie, B. S., Rowley, D. B., & Ingalls, M. Cenozoic paleoaltimetry of the SE margin of the
Tibetan Plateau: constraints on the tectonic evolution of the region. *Earth Planet. Sci. Lett.* **432**,
415–424 (2015).
- Scharer, U., Z. Lian-Sheng, and P. Tapponnier, Duration of strike-slip movements in large shear zones:

- The Red River belt, China. *Earth Planet. Sci. Lett.* **126**, 379–397 (1994).
Toderani, A., Zhang, B., Speranza, F., & Chen, S. Paleomagnetism of the middle Cenozoic Mula Basin
(East Tibet): Evidence for km-scale crustal blocks rotated by midlower crust drag. *Geochemistry,*
*Geophysics, Geosystems*, **21**, e2020GC009225 (2020).
Tong, Y., Yang, Z., Mao, C., Pei, J., Pu, Z., Xu, Y., 2017. Paleomagnetism of Eocene red-beds in the
eastern part of the Qiangtang Terrane and its implications for uplift and southward crustal
extrusion in the southeastern edge of the Tibetan Plateau. *Earth Planet. Sci. Lett.* **475**, 1–14.
Wang, E., Kirby, E., Furlong, K. P., van Soest, M., Xu, G., Shi, X., Kamp, P. J. J. & Hodges, K. V.
Two-phase growth of high topography in eastern Tibet during the Cenozoic. *Nature Geosci.* **5**,
640–645 (2012).
Zhang, H. P., Oskin, M. E., Liu-Zeng, J., Zhang, P. Z., Reiners, P. W. & Xiao, P. Pulsed exhumation of
interior eastern Tibet: implications for relief generation mechanisms and the origin of
high-elevation planation surfaces. *Earth Planet. Sci. Lett.* **449**, 176–185 (2016).
Zheng, H. B., Clift, P. D., He, M. Y., et al. Formation of the First Bend in the late Eocene gave birth to
the modern Yangtze River, China. *Geology* **49**, 35–39 (2021).

**Reviewer #4**

Review of “*Existence of a continental-scale river system in eastern Tibet during the late*
*Cretaceous–early Palaeogene*”

I am assessing this paper on the basis of its integrated regional data, its innovative
interpretations, and the potential impact of those interpretations on our understanding of some
large-scale geomorphic anomalies along the eastern and SE margin of the Tibetan Plateau.
This approach likely stands in contrast to some other reviewers who are more familiar with
details of the local geologic-stratigraphic-petrographic-chronologic data sets that this
submission exploits and builds upon. The key scientific question that this paper addresses is
the origin of the widespread, high-altitude, low-relief surfaces that characterize much of the
SE Tibetan Plateau: a region where local relief at present is typically quite high, with deeply
incised river gorges, and rock-uplift rates are rapid in a global context. The presence of these
long-lived, low-relief surfaces at rather high altitudes has long piqued our curiosity: why are
they there; how did they form; what are modern analogues of their formative sequence; what
data can be used to test various hypotheses?

The authors argue that, despite some coeval tectonic uplift, a long-lived, ~north-to-south
river system in eastern Tibet during Late Cretaceous to Paleogene times created abundant,
low-relief surfaces during a time of relative stability (or in the face of ongoing, but slow rock
uplift) prior to the Indo-Aisan collision and the main Himalayan orogeny. The authors support
this scenario by comparing different data sets from these terranes: contrasts in cooling
histories from the proposed river corridor (versus the bounding terranes); contrasts in detrital
mineral compositional abundances and U/Pb zircon cooling ages within the “drainage
corridor” versus outside of it; mixing models that optimize inputs from diverse source areas in
order to “match” the observed age abundances; etc.

The cooling histories of the compiled thermochronological records (Figure 1b) make a
rather persuasive case that slow Late Cretaceous to Early Tertiary cooling in Songpan-Garzi
and Yidun terranes contrasts markedly with the regions of significantly more rapid Late
Cretaceous-Early Tertiary cooling to the east and west of these terranes. Hence, while
considerable rock uplift, erosion, and bedrock cooling was going on to the east and west.
during Late Cretaceous to Paleogene times, this north-south corridor in eastern Tibet appears

quite stable. To support their interpretation of an integrated fluvial system draining southward
to the Neotethyan ocean, the authors combine paleocurrent analysis with detrital mineralogy
and detrital zircon U/Pb cooling ages to show a noteworthy consistency among dated
sampling sites spanning ~600 km from north to south along the proposed fluvial corridor. For
me, the match between (i) the detrital zircon ages from the Songpan-Ganzi and Yidun terranes
(proposed source areas in the north) with (ii) the suite of consistent detrital zircon ages from
depositional basins spanning 600-750 km from north to south provides critical support to their
hypothesized drainage basin geometry and the proposed timing of its existence as an
integrated depositional system. To me, this spatial-temporal consistency is a key factor
supporting the interpretation offered by these authors.

I suspect that, for some readers, examination of the extensive supplementary data will be
needed to convince them of the validity of the authors' hypotheses. I find both (1) their multi-
dimensional scaling plots of detrital data sets and potential source areas and (2) their modeled
relative contributions from potential source areas quite persuasive.

I note that previous reviews brought up many specific issues and questions, commonly
related to the characteristics of a given source area and an alternative interpretation. I am not
qualified to judge the merits (or validity) of these objections. But, I did find that this
contribution's authors gave quite convincing justifications for their choices and interpretations.
Overall, this provocative, innovative synthesis and interpretation provides a potential
resolution to a long-standing problems related to how there low-relief, high-elevation surfaces
in SE and Eastern Tibet developed. I believe it is worthwhile to get this data set and
interpretation "out there" for the interested audience to contemplate and to try to test with new
data or re-analysis. I also think that the "problem" that this paper addresses is a long-standing
and puzzling one: a problem that has come into clearer focus in recent decades as (i)
high-resolution digital topography has become available of even remote or restricted areas
(thereby enabling clear topographic syntheses, comparisons across regions, and identification
of "anomalies") and (ii) as high-resolution, low-cost, and high-throughput analytical
techniques have enable thousands of analyses to be made and synthesized, and (iii) as
improved and diversified numerical modeling approaches have enabled more rigorous
evaluation of hypotheses. This contribution from the edge of Tibet exploits all of these
technologies in a creative synthesis that is sure to inspire (and provoke) further research
focused on the evolution of large-scale dynamic orogens

**Reply:** We sincerely thank reviewer for his/her positive and constructive comments about our
work. We have revised the manuscript based on the reviewer' comments and suggestions.

Note to editors/authors: I show my "linguistic/grammatical/clarification" suggestions below
in red text.

Note that the Yangtze River is not identified / labelled in any figure that I could find!

**Reply:** Thanks, "the Yangtze River" has been labeled in Fig. 1a.

"comprise" means" to be composed of" So eastern Tibet comprises these provinces, not
the other way around.

**Reply:** Yes, it is our original idea that eastern Tibet comprises these provinces (e.g.,
Songpan-Ganzi and Yidun terranes).

with sustained topography topographic relief

**Reply:** Correction has been made in the revised manuscript.

SUCH regional differences in erosion/exhumation rates could be explained by a
**Reply:** Correction has been made in the revised manuscript.
along the foredeep depozone (i.e., SW Sichuan, Xichang, Huili, and Chuxiong basins: Fig.
1a)

**Reply:** Correction has been made in the revised manuscript.
139-41 Several meters to tens of meters “Thick, cross-stratified sandstone beds with cross-
stratification represent channel deposits of southward-flowing, low-energy (“energy” or
“gradient”?) rivers and associated floodplains, and/or exorheic lakes. I don’t think that
crevasse splays or lake deposits should be cited as indicative of overall paleocurrent
directions for large river systems, especially for thick sandstone deposits. What indicates that
these rivers are “low-energy”? Are there complete channel cross sections and longitudinal
sections to enable you to deduce “energy” versus gradient or simply associated grain size?

**Reply:** Thanks for this valuable comment. We have reorganized this paragraph (revised
manuscript, lines 135–145). Grain size of late Cretaceous–early Palaeogene fluvial deposits
generally ranges from fine to coarse sand, and only a few thin, clast-supported conglomerate
layers were locally found. Cross-beddings within sandstone bodies are commonly low- angle.
More importantly, finer facies including floodplains, lacustrine, and crevasse splays are
widespread in studied basins (Fig. R1). These observations appear to testify to low-energy
current-driven sedimentation. But honestly, we have not done targeted sedimentological work
(e.g., detailed analysis of outcrop architecture) to deduce “energy” of river flow, thus
the “low-energy” is not mentioned in the revised manuscript.

The paleocurrent orientations presented this study were primarily determined from
cross-stratified sandstones and ripple beddings within fluvial sandstone units (please see Deng
et al., 2018). Moreover, a dominantly southward paleo-flow direction has also been revealed
by limited pebbly sandstones and conglomerate of basal granule lag from late Cretaceous
fluvial deposits in the Chuxiong Basin (Xue et al., 2019), further supporting an existence of a
south-flowing palaeo-drainage system.

**References**

Xue C, D., Xiang K., Hu, T. Y., et al. Sedimentary Environments of Late Cretaceous Ore-bearing
Sequences at the Guihua Copper Ore Field in the Northern Chuxiong Basin, Yunnan Province, SW
China. *Acta Sedimentologica Sinica*, **37**, 491-501 (2019). doi: 10.14027/j.issn.1000-0550.2018.153
“Leshan section” in the Sichuan Basin, (given that the Leshan section is not identified in
the figure.

**Reply:** In Fig. 1a, the abbreviation of “LS” represents “Leshan section” (please see caption of
Fig. 1). Please note that we used the “Leshan area” instead of the “Leshan section” in the
revised manuscript.

“genuine”?? Does this mean “statistically significant”?

**Reply:** Yes, correction has been made in the revised manuscript as reviewer' suggestion.

173-5 “The consistent provenance signal from the different basins requires the existence of a
continuous fluvial system during the late Cretaceous–early Palaeogene.” Does it
truly “require”? That may well be the most likely scenario, but it doesn’t “require” this
scenario, in my opinion.

**Reply:** We agree with the comment on the use of “require”, and replace it with “argue for”
(revised manuscript, line 205).

showing main tectonic units “(red text)” (in reference to what represents these units in the

figure) and major river systems (except for the Yangtze! Add a label for this river!)

**Reply:** Thank you for this helpful comment. Correction of caption has been made in the
revised manuscript as reviewer' suggestion; and “the Yangtze River” has been labeled in Fig.
1a.

low-relief plateau areas 24, 31–34, 36, 58. New suggested text: Hexagons indicate sites
with Cretaceous-Tertiary cooling histories (shown in 1B) Dashed

**Reply:** This helpful sentence has been added in the revised manuscript, as suggested by
reviewer.

Comments on Figures

Figure 1a. Nowhere is the Yangtse River labeled on this figure. Its name should be clearly
identified. Note that in the figure caption, no description is given of the light blue vertical
band from 40-50 Ma in Fig 1b. What is that? I presume it's a proposed “boundary” between
rapid L Cret-Paleocene cooling versus post-50-Ma rapid cooling. Why not add blue or red
labels for rapid cooling pre-50 Ma and post-40 Ma, respectively? I also note that the chosen
level of transparency of some of the “yellow” low-relief surfaces makes these surfaces appear
to be a different color (more orange than yellow, where superimposed above the orange
swath), and that there is a strange (inconsistent?) mix of yellow and orange surfaces just
above the red “5” in the figure. These issues should all be readily corrected. Could a label be
added to the orange region so that it's more self-explanatory? Similarly, a label on the
blue-dashed line (“hypothesized drainage divide”) would make its significance more obvious.

**Reply:** The reviewer brings up some important points. As suggested by reviewer, (i) “the
Yangtze River” has been labeled in Fig. 1a; (ii) the meaning/definition of light blue vertical
band has been included in revised caption of Fig. 1; (iii) the color of low-relief surfaces
(yellow) within late Triassic to early Cretaceous foredeep depozone (orange) has
appropriately been adjusted; (iv) in order to make its significance more obvious, we also
added labels of “hypothesized drainage divide” and “late Triassic to early Cretaceous foredeep
depozone” on the modified Fig. 1b.

Figure 1b. Add a legend indicating blue lines for rapid cooling prior to 50 Ma, versus red lines
for rapid cooling since 50 Ma.

**Reply:** The legend showing definitions of blue and red lines has been added in Fig. 1b.

Figure 2. How about helping your readers along with a title box, “Drainage Scenarios” and
cryptic summary titles for each scenario's panel?

**Reply:** Thank you for this valuable comment, succinct summary titles for four scenario's
panels have been added in Fig. 2.

Figure 3. The rationale is unstated for the red and blue lines for the probability density
functions. Please make that clear!

**Reply:** We have expressed rationale of the red and blue lines for the probability density
functions.

Comments on Supplemental Figures

Supp Figure 3. Illustrative and quite compelling figure!! Spell out the names of the sections
(CX, HL, etc) for each group of samples.

**Reply:** As suggested by reviewer, full names of each studied basin/area were presented.

Reviewers' Comments:

Reviewer #2:

Remarks to the Author:

Dear Editor,

At your request, I revisited the concerns of Reviewer 1 and the authors' rebuttal relating to comments 5, 6, and 15. For all three of the comments, I consider the authors' responses sufficient and sound.

In comment 5, Reviewer 1 raises concerns with biostratigraphic dating techniques and argues that MDAs from DZs should be used instead. This is supported by reference to another study in which there was disagreement and DZ data more closely approximated the true depositional age. I agree with the authors' rebuttal, which argued that the lack of young DZ ages justified their reliance on biostratigraphic data. I found that the justification for the age interpretations in the supplementary files was thorough, transparent, and appropriately referenced. Furthermore, the provenance interpretations made by the authors explain why there is a lack of DZ ages that approximate the true depositional age. My personal experience working with DZ data tells me that MDAs are often in disagreement with the TDA, even when large-n datasets are used. This is mainly a function of the tectonic setting, not the number of analyses.

In comment 6, Reviewer 1 implies that the authors should conduct large-n DZ dating (300 or more analyses) so that DZ data could be used to calculate MDAs. This comment implies that the authors missed the younger age components because they only dated ~100-150 grains per sample. I disagree with Reviewer 1 in this case. According to Vermeesch (2004), DZ studies can be 95% confident that no population of grains constituting 5% or more of the source was missed when a threshold of 117 analyses is achieved. The authors generally met this requirement, indicating that if there were young populations that were missed, they were likely a minor source (<5%). Furthermore, the authors dated several different samples, most of which yielded very similar results. The probability that the young population was missed in each of the datasets would be much less than 5%. Finally, the authors compiled some data from the literature, bringing the number of grains for the composite sample datasets to >300 in most cases.

In comment 13, Reviewer 1 requested weighted mean ages for the standards so that results from this study can be compared to standard ages. The authors sufficiently addressed this request and comparison indicates the data are of high precision and accuracy.

Respectfully,

Dr. Andrew K Laskowski

Reviewer #3:

Remarks to the Author:

The authors provided a detailed response to my previous comments, but unfortunately, I am not fully convinced by the authors in two aspects.

First, the authors presented some evidence to support a fluvial origin of the deposit in the studied basins, but those sedimentary structures could also develop in shallow lake or floodplain environment, which do not require a perennial river as the author suggested.

Second, regarding the southward-flowing river to the South China Sea (Clift et al., 2020) or to the Neo-Tethyan Ocean (This study), the authors' response is that "a paleo-Jinsha River flowed south to the South China Sea from the SE Tibetan Plateau following the initial surface uplift of eastern Tibet in

the late Eocene (ca. 35 Ma) (Fig. R15). Yet our proposed continental-scale fluvial system of this study existed during late Cretaceous to early Paleogene (ca. 100–50 Ma), predating the India–Eurasia collision and plateau uplift.” And therefore no contradiction between each other. However, this argument largely relies on the reliability of the “late Cretaceous-early Paleogene” age of the studied basins. I understand the difficulty to precisely constrain the age of these red bed basins, and I acknowledge that the authors have tried their best to constrain the age of these basins, but as acknowledged by the authors, the exact ages of these deposits are not yet known, and we cannot preclude the possibility that the deposits in the basins, e.g., Huili, Xichang, can extend into late Eocene. If this was the case, then the southward-flowing river from the Sichuan basin would join the paleo-Red River to the South China Sea as Clark et al. (2004) suggested.

At last, I am also curious what is the difference between “a dendritic paleo-Red River to the South China Sea” and “a continental-scale paleo-drainage system to the Neo-Tethyan Ocean” on the formation of a low-relief landscape?

Reviewer #4:

Remarks to the Author:

Review of “Existence of a continental-scale river system in eastern Tibet during the late Cretaceous–early Palaeogene”

I am assessing this paper on the basis of its integrated regional data, its innovative interpretations, and the potential impact of those interpretations on our understanding of some large-scale geomorphic anomalies along the eastern and SE margin of the Tibetan Plateau. This approach likely stands in contrast to some other reviewers who are more familiar with details of the local geologic-stratigraphic-petrographic-chronologic data sets that this submission exploits and builds upon. The key scientific question that this paper addresses is the origin of the widespread, high-altitude, low-relief surfaces that characterize much of the SE Tibetan Plateau: a region where local relief at present is typically quite high, with deeply incised river gorges, and rock-uplift rates are rapid in a global context. The presence of these long-lived, low-relief surfaces at rather high altitudes has long piqued our curiosity: why are they there; how did they form; what are modern analogues of their formative sequence; what data can be used to test various hypotheses?

The authors argue that, despite some coeval tectonic uplift, a long-lived, ~north-to-south river system in eastern Tibet during Late Cretaceous to Paleogene times created abundant, low-relief surfaces during a time of relative stability (or in the face of ongoing, but slow rock uplift) prior to the Indo-Aisan collision and the main Himalayan orogeny. The authors support this scenario by comparing different data sets from these terranes: contrasts in cooling histories from the proposed river corridor (versus the bounding terranes); contrasts in detrital mineral compositional abundances and U/Pb zircon cooling ages within the “drainage corridor” versus outside of it; mixing models that optimize inputs from diverse source areas in order to “match” the observed age abundances; etc.

The cooling histories of the compiled thermochronological records (Figure 1b) make a rather persuasive case that slow Late Cretaceous to Early Tertiary cooling in Songpan-Garzi and Yidun terranes contrasts markedly with the regions of significantly more rapid Late Cretaceous-Early Tertiary cooling to the east and west of these terranes. Hence, while considerable rock uplift, erosion, and bedrock cooling was going on to the east and west. during Late Cretaceous to Paleogene times, this north-south corridor in eastern Tibet appears quite stable. To support their interpretation of an integrated fluvial system draining southward to the Neotethyan ocean, the authors combine paleocurrent analysis with detrital mineralogy and detrital zircon U/Pb cooling ages to show a noteworthy consistency among dated sampling sites spanning ~600 km from north to south along the proposed fluvial corridor. For me, the match between (i) the detrital zircon ages from the Songpan-Ganzi and Yidun terranes (proposed source areas in the north) with (ii) the suite of consistent detrital zircon ages from depositional basins spanning 600–750

km from north to south provides critical support to their hypothesized drainage basin geometry and the proposed timing of its existence as an integrated depositional system. To me, this spatialtemporal consistency is a key factor supporting the interpretation offered by these authors. I suspect that, for some readers, examination of the extensive supplementary data will be needed to convince them of the validity of the authors' hypotheses. I find both (1) their multidimensional scaling plots of detrital data sets and potential source areas and (2) their modeled relative contributions from potential source areas quite persuasive. I note that previous reviews brought up many specific issues and questions, commonly related to the characteristics of a given source area and an alternative interpretation. I am not qualified to judge the merits (or validity) of these objections. But, I did find that this contribution's authors gave quite convincing justifications for their choices and interpretations. Overall, this provocative, innovative synthesis and interpretation provides a potential resolution to a long-standing problems related to how there low-relief, high-elevation surfaces in SE and Eastern Tibet developed. I believe it is worthwhile to get this data set and interpretation "out there" for the interested audience to contemplate and to try to test with new data or reanalysis. I also think that the "problem" that this paper addresses is a long-standing and puzzling one: a problem that has come into clearer focus in recent decades as (i) high-resolution digital topography has become available of even remote or restricted areas (thereby enabling clear topographic syntheses, comparisons across regions, and identification of "anomalies") and (ii) as high-resolution, low-cost, and high-throughput analytical techniques have enable thousands of analyses to be made and synthesized, and (iii) as improved and diversified numerical modeling approaches have enabled more rigorous evaluation of hypotheses. This contribution from the edge of Tibet exploits all of these technologies in a creative synthesis that is sure to inspire (and provoke) further research focused on the evolution of large-scale dynamic orogens. Note to editors/authors: I show my "linguistic/grammatical/clarification" suggestions below in red text.

58 Note that the Yangtze River is not identified/labelled in any figure that I could find!

"comprise" means "to be composed of" So eastern Tibet comprises these provinces, not the other way around.

with sustained topography topographic relief

SUCH regional differences in erosion/exhumation rates could be explained by a

along the foredeep depozone (i.e., SW Sichuan, Xichang, Huili, and Chuxiong basins: Fig. 1a)

139-41 Several meters to tens of meters "Thick, cross-stratified sandstone beds with crossstratification represent channel deposits of southward-flowing, low-energy ("energy" or "gradient"?) rivers and associated floodplains, and/or exorheic lakes I don't think that crevasse splays or lake deposits should be cited as indicative of overall paleocurrent directions for large river systems, especially for thick sandstone deposits. What indicates that these rivers are "low-energy"? Are there complete channel cross sections and longitudinal sections to enable you to deduce "energy" versus gradient or simply associated grain size?

"Leshan section" in the Sichuan Basin, (given that the Leshan section is not identified in the figure.

"genuine"?? Does this mean "statistically significant"?

173-5 "The consistent provenance signal from the different basins requires the existence of a continuous fluvial system during the late Cretaceous-early Palaeogene." Does it truly "require"? That may well be the most likely scenario, but it doesn't "require" this scenario, in my opinion.

showing main tectonic units "(red text)" (in reference to what represents these units in the figure) and major river systems (except for the Yangtze! Add a label for this river!)

low-relief plateau areas^{24,31-34,36,58}. New suggested text: Hexagons indicate sites with Cretaceous-Tertiary cooling histories (shown in 1B) Dashed

Comments on Figures

Figure 1a. Nowhere is the Yangtse River labeled on this figure. Its name should be clearly identified. Note that in the figure caption, no description is given of the light blue vertical band from 40-50 Ma in Fig 1b. What is that? I presume it's a proposed "boundary" between rapid L Cret-Paleocene cooling versus post-50-Ma rapid cooling. Why not add blue or red labels for rapid cooling pre-50 Ma and post-40 Ma, respectively? I also note that the chosen level of transparency of some of the "yellow" low-relief surfaces makes these surfaces appear to be a different color (more orange than yellow, where superimposed above the orange swath), and that there is a strange (inconsistent?) mix of yellow and orange surfaces just above the red "5" in the figure. These issues should all be readily corrected. Could a label be added to the orange region so that it's more self-explanatory? Similarly, a label on the blue-dashed line ("hypothesized drainage divide") would make its significance more obvious.

Figure 1b. Add a legend indicating blue lines for rapid cooling prior to 50 Ma, versus red lines for rapid cooling since 50 Ma.

Figure 2. How about helping your readers along with a title box, "Drainage Scenarios" and cryptic summary titles for each scenario's panel?

Figure 3. The rationale is unstated for the red and blue lines for the probability density functions. Please make that clear!

Comments on Supplemental Figures

Supp Figure 3. Illustrative and quite compelling figure!! Spell out the names of the sections (CX, HL, etc) for each group of samples.

Response letter

In this letter we provide our detailed response (in blue text) to the comments of the three reviewers (in black) and explain all changes performed on the manuscript.

Response to the comments of the three reviewers

Reviewer #2

Dear Editor,

At your request, I revisited the concerns of Reviewer 1 and the authors' rebuttal relating to comments 5, 6, and 15. For all three of the comments, I consider the authors' responses sufficient and sound. In comment 5, Reviewer 1 raises concerns with biostratigraphic dating techniques and argues that MDAs from DZs should be used instead. This is supported by reference to another study in which there was disagreement and DZ data more closely approximated the true depositional age. I agree with the authors' rebuttal, which argued that the lack of young DZ ages justified their reliance on biostratigraphic data. I found that the justification for the age interpretations in the supplementary files was thorough, transparent, and appropriately referenced. Furthermore, the provenance interpretations made by the authors explain why there is a lack of DZ ages that approximate the true depositional age. My personal experience working with DZ data tells me that MDAs are often in disagreement with the TDA, even when large-n datasets are used. This is mainly a function of the tectonic setting, not the number of analyses.

In comment 6, Reviewer 1 implies that the authors should conduct large-n DZ dating (300 or more analyses) so that DZ data could be used to calculate MDAs. This comment implies that the authors missed the younger age components because they only dated ~100-150 grains per sample. I disagree with Reviewer 1 in this case. According to Vermeesch (2004), DZ studies can be 95% confident that no population of grains constituting 5% or more of the source was missed when a threshold of 117 analyses is achieved. The authors generally met this requirement, indicating that if there were young populations that were missed, they were likely a minor source (<5%). Furthermore, the authors dated several different samples, most of which yielded very similar results. The probability that the young population was missed in each of the datasets would be much less than 5%. Finally, the authors compiled some data from the literature, bringing the number of grains for the composite sample datasets to >300 in most cases.

In comment 13, Reviewer 1 requested weighted mean ages for the standards so that

results from this study can be compared to standard ages. The authors sufficiently addressed this request and comparison indicates the data are of high precision and accuracy.

Respectfully,

Dr. Andrew K Laskowski

Reply: We thank Dr. Andrew Laskowski for his detailed comments on the three points. In the “Methods” section, we added the following sentence for clarification: “In this study, all samples from the different basins show consistent detrital zircon components, and each sample yielded 68–123 concordant ages, which generally meets statistical requirements” (lines 349–351 of revised manuscript).

Reviewer #3

The authors provided a detailed response to my previous comments, but unfortunately, I am not fully convinced by the authors in two aspects.

First, the authors presented some evidence to support a fluvial origin of the deposit in the studied basins, but those sedimentary structures could also develop in shallow lake or floodplain environment, which do not require a perennial river as the author suggested.

Reply: Indeed, sedimentary structures such as small-scale cross stratifications or climbing ripple lamination can also be observed in siltstone to mudstone beds of shallow-lacustrine or floodplain facies. However, the presence of meter to tens of meter thick amalgamated sandstone beds which are laterally continuous over scales of hundreds of meters, indicate a fluvial origin for these sedimentary deposits. Characteristic fluvial sedimentary structures include medium- to large-scale cross stratifications, upward fining sequences, basal granule lags, and lenticular or tabular sandstone beds with erosional contacts. Moreover, paleocurrent directions based on trough and tabular cross-beds are consistently indicating a flow to the south or southeast, further corroborating the existence of a perennial fluvial system. Likewise, the sedimentological descriptions and interpretations of the depositional environment from geological surveys (as summarized in the Supplementary Information) indicate that the fluvial-lacustrine facies associations and the freshwater fossil assemblages of

the studied basins are best interpreted as typical components of a large river system. We re-iterate that hydrologically open lake basins (Carroll and Bohacs, 1999; Bohacs et al., 2000) and floodplains are integral parts of large river systems, as can be seen along the largest rivers of the world (Miall et al., 1996; Chen et al., 2007; Ashworth et al., 2012). These arguments, together with the lack of thick evaporite sequences, make the existence of a large throughgoing river system very likely. As we already addressed this issue in the last revision of the manuscript (lines 135–153 of revised manuscript), we did not make further changes.

References:

- Ashworth, P. J. & Lewin, J. How do big rivers come to be different? *Earth Sci Rev.* **114**, 84–107 (2012).
- Bohacs, K. M., A. R. Carroll, J. E. Neal, P. J. Mankiewicz. Lake-basin type, source potential, and hydrocarbon character: an integrated-sequence-stratigraphic–geochemical framework, in E. H. Gierlowski-Kordesch and K. R. Kelts, eds., *Lake basins through space and time: AAPG Studies in Geology* 46, p. 3–34 (2000).
- Carroll, A. R., Bohacs, K.M. Stratigraphic classification of ancient lakes: balancing tectonic and climatic controls. *Geology* **27**, 99–102 (1999).
- Chen, Z., Xu, K., Watanabe, M. Dynamic hydrology and geomorphology of the Yangtze River. In: Gupta, A. (Ed.), *Large Rivers: Geomorphology and Management*. Wiley, Chichester, pp. 457–469, (2007).
- Miall, A. D. *The Geology of Fluvial Deposits: Sedimentary Facies, Basin Analysis, and Petroleum Geology*, Springer (1996).

Second, regarding the southward-flowing river to the South China Sea (Clift et al., 2020) or to the Neo-Tethyan Ocean (This study), the authors’ response is that “a paleo-Jinsha River flowed south to the South China Sea from the SE Tibetan Plateau following the initial surface uplift of eastern Tibet in the late Eocene (ca. 35 Ma) (Fig. R15). Yet our proposed continental-scale fluvial system of this study existed during late Cretaceous to early Paleogene (ca. 100–50 Ma), predating the India–Eurasia collision and plateau uplift.” And therefore no contradiction between each other. However, this argument largely relies on the reliability of the “late Cretaceous-early Paleogene” age of the studied basins. I understand the difficulty to precisely constrain the age of these red bed basins, and I acknowledge that the authors have tried their best to constrain the age of these basins, but as acknowledged by the authors, the exact ages of these deposits are not yet known, and we cannot preclude the

possibility that the deposits in the basins, e.g., Huili, Xichang, can extend into late Eocene. If this was the case, then the southward-flowing river from the Sichuan basin would join the paleo-Red River to the South China Sea as Clark et al. (2004) suggested.

Reply: As acknowledged by the reviewer, the exact ages of these deposits are difficult to constrain, but young (i.e., late Eocene or younger) zircons are completely lacking in our samples. Given that late Eocene plutons are common across southeastern Tibet (e.g., Lu et al., 2012; Deng et al., 2014), the absence of late Eocene zircon ages implies that the studied continental red-beds are older than late Eocene. Still, we cannot preclude the possibility mentioned by the reviewer that the large-scale river discharged to the Proto-South China Sea at a late stage of the K₂–E₁ time interval. However, in the current depositional area of the Red River (the Yinggehai-Song Hong Basin of the South China Sea), many boreholes revealed that the Cenozoic deposits at the bottom of the boreholes are not older than late Eocene (~35 Ma) (e.g., Clift et al., 2006, 2008). Regionally, the Proto-South China Sea during the late Cretaceous to early Cenozoic period is characterized by a series of deep, rapidly-subsiding small-scale rift basins that formed during back-arc extension (see review of Morley et al., 2012). This tectonic setting makes it difficult to test the hypothesis suggested by the reviewer. Considering these uncertainties and the reviewer's concern, we added the following sentence to the manuscript (lines 255–259) **“It is also possible that the paleo-drainage network discharged into the Neo-Tethyan Ocean during the late Cretaceous and Palaeocene, but later changed its course as a result of the India-Asia collision, and flowed into the proto-South China Sea starting in late Eocene.”**

References:

- Clift, P. D., Blusztajn, J. & Nguyen, A. D. Large- scale drainage capture and surface uplift in eastern Tibet- SW China before 24 Ma inferred from sediments of the Hanoi Basin, Vietnam. *Geophys. Res. Lett.* **33**, L19403 (2006).
- Clift, P. D., Long, H. V., Hinton, R., Ellam, R. M., Hannigan, R., Tan, M. T. et al. Evolving east Asian river systems reconstructed by trace element and Pb and Nd isotope variations in modern and ancient Red Rive-Song Hong sediments. *Geochem., Geophys., Geosyst.* **9**, Q04039 (2008).
- Deng, J., Wang, Q. F., Li, G. J. & Santosh, M. Cenozoic tectono-magmatic and metallogenic processes in the Sanjiang region, southwestern China. *Earth-Science Reviews* **138**, 268–299 (2014).

- Lu, Y.J., Kerrich, R., Cawood, P.A., McCuaig, T.C., Hart, C.J.R., Li, Z.X., Hou, Z.Q., Bagas, L. Zircon SHRIMP U–Pb geochronology of potassic felsic intrusions in western Yunnan, SW China: constraints on the relationship of magmatism to the Jinsha su-ture. *Gondwana Res.* **22**, 737–747 (2012).
- Morley, C.K. Late Cretaceous–Early Palaeogene tectonic development of SE Asia. *Earth Sci Rev.* **115**, 27–75 (2012).

At last, I am also curious what is the difference between “a dendritic paleo-Red River to the South China Sea” and “a continental-scale paleo-drainage system to the Neo-Tethyan Ocean” on the formation of a low-relief landscape?

Reply: A dendritic palaeo-Red River to the South China Sea, as suggested by Clark et al. (2004), would be largely controlled by a regionally eastward-tilt of the topography, possibly resulting from plateau uplift. Such a tectonic setting makes it difficult to develop a large-scale low-relief surface. In contrast, our proposed continental-scale paleo-drainage system to the Neo-Tethyan Ocean follows an inherited long-wavelength topographic depression (as explained on lines 105–111 of the revised manuscript), which provides more favourable boundary conditions for the formation of the extensive low-relief landscape.

Reference: Clark, M. K., Schoenbohm, L. M., Royden, L. H., Whipple, K. X., Burchfiel, B. C., Zhang, X., Tang, W., Wang, E. & Chen, L. Surface uplift, tectonics, and erosion of eastern Tibet from large-scale drainage patterns. *Tectonics* **23**, TC1006 (2004).

Reviewer #4

Review of “*Existence of a continental-scale river system in eastern Tibet during the late Cretaceous–early Palaeogene*”.

Reply: As explained in the Cover Letter, the comments we received from Reviewer #4 were those of the last round of reviews. We believe that we satisfactorily addressed these comments in our last revision and did therefore not change anything in this current revision of the manuscript.